

# Cover your bases: asymptotic distributions of the profile likelihood ratio when constraining effective field theories in high-energy physics

**Florian Urs Bernlochner[1]⋆, Daniel C. Fry[1], Stephen Burns Menary[2]† and Eric Persson[1]**

**1** Physikalisches Institut der Rheinischen Friedrich-Wilhelms-Universität Bonn,
Nußallee 12, 53115 Bonn, Germany
**2** Department of Physics and Astronomy, University of Manchester, Oxford Rd,
Manchester, M13 9PL, United Kingdom

⋆ florian.bernlochner@uni-bonn.de , † sbmenary@gmail.com

## Abstract

We investigate the asymptotic distribution of the profile likelihood ratio (PLR) when constraining effective field theories (EFTs) and show that Wilks' theorem is often violated, meaning that we should not assume the PLR to follow a $\chi^2$-distribution. We derive the correct asymptotic distributions when either one or two real EFT couplings modulate observable cross sections with a purely linear or quadratic dependence. We then discover that when both the linear and quadratic terms contribute, the PLR distribution does not have a simple form. In this case we provide a partly-numerical solution for the one-parameter case. Using a novel approach, we find that the constants which define our asymptotic distributions may be obtained experimentally using a profile of the Asimov likelihood contour. Our results may be immediately used to obtain the correct coverage when deriving real-world EFT constraints using the PLR as a test-statistic.

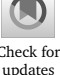

# 1  Introduction

Effective field theories (EFTs) have become a powerful tool in the search for new high-energy physics. They are now an important part of many experimental programs at the Large Hadron Collider (LHC) and phenomenological analyses beyond (see e.g. [1,2] for recent summaries). Much of their popularity may be attributed to their versatility, being applicable to a wide variety of channels, their strong theoretical backing (see e.g. [3,4]), and the availability of practical tools to aid their use in-the-wild [4–9].

A common application of an EFT is to parameterise all the possible ways in which new physics at some high-energy interaction scale $\Lambda_{\text{EFT}}$ may indirectly modify the event rates and kinematic distributions observed when experimentally probing a lower scale $\Lambda \ll \Lambda_{\text{EFT}}$. After measuring e.g. a suite of differential cross sections, we may use the profile likelihood ratio

(PLR) as a test-statistic with which to perform frequentist hypothesis tests over the EFT parameter space. Such analyses have two primary aims: (i) to search for deviations from Standard Model (SM) behaviour which indicate the presence of new physics, obtaining a description of its low energy behaviour, and (ii) to constrain high-energy models based on their expected contributions at the experimental interaction scale. We do not assume any specific UV-complete model, instead performing a general search for anomalies in such a way that specific models may be constrained after-the-fact if desired.

To compute frequentist confidence limits, we must know how the PLR is expected to be distributed under all reasonable parameter hypotheses. A common practice is to assume that this follows an asymptotic form, and ideally use Monte Carlo estimation at several points in parameter space to validate this assumption. Following Wilks' theorem [10], it is common to assume that the PLR is expected to follow a $\chi^2$-distribution with degrees of freedom equal to the number of parameters profiled. However, this approach is not always correct, and in fact **the asymptotic distribution of the PLR is not a $\chi^2$-distribution when the EFT parameterisation is not dominated by the linear component**. This is due to the breakdown of a key condition of Wilks' theorem: that the best-fit statistical model lies in the bulk of the function space profiled, not near to its boundaries. To address this, in this work we

1. Explain when and why the PLR may not be assumed to follow a $\chi^2$-distribution.

2. Provide the correct asymptotic distributions when one or two EFT parameters are profiled, each providing a purely linear or quadratic contribution.

3. Discover why a simple distribution does not exist when a single EFT parameter is sensitive to both the linear and quadratic contributions. In this case we derive a partly-numerical solution for deriving the expected PLR distribution.

4. Highlight opportunities for future work to extend our understanding further.

This work is motivated by the following idea: the power and rigour of a modern high-energy physics analysis are defined by both the quality of the experimental measurement *and* the quality of the statistical analysis performed on it. We spend much time and money on performing world-leading measurements, and should also invest in ensuring that the statistical analysis is as powerful and rigorous as possible.

## 1.1 Background

In a high-energy physics context, Algeri et al [11] discuss the ways in which Wilks' theorem may break down, and Cowan et al [12] present various asymptotic distributions of the PLR and PLR-like test statistics or Fowlie et al present a fast algorithm to determine $p$-values [13]. However, similar effects have been encountered elsewhere in the statistics literature. Chernoff [14] was the first to derive the asymptotic distribution of the likelihood ratio test for the location parameter of a Gaussian random variable when the true parameter is on a boundary of the parameter space, showing that it behaves like a random variable with half the probability mass concentrated at zero and the rest $\chi^2$-distributed with one degree of freedom. Moran [15] and Chant [16] investigate the relationship between optimal tests and tests based on maximum likelihood estimators under similar conditions. Self and Liang [17] summarize and extend the earlier results and is particularly readable with examples covering many common situations that practitioners may face. Feng [18] provides method for constructing confidence regions that is easy to use and has asymptotically correct coverage probability. Similarly to the high-energy physics community, other fields have independently investigated the issue, with particular attention paid to the specific idiosyncracies of each respective field, e.g. financial econometrics (Andrews [19]) and biostatistics (Pinheiro and Bates [20]).

## 1.2 Overview of EFT Fits

Let us consider new physics (NP) contributions that are only weakly coupled at the experimental interaction scale. We then consider the leading order case in which all diagrams with two or more new physics couplings are expected to contribute a small effect and are neglected. The expected cross section is proportional to the squared transition amplitude

$$\sigma(f;c) \propto \left| \mathcal{M}_{\text{SM}}(f) + \sum_{c_\alpha \in c} c_\alpha \mathcal{M}_{\text{NP}}^{(c_\alpha)}(f) \right|^2, \tag{1}$$

where $f$ labels some final state property, $\mathcal{M}_{\text{SM}}$ is the transition amplitude considering diagrams with only Standard Model couplings, $\mathcal{M}_{\text{NP}}^{(c_\alpha)}$ considers diagrams which contain one NP coupling modulated by real coupling strength $c_\alpha \in \mathbb{R}$, and $c$ is the set of all relevant coupling strengths indexed by $\alpha$. We note that a complex coupling strength can always be expanded into one real and one imaginary part each modulated by real coefficients, and so this treatment remains general. Expanding and collecting terms, we find a quadratic parameterisation

$$\sigma(f;c) = s(f) + \sum_\alpha c_\alpha l_\alpha(f) + \sum_{\alpha,\beta \neq \alpha} c_\alpha c_\beta t_{\alpha,\beta}(f) + \sum_\alpha c_\alpha^2 n_\alpha(f), \tag{2}$$

where all $l_\alpha$, $t_{\alpha,\beta}$ and $n_\alpha$ are functions of $f$ only. By collecting the final states into a series of differential bins which integrate over the final state property, we can write

$$\mu(c) = s + \sum_\alpha c_\alpha l_\alpha + \sum_{\alpha,\beta \neq \alpha} c_\alpha c_\beta t_{\alpha,\beta} + \sum_\alpha c_\alpha^2 n_\alpha, \tag{3}$$

where $\mu$ is a vector of cross sections and $\{l_\alpha, t_{\alpha,\beta}, n_\alpha\}$ are individual component vectors which may be evaluated analytically or using Monte Carlo estimation. All elements of $\mu$, s and $n_\alpha$ are positive definite, whereas the elements of $l_\alpha$ and $t_{\alpha,\beta}$ may be either positive or negative.

Consider that $x$ labels a vector of observed cross sections. In the asymptotic limit we assume that this is distributed according to a Gaussian statistical model

$$p_x(x|c) = \frac{1}{\sqrt{2\pi|\Sigma|}} e^{-\frac{1}{2}\chi^2(x;c)}, \tag{4}$$

$$\chi^2(x;c) = (x - \mu(c))^T \Sigma^{-1}(x - \mu(c)), \tag{5}$$

where $\Sigma$ is the covariance matrix. The profile likelihood ratio (PLR) test-statistic is then

$$q(x;c) = -2\ln\left[ \frac{p_x(x|c)}{\max_{c'} p_x(x|c')} \right] = \chi^2(x;c) - \min_{c'} \chi^2(x;c'). \tag{6}$$

To derive the $p$-value for a given value of $c$, after observing data $x$, we must know how $q(x;c)$ is expected to be distributed when $c$ is hypothesised to be the true value $c_{\text{true}}$. Let us denote this important special case $q_{c_{\text{true}}} := q(x;c = c_{\text{true}})$. Our task is to derive the distribution of $q_{c_{\text{true}}}$ for every possible value of $c_{\text{true}}$.

## 2 When does Wilks' theorem not apply?

We now use a simple example to demonstrate when Wilks' theorem does not apply. Here $x$ is the measurement of a single bin with variance $\Sigma = 1$, we have only one Wilson coefficient $c$, and we choose offset units such that $s = 0$. We consider the following two models:

1. Linear case: $\mu(c) = c$, representing models in which which the dependence on new physics is dominated by interference with the Standard Model.

2. Quadratic case: $\mu(c) = c^2$, representing models in which the coupling is purely imaginary or interference is small compared with the pure new physics contribution.

In each case, let us consider the behaviour for $c_{\text{true}} = 0.5$. Fig 1 (left column) shows the expected densities of $x$, where the dotted line represents a Gaussian distribution $\mathcal{G}$ and the shaded histogram shows the distribution of 1M pseudo-datasets, labelled "Toys". The toys should be understood to represent the true distribution of a quantity.

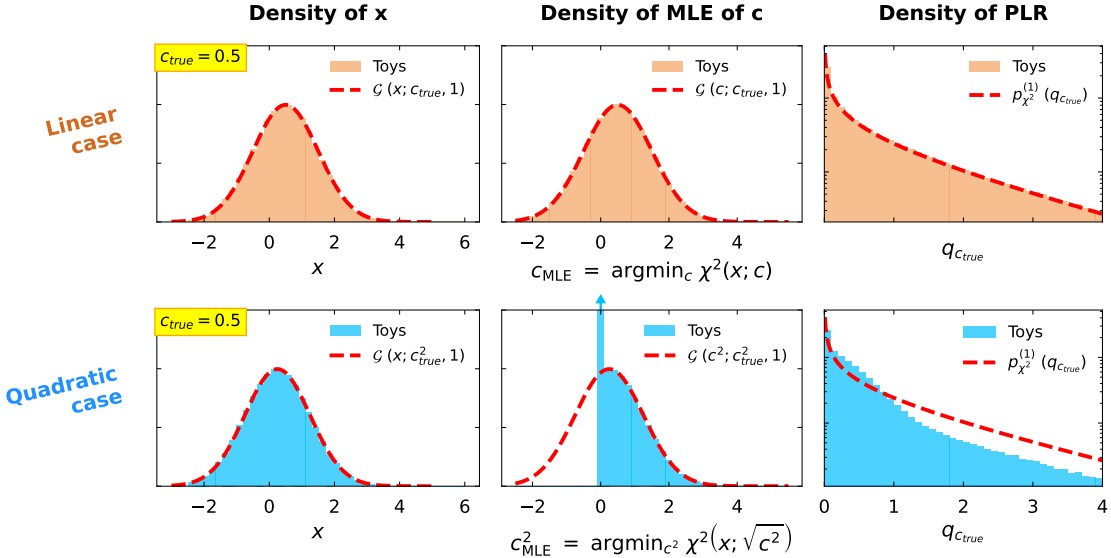

Figure 1: Visualisation of how Wilks' theorem is violated when a parameterisation cannot fit data fluctuations which fall beyond some boundary. This behaviour is observed in quadratic EFT fits (bottom row), which cannot access data below the SM hypothesis.

The second column shows how the maximum likelihood estimates $c_{\text{MLE}}$ and $c_{\text{MLE}}^2$ are distributed. In the linear case, we can minimise the $\chi^2$ by setting $c_{\text{MLE}} = x$ for any value of $x$. Therefore $c_{\text{MLE}}$ follows the same Gaussian distribution as $x$. In the quadratic case, we can only set $c_{\text{MLE}}^2 = x$ for positive values of $x$. Any fluctuations below $x = 0$ result in $c_{\text{MLE}}^2 = 0$. **Since the optimisation step reaches the boundary of its parameter space, Wilks' theorem is violated.** For a quadratic EFT model this boundary will always occur at the Standard Model hypothesis. The expected density is a Gaussian distribution for $c_{\text{MLE}}^2 > 0$ and a delta function at $c_{\text{MLE}}^2 = 0$, with amplitudes fixed by the expected fraction of observations for which $x \geq 0$.

The third column shows how $q_{c_{\text{true}}}$ is distributed. In the linear case, Wilks theorem is valid and $q_{c_{\text{true}}}$ follows a $\chi^2$-distribution with one degree of freedom $p_{\chi^2}^{(1)}(q_{c_{\text{true}}})$. In the quadratic case, the inability to express fluctuations below the Standard Model hypothesis has manifested as a significant deviation from the $\chi^2$-distribution, and Wilks' theorem should not be assumed when computing $p$-values.

## 3 Solution with one (linear or quadratic) parameter

We may now compute the expected distribution of $q_{c_{\text{true}}}$ for linear and quadratic fits with one Wilson coefficient and many differential bins. In the linear case we reproduce the result of

Wilks' theorem, but in the quadratic case we do not. For simplicity let us define $n := n_\alpha$ and $l := l_\alpha$. We do not consider statistical models containing non-Gaussian nuisance parameters, but note that using Gaussian nuisance parameters with fixed-variance (on the cross-sections) is equivalent to including their covariance within the covariance matrix.

The statistical analysis becomes easier if we transform co-ordinates in the following way. Firstly we write $x = z + \mu(c_{\text{true}})$ where $z$ is a multi-dimensional Gaussian random variable with covariance $\Sigma$ centred on 0. Secondly we project all of $\{z, \mu, s, l, n\}$ onto the unit eigenvectors of $\Sigma$ so that every component of $z$ becomes uncorrelated. Finally we normalise each component by the square-root of the corresponding eigenvalue, so that it has unit variance. Using bars to label transformed quantities, in this basis **we obtain that $\bar{z}$ is a vector of independent normally distributed random variables**.

## 3.1 Linear case

For the linear case with $|n| = 0$ and $|l| > 0$, in our transformed basis we find

$$
\begin{aligned}
\chi^2(\bar{z}; c) &= \left| \bar{z} - (c - c_{\text{true}}) \bar{l} \right|^2, \\
q_{c_{\text{true}}} &= |\bar{z}|^2 - \min_{c'} \left[ \left| \bar{z} - (c' - c_{\text{true}}) \bar{l} \right|^2 \right] \\
&\rightarrow \min_{c'}[\dots] \text{ at } c' = c_{\text{true}} + \frac{\bar{Z}_l}{\bar{L}^2} \rightarrow \\
&= \hat{\bar{Z}}_l^2,
\end{aligned}
\tag{7}
$$

where the term in square brackets in minimised by setting the derivative equal to zero and

$$
\bar{L}^2 = \sum_i \bar{l}_i^2, \qquad \hat{\bar{l}}_i = \frac{\bar{l}_i}{\sqrt{\bar{L}^2}}, \qquad \bar{Z}_l = \sum_i \bar{l}_i \bar{z}_i, \qquad \hat{\bar{Z}}_l = \sum_i \hat{\bar{l}}_i \bar{z}_i,
\tag{8}
$$

where $i, j$ label vector components. We can understand $\hat{\bar{Z}}_l$ to be the projection of vector $\bar{z}$ onto the unit-vector $\hat{\bar{l}}$. Since $\hat{\bar{Z}}_l$ is a normalised linear combination of $\bar{z}_i$, each of which is independent and normally distributed, **so $\hat{\bar{Z}}_l$ must be a normally distributed random variable**. This means that $q_{c_{\text{true}}}$ is distributed like a $\chi^2$ with one degree of freedom. Thus we recover the result of Wilks' theorem for the linear case.

## 3.2 Quadratic case

For the quadratic case with $|l| = 0$ and $|n| > 0$, we find

$$
\begin{aligned}
\chi^2(\bar{z}; c) &= \left| \bar{z} - (c^2 - c_{\text{true}}^2) \bar{n} \right|^2, \\
q_{c_{\text{true}}} &= \bar{z}^2 - \min_{c'^2} \left[ \left| \bar{z} - (c'^2 - c_{\text{true}}^2) \bar{n} \right|^2 \right] \\
&= \begin{cases} \hat{\bar{Z}}_n^2 & \text{if } \hat{\bar{Z}}_n \geq -\bar{N} c_{\text{true}}^2 \\ -\bar{N}^2 c_{\text{true}}^4 - 2\bar{N} c_{\text{true}}^2 \hat{\bar{Z}}_n & \text{otherwise} \end{cases},
\end{aligned}
\tag{9}
$$

where

$$
\bar{N}^2 = \sum_i \bar{n}_i^2, \qquad \hat{\bar{n}}_i = \frac{\bar{n}_i}{\sqrt{\bar{N}^2}}, \qquad \bar{Z}_n = \sum_i \bar{n}_i \bar{z}_i, \qquad \hat{\bar{Z}}_n = \sum_i \hat{\bar{n}}_i \bar{z}_i.
\tag{10}
$$

Once again we have minimised the term in square brackets by setting the derivative equal to zero and selecting the roots which correspond to global $\chi^2$ minima. We see that there are two possible cases. This can be understood by considering how the shape of the $\chi^2$-profile

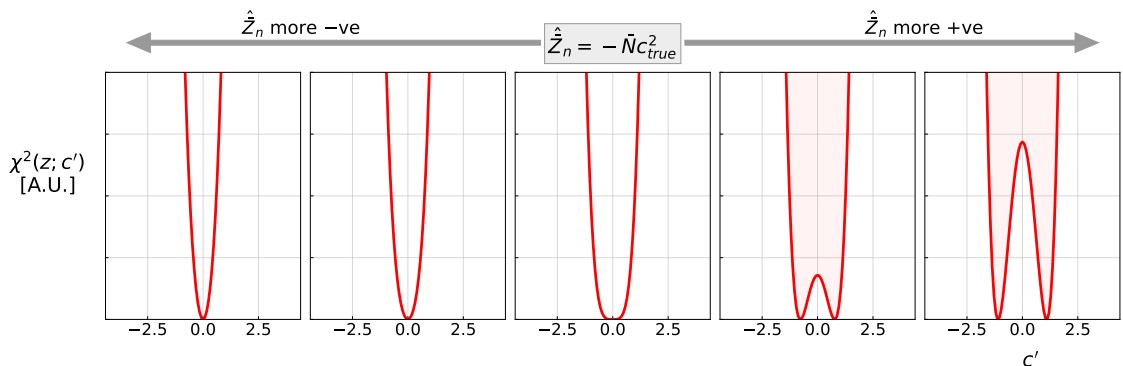

Figure 2: Illustration showing how the $\chi^2$-profile for a purely quadratic EFT model depends on $\hat{\bar{Z}}_n$. When $\hat{\bar{Z}}_n > -\bar{N}c_{\text{true}}^2$, the profile has two equal $\chi^2$-minima. Below this threshold it has only one, always at $c' = 0$. This causes the conditional statement in Eq 9.

depends on $\hat{\bar{Z}}_n$, as illustrated by the different panels in Fig 2. When $\hat{\bar{Z}}_n > -\bar{N}c_{\text{true}}^2$ the profile has two equal $\chi^2$-minima at $c'^2 = c_{\text{true}}^2 + \bar{Z}_n/\bar{N}^2$. Otherwise there is only one minimum at $c' = 0$. Using the result of Eq 9 we identify the following three cases:

$\hat{\bar{Z}}_n \geq \bar{N}c_{\text{true}}^2$,      $q_{c_{\text{true}}} \geq \bar{N}^2 c_{\text{true}}^4$ and $q_{c_{\text{true}}}$ follows a $\chi^2$-distribution with one degree of freedom.

$|\hat{\bar{Z}}_n| < \bar{N}c_{\text{true}}^2$,      $q_{c_{\text{true}}} < \bar{N}^2 c_{\text{true}}^4$ and $q_{c_{\text{true}}}$ follows a $\chi^2$-distribution with one degree of freedom.

$\hat{\bar{Z}}_n \leq -\bar{N}c_{\text{true}}^2$,      $q_{c_{\text{true}}} \geq \bar{N}^2 c_{\text{true}}^4$ and $q_{c_{\text{true}}}$ follows a Gaussian distribution with mean $-\bar{N}^2 c_{\text{true}}^4$ and variance $4\bar{N}^2 c_{\text{true}}^4$.

Since $\hat{\bar{Z}}_n$ is symmetrically distributed around 0, the first and last cases must occur with equal frequency, and we can write the final distribution of $q_{c_{\text{true}}}$ as

$$p_q\left(q_{c_{\text{true}}}\right) = \begin{cases} p_{\chi^2}^{(1)}\left(q_{c_{\text{true}}}\right) & q_{c_{\text{true}}} < \bar{N}^2 c_{\text{true}}^4 \\ \frac{1}{2}p_{\chi^2}^{(1)}\left(q_{c_{\text{true}}}\right) + \mathcal{G}\left(q_{c_{\text{true}}}; -\bar{N}^2 c_{\text{true}}^4, 2\bar{N}c_{\text{true}}^2\right) & q_{c_{\text{true}}} \geq \bar{N}^2 c_{\text{true}}^4 \end{cases} . \tag{11}$$

The quadratic case therefore deviates from Wilks' theorem by the introduction of a normally distributed term which turns on when $q_{c_{\text{true}}} \geq \bar{N}^2 c_{\text{true}}^4$.

### 3.2.1 Derivation using an Asimov approach

We can use an alternative method to derive the expected distribution of $q_{c_{\text{true}}}$ in the quadratic case. In Fig 1 (bottom middle) we see that $c_{\text{MLE}}^2$ would be normally distributed if it were allowed to take negative values. Let us consider our observation to be represented by a random variable $c^2$ which is distributed according to

$$p_{c^2}\left(c^2|c_{\text{true}}^2\right) = \mathcal{G}\left(c^2; c_{\text{true}}^2, \sigma_{c^2}\right), \tag{12}$$

where $\sigma_{c^2}$ describes the expected width of the distribution. We can write the observation as $c^2 = c_{\text{true}}^2 + \sigma_{c^2} z_c$ where $z_c$ is a normally distributed random variable. Then

$$
\begin{aligned}
q_{c_{\text{true}}} &= -2\ln\left[\frac{p_{c^2}\left(c^2|c_{\text{true}}^2\right)}{\max_{c'} p_{c^2}\left(c^2|c'^2\right)}\right] \\
&= \frac{\left(c^2 - c_{\text{true}}^2\right)^2}{\sigma_{c^2}^2} - \min_{c'^2}\left[\frac{\left(c^2 - c'^2\right)^2}{\sigma_{c^2}^2}\right] \\
&= \begin{cases} z_c^2 & \text{when } z_c \geq -\frac{c_{\text{true}}^2}{\sigma_{c^2}} \\ -\frac{c_{\text{true}}^4}{\sigma_{c^2}^2} - \frac{2c_{\text{true}}^2}{\sigma_{c^2}} z_c & \text{otherwise} \end{cases},
\end{aligned}
$$

(13)

where the minimisation is achieved by setting $c'^2 = c^2$ when $c^2 \geq 0$ and $c'^2 = 0$ otherwise. Once again we identify the three cases

$$
z_c \geq \frac{c_{\text{true}}^2}{\sigma_{c^2}} \qquad q_{c_{\text{true}}} \geq \frac{c_{\text{true}}^4}{\sigma_{c^2}^2} \text{ and } p_q\left(q_{c_{\text{true}}}\right) \propto p_{\chi^2}^{(1)}\left(q_{c_{\text{true}}}\right),
$$

$$
|z_c| < \frac{c_{\text{true}}^2}{\sigma_{c^2}} \qquad q_{c_{\text{true}}} < \frac{c_{\text{true}}^4}{\sigma_{c^2}^2} \text{ and } p_q\left(q_{c_{\text{true}}}\right) \propto p_{\chi^2}^{(1)}\left(q_{c_{\text{true}}}\right),
$$

$$
z_c \leq -\frac{c_{\text{true}}^2}{\sigma_{c^2}} \qquad q_{c_{\text{true}}} \geq \frac{c_{\text{true}}^4}{\sigma_{c^2}^2} \text{ and } p_q\left(q_{c_{\text{true}}}\right) \propto \mathcal{G}\left(q_{c_{\text{true}}}; -\frac{c_{\text{true}}^4}{\sigma_{c^2}^2}, \frac{2c_{\text{true}}^2}{\sigma_{c^2}}\right),
$$

leading to the combined distribution

$$
p_q\left(q_{c_{\text{true}}}\right) = \begin{cases} p_{\chi^2}^{(1)}\left(q_{c_{\text{true}}}\right) & q_{c_{\text{true}}} < \frac{c_{\text{true}}^4}{\sigma_{c^2}^2} \\ \frac{1}{2}p_{\chi^2}^{(1)}\left(q_{c_{\text{true}}}\right) + \mathcal{G}\left(q_{c_{\text{true}}}; -\frac{c_{\text{true}}^4}{\sigma_{c^2}^2}, \frac{2c_{\text{true}}^2}{\sigma_{c^2}}\right) & q_{c_{\text{true}}} \geq \frac{c_{\text{true}}^4}{\sigma_{c^2}^2} \end{cases}.
$$

(14)

Thus we recover the result of Eq 11 where $\bar{N}^2 = \sigma_{c^2}^{-2}$. The value of $\sigma_{c^2}$ may be reliably estimated by profiling the likelihood contour $\mathcal{L}_{\text{asimov}}$ of an Asimov dataset [12] and searching for the location where

$$
\log \mathcal{L}_{\text{asimov}}\left(c^2 = c_{\text{true}}^2 + \sigma_{c^2}\right) = \log \mathcal{L}_{\text{asimov}}\left(c^2 = c_{\text{true}}^2\right) - \frac{1}{2}.
$$

(15)

Three notable features of the Asimov approach are that (i) it can be applied without calculating $\bar{N}$, making computation easier, (ii) it may be used to cross-check the result obtained by applying Eq 11, and (iii) we speculate that it may be applied (with caution) in less idealised circumstances, such as those with non-Gaussian nuisance parameters, to obtain an approximate form of the asymptotic distribution.

### 3.2.2 An example

We now use a toy example to test the results of Eqs 11 and 14. We consider a single Wilson coefficient $c$ with a purely quadratic dependence and

$$
\bar{n} = \{0.0,\ 0.0,\ 0.1,\ 0.2,\ 0.3,\ 0.5,\ 0.8\}.
$$

(16)

Fig 3 (top row) shows the probability density function (PDF) of $q_{c_{\text{true}}}$ for increasing values of $c_{\text{true}}^2$. Pseudo-experiments are shown as the shaded histogram and labelled Toys. Overlayed lines show the results of our two methods as well as the $\chi^2$-distribution if Wilks' theorem

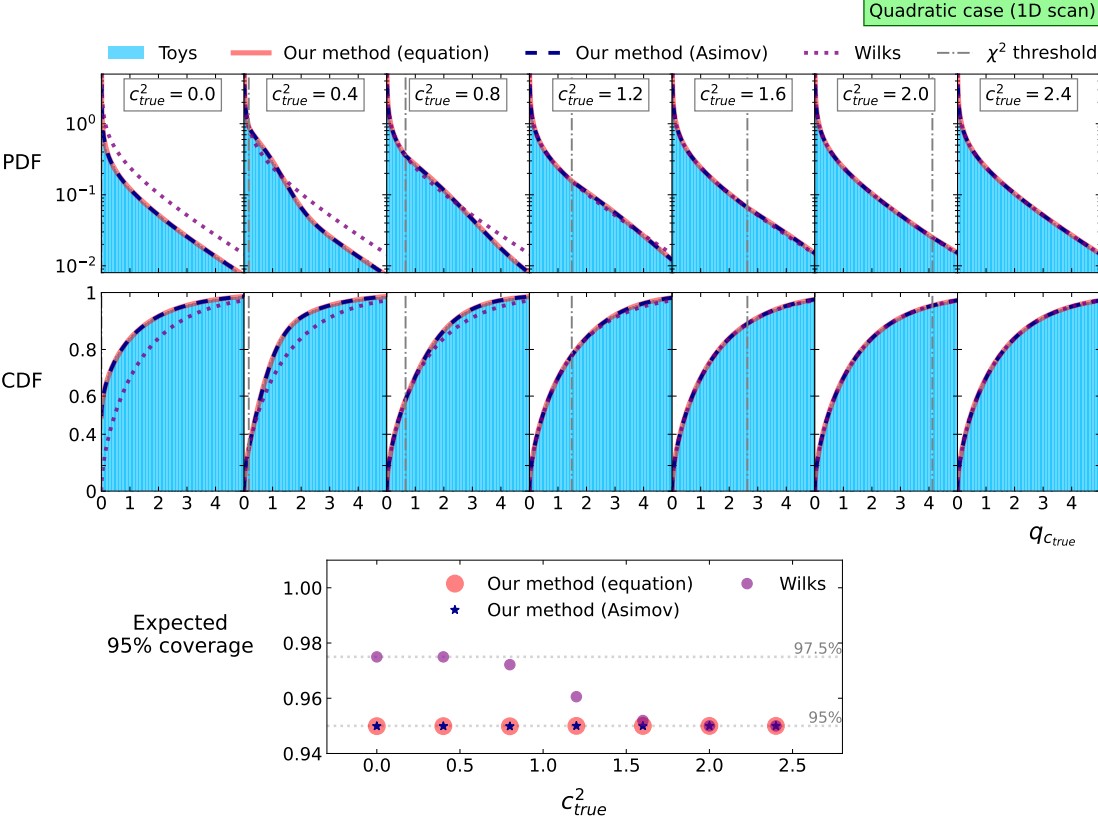

Figure 3: Demonstration of the validity of our method when considering a single Wilson coefficient which modulates the purely quadratic contributions described by Eq 16.

were assumed. A vertical dotted line represents the threshold above which Wilks' theorem is violated. The second row shows the corresponding cumulative distribution functions (CDFs), which are used to derive $p$-values.

When $c^2_{\text{true}} = 0$, the PDF is equal to $\frac{1}{2}\delta\left(q_{c_{\text{true}}}\right) + \frac{1}{2}p^{(1)}_{\chi^2}\left(q_{c_{\text{true}}}\right)$. As $c^2_{\text{true}}$ increases, the $\chi^2$-validity threshold moves upwards from 0. When $c^2_{\text{true}} \gg 0$, the threshold is high enough that Wilks' theorem is able to describe the distribution over all relevant values of $q_{c_{\text{true}}}$. This is because Wilks' theorem is only valid when it is unlikely that parameter profiling will encounter the boundary at $c = 0$. This condition is only satisfied when the true hypothesis is far from the Standard Model. By contrast, our method agrees well with pseudo-experiments for all values of $c^2_{\text{true}}$. Furthermore we observe that the two implementations of our method are essentially identical.

The bottom panel of Fig 3 shows the coverage one obtains when deriving 95% confidence limits when assuming Wilks' theorem compared with our method. When $c^2_{\text{true}} \sim 0$, the Wilks case fails to exclude $c^2_{\text{true}}$ in half of all cases expected. This is because 50% of all extreme fluctuations must fall in the negative-side tail of the random variable $c^2$, which Wilks assumes to result in large $q_{c_{\text{true}}}$ but in reality are collapsed onto $q_{c_{\text{true}}} \sim 0$. The exclusion rate is therefore 2.5%, i.e. exactly half of the expected 5%, and the coverage is 97.5% for hypotheses close to the Standard Model. As $c^2_{\text{true}}$ increases to approximately $2\sigma_{c^2}$, at which point the expected 95% confidence interval is no longer expected to include $c^2_{\text{true}} = 0$, we observe that the Wilks case converges to the target 95% coverage. By contrast, our method provides the correct coverage for all values of $c^2_{\text{true}}$. Note that $\sigma_{c^2} = 0.99$ in this example.

# 4 Solution with two (linear or quadratic) parameters

We now extend our arguments to discuss fits with two Wilson coefficients $c_1$ and $c_2$, each of which may provide *either* linear *or* quadratic EFT contributions. We do not consider cases with significant interference between the different EFT contributions, i.e. we assume $t_{1,2} \approx 0$. The behaviour of the PLR varies depending on whether the individual EFT contributions are linear or quadratic. We will now consider each case in turn.

## 4.1 Linear x linear case

Once again we work in our de-correlated and normalised basis in which observations are a vector of independent normal random variables $\bar{z}$. Say that our Wilson coefficients $c = (c_1, \ c_2)$ provide linear contributions of $\bar{l}_1$ and $\bar{l}_2$ and are hypothesised to have true values of $c_{\text{true}} = (c_{\text{t},1}, \ c_{\text{t},2})$ respectively. Following the same approach as the one-dimensional solutions, we define the quantities

$$
\begin{aligned}
\bar{L}_1^2 = \bar{l}_1 \cdot \bar{l}_1 , \qquad \hat{\bar{l}}_1 = \frac{\bar{l}_1}{\sqrt{\bar{L}_1^2}} , \qquad \bar{Z}_1 = \bar{l}_1 \cdot \bar{z} , \qquad \hat{\bar{Z}}_1 = \hat{\bar{l}}_1 \cdot \bar{z} , \\
\bar{L}_2^2 = \bar{l}_2 \cdot \bar{l}_2 , \qquad \hat{\bar{l}}_2 = \frac{\bar{l}_2}{\sqrt{\bar{L}_2^2}} , \qquad \bar{Z}_2 = \bar{l}_2 \cdot \bar{z} , \qquad \hat{\bar{Z}}_2 = \hat{\bar{l}}_2 \cdot \bar{z} , \\
\bar{M}^2 = \bar{l}_1 \cdot \bar{l}_2 , \qquad \rho = \hat{\bar{l}}_1 \cdot \hat{\bar{l}}_2 .
\end{aligned}
\tag{17}
$$

Compared with earlier sections, we have introduced the additional quantities $\bar{M}^2$ and $\rho$, which respectively represent the un-normalised and normalised inner products between $\bar{l}_1$ and $\bar{l}_2$. The $\chi^2$ is then

$$
\begin{aligned}
\chi^2(\bar{z}; c) &= \left| \bar{z} - (c_1 - c_{\text{t},1}) \bar{l}_1 - (c_2 - c_{\text{t},2}) \bar{l}_2 \right|^2 \\
&= |\bar{z}|^2 + (c_1 - c_{\text{t},1})^2 \bar{L}_1^2 + (c_2 - c_{\text{t},2})^2 \bar{L}_2^2 + 2(c_1 - c_{\text{t},1})(c_2 - c_{\text{t},2}) \bar{M}^2 \\
&\quad - 2(c_1 - c_{\text{t},1}) \bar{L}_1 \hat{\bar{Z}}_1 - 2(c_2 - c_{\text{t},2}) \bar{L}_2 \hat{\bar{Z}}_2 .
\end{aligned}
\tag{18}
$$

Once again we notice that $\hat{\bar{Z}}_1$ and $\hat{\bar{Z}}_2$ are normally distributed random variables. However, if $\hat{\bar{l}}_1$ and $\hat{\bar{l}}_2$ are not orthogonal then $\hat{\bar{Z}}_1$ and $\hat{\bar{Z}}_2$ must be correlated. Fortunately we can interpret $\rho$ as exactly this correlation coefficient. Defining $\hat{\bar{l}}_\perp$ as a unit-vector perpendicular to $\hat{\bar{l}}_1$, then $\bar{Z}_\perp = \bar{z} \cdot \hat{\bar{l}}_\perp$ and we can write

$$
\hat{\bar{Z}}_2 = \rho \hat{\bar{Z}}_1 + \sqrt{1 - \rho^2} \hat{\bar{Z}}_\perp .
\tag{19}
$$

The $\chi^2$ function then becomes

$$
\begin{aligned}
\chi^2(\bar{z}; c) &= |\bar{z}|^2 + (c_1 - c_{\text{t},1})^2 \bar{L}_1^2 + (c_2 - c_{\text{t},2})^2 \bar{L}_2^2 + 2(c_1 - c_{\text{t},1})(c_2 - c_{\text{t},2}) \bar{M}^2 \\
&\quad - 2(c_1 - c_{\text{t},1}) \bar{L}_1 \hat{\bar{Z}}_1 - 2(c_2 - c_{\text{t},2}) \rho \bar{L}_2 \hat{\bar{Z}}_1 - 2(c_2 - c_{\text{t},2}) \sqrt{1 - \rho^2} \bar{L}_2 \hat{\bar{Z}}_\perp ,
\end{aligned}
\tag{20}
$$

where $\hat{\bar{Z}}_1$ and $\hat{\bar{Z}}_\perp$ are *independent* normally distributed random variables. Once again we find the maximum likelihood estimates $c_{\text{MLE}} = (c_{\text{MLE},1}, \ c_{\text{MLE},2})$ by finding the global minima of the $\chi^2$ function. Where $\nabla_{c_i}$ represents a gradient with respect to $c_i$, we have

$$
\begin{aligned}
\nabla_{c_1} \chi^2(\bar{z}; c) &= 2 \left[ (c_1 - c_{\text{t},1}) \bar{L}_1^2 + (c_2 - c_{\text{t},2}) \bar{M}^2 - \bar{L}_1 \hat{\bar{Z}}_1 \right] , \\
\nabla_{c_2} \chi^2(\bar{z}; c) &= 2 \left[ (c_2 - c_{\text{t},2}) \bar{L}_2^2 + (c_1 - c_{\text{t},1}) \bar{M}^2 - \rho \bar{L}_2 \hat{\bar{Z}}_1 - \sqrt{1 - \rho^2} \bar{L}_2 \hat{\bar{Z}}_\perp \right] ,
\end{aligned}
\tag{21}
$$

and so

$$\left(c_{\mathrm{MLE},1}-c_{\mathrm{t},1}\right)\bar{L}_1^2 \ + \ \left(c_{\mathrm{MLE},2}-c_{\mathrm{t},2}\right)\bar{M}^2 \ - \ \bar{L}_1\hat{\bar{Z}}_1 \ = \ 0\,, \tag{22}$$
$$\left(c_{\mathrm{MLE},2}-c_{\mathrm{t},2}\right)\bar{L}_2^2 \ + \ \left(c_{\mathrm{MLE},1}-c_{\mathrm{t},1}\right)\bar{M}^2 \ - \ \rho\bar{L}_2\hat{\bar{Z}}_1 \ - \ \sqrt{1-\rho^2}\bar{L}_2\hat{\bar{Z}}_\perp \ = \ 0\,.$$

Noticing that $\bar{M} = \rho\bar{L}_1\bar{L}_2$ and solving these simultaneous equations we find

$$c_{\mathrm{MLE},1} \ = \ c_{\mathrm{t},1} \ + \ \frac{1}{\bar{L}_1}\left(\hat{\bar{Z}}_1 \ - \ \frac{\rho}{\sqrt{1-\rho^2}}\,\hat{\bar{Z}}_\perp\right)\,, \tag{23}$$
$$c_{\mathrm{MLE},2} \ = \ c_{\mathrm{t},2} \ + \ \frac{1}{\bar{L}_2}\,\frac{1}{\sqrt{1-\rho^2}}\,\hat{\bar{Z}}_\perp\,.$$

Plugging these solutions back into Eq 20 we find

$$\chi^2(\bar{z};c_{\mathrm{MLE}}) \ = \ |\bar{z}|^2 \ - \ \hat{\bar{Z}}_1^2 \ - \ \hat{\bar{Z}}_\perp^2\,, \tag{24}$$

and so

$$\begin{aligned}
q_{c_{\mathrm{true}}} \ &= \ \chi^2(\bar{z};c_{\mathrm{true}}) \ - \ \chi^2(\bar{z};c_{\mathrm{MLE}}) \\
&= \ |\bar{z}|^2 \ - \ \left(|\bar{z}|^2 \ - \ \hat{\bar{Z}}_1^2 \ - \ \hat{\bar{Z}}_\perp^2\right) \\
&= \ \hat{\bar{Z}}_1^2 \ + \ \hat{\bar{Z}}_\perp^2\,.
\end{aligned} \tag{25}$$

Since $\hat{\bar{Z}}_1$ and $\hat{\bar{Z}}_\perp$ are independent normally distributed random variables, we infer that $q_{c_{\mathrm{true}}}$ follows a $\chi^2$-distribution with two degrees of freedom. Thus we recover the result of Wilks' theorem.

## 4.2 Linear x quadratic case

Say that Wilson coefficient $c_1$ still modulates a linear contribution $\bar{l}_1$, but $c_2$ now modulates a quadratic contribution $\bar{n}_2$. Then

$$\bar{N}_2^2 = \bar{n}_2\cdot\bar{n}_2\,, \qquad \hat{\bar{n}}_2 = \frac{\bar{n}_2}{\sqrt{\bar{N}_2^2}}\,, \qquad \bar{Z}_2 = \bar{n}_2\cdot\bar{z}\,, \qquad \hat{\bar{Z}}_2 = \hat{\bar{n}}_2\cdot\bar{z}\,, \tag{26}$$
$$\bar{M}^2 = \bar{l}_1\cdot\bar{n}_2 = \rho\bar{L}_1\bar{N}_2\,, \qquad \rho = \hat{\bar{l}}_1\cdot\hat{\bar{n}}_2\,.$$

Once again we note that $\hat{\bar{Z}}_1$ and $\hat{\bar{Z}}_2$ have correlation coefficient $\rho$, now defined as the inner product between $\hat{\bar{l}}_1$ and $\hat{\bar{n}}_2$, and we transform into an uncorrelated basis according to Eq 19. The $\chi^2$ function is now

$$\begin{aligned}
\chi^2(\bar{z};c) \ &= \ \left|\bar{z} \ - \ (c_1-c_{\mathrm{t},1})\bar{l}_1 \ - \ \left(c_2^2-c_{\mathrm{t},2}^2\right)\bar{n}_2\right|^2 \\
&= \ |\bar{z}|^2 \ + \ \left(c_1-c_{\mathrm{t},1}\right)^2\bar{L}_1^2 \ + \ \left(c_2^2-c_{\mathrm{t},2}^2\right)^2\bar{N}_2^2 \ + \ 2\left(c_1-c_{\mathrm{t},1}\right)\left(c_2^2-c_{\mathrm{t},2}^2\right)\bar{M}^2 \\
&\quad - \ 2\left(c_1-c_{\mathrm{t},1}\right)\bar{L}_1\hat{\bar{Z}}_1 \ - \ 2\left(c_2^2-c_{\mathrm{t},2}^2\right)\rho\bar{N}_2\hat{\bar{Z}}_1 \ - \ 2\left(c_2^2-c_{\mathrm{t},2}^2\right)\sqrt{1-\rho^2}\bar{N}_2\hat{\bar{Z}}_\perp\,.
\end{aligned} \tag{27}$$

Taking the gradients we find

$$\nabla_{c_1}\chi^2(\bar{z};c) \ = \ 2\left[\left(c_1-c_{\mathrm{t},1}\right)\bar{L}_1^2 \ + \ \left(c_2^2-c_{\mathrm{t},2}^2\right)\bar{M}^2 \ - \ \bar{L}_1\hat{\bar{Z}}_1\right]\,, \tag{28}$$
$$\nabla_{c_2}\chi^2(\bar{z};c) \ = \ 4c_2\left[\left(c_2^2-c_{\mathrm{t},2}^2\right)\bar{N}_2^2 \ + \ \left(c_1-c_{\mathrm{t},1}\right)\bar{M}^2 \ - \ \rho\bar{N}_2\hat{\bar{Z}}_1 \ - \ \sqrt{1-\rho^2}\bar{N}_2\hat{\bar{Z}}_\perp\right]\,.$$

Setting these equal to zero to obtain the stationary points, we see that there are two different regimes for $c_2$: either Eq 28 has three real roots, corresponding to two equal-minima and one maximum, or it has one real root corresponding to a single minimum. We label these classes "A" and "B", and will now consider them in turn.

**Class A:** In this category, the root at $c_2 = 0$ is a local maximum and we obtain the two global $\chi^2$-minima when

$$
\begin{aligned}
\left(c_{\text{MLE},1} - c_{\text{t},1}\right)\bar{L}_1^2 + \left(c_{\text{MLE},2}^2 - c_{\text{t},2}^2\right)\bar{M}^2 - \bar{L}_1\hat{\bar{Z}}_1 &= 0, \\
\left(c_{\text{MLE},2}^2 - c_{\text{t},2}^2\right)\bar{N}_2^2 + \left(c_{\text{MLE},1} - c_{\text{t},1}\right)\bar{M}^2 - \rho\bar{N}_2\hat{\bar{Z}}_1 - \sqrt{1-\rho^2}\bar{N}_2\hat{\bar{Z}}_\perp &= 0.
\end{aligned}
\tag{29}
$$

These events are the same as in the linear case, except that we make the substitutions $\left(c_2,\ c_{\text{t},2},\ \bar{l}_2\right) \to \left(c_2^2,\ c_{\text{t},2}^2,\ \bar{n}_2\right)$. The solutions are therefore

$$
\begin{aligned}
c_{\text{MLE},1} &= c_{\text{t},1} + \frac{1}{\bar{L}_1}\left(\hat{\bar{Z}}_1 - \frac{\rho}{\sqrt{1-\rho^2}}\,\hat{\bar{Z}}_\perp\right), \\
c_{\text{MLE},2}^2 &= c_{\text{t},2}^2 + \frac{1}{\bar{N}_2}\frac{1}{\sqrt{1-\rho^2}}\,\hat{\bar{Z}}_\perp.
\end{aligned}
\tag{30}
$$

This result occurs whenever Eq 28 has three real roots, i.e. when

$$
c_{\text{t},2}^2 + \frac{1}{\bar{N}_2}\rho\hat{\bar{Z}}_1 + \frac{1}{\bar{N}_2}\sqrt{1-\rho^2}\hat{\bar{Z}}_\perp - \frac{\bar{M}^2}{\bar{N}_2^2}\left(c_{\text{MLE},1} - c_{\text{t},1}\right) \geq 0.
\tag{31}
$$

Plugging in our solution for $c_{\text{MLE},1}$ and re-arranging gives

$$
\hat{\bar{Z}}_\perp \geq -\hat{\bar{Z}}_{\perp,0},
\tag{32}
$$

where $\hat{\bar{Z}}_{\perp,0} = \bar{N}_2\sqrt{1-\rho^2}\,c_{\text{t},2}^2$. Eq 32 determines which events are designated class A. Plugging our maximum likelihood estimates into Eq 27 then propagating onto $q_{c_{\text{true}}}$ gives

$$
q_{c_{\text{true}}} = \hat{\bar{Z}}_1^2 + \hat{\bar{Z}}_\perp^2,
\tag{33}
$$

and so $q_{c_{\text{true}}}$ follows a $\chi^2$-distribution with two degrees of freedom for class A events.

**Class B:** In this category, the root at $c_2 = 0$ is a global minimum and we find the maximum likelihood estimates

$$
c_{\text{MLE},1} = c_{\text{t},1} + \frac{1}{\bar{L}_1}\hat{\bar{Z}}_1 + \frac{\bar{N}_2}{\bar{L}_1}\rho\,c_{\text{t},2}^2, \qquad c_{\text{MLE},2} = 0,
\tag{34}
$$

which lead to

$$
q_{c_{\text{true}}} = \hat{\bar{Z}}_1^2 - 2\,\hat{\bar{Z}}_{\perp,0}\,\hat{\bar{Z}}_\perp - \hat{\bar{Z}}_{\perp,0}^2,
\tag{35}
$$

and so $q_{c_{\text{true}}}$ does not follow a $\chi^2$-distribution for class B events, which occur when Eq 32 is not satisfied.

We can now use the usual change-of-variables method to calculate the density $p_q\left(q_{c_{\text{true}}}\right)$ from the known densities $\mathcal{N}(\hat{\bar{Z}}_1)$ and $\mathcal{N}(\hat{\bar{Z}}_\perp)$. Since $\{\hat{\bar{Z}}_1, \hat{\bar{Z}}_\perp\} \to q_{c_{\text{true}}}$ is a $2 \to 1$ transformation, we must integrate over the spare degree of freedom. For a given $\hat{\bar{Z}}_\perp$, Eq 32 tells us whether to evaluate the integrand using the class A or B solutions. To simplify the transition

between class A and B regions, we define $\hat{\tilde{Z}}_{\perp}$ as the integration parameter. We then obtain $\hat{\tilde{Z}}_1$ according to

$$\hat{\tilde{Z}}_1^{(A)} = \pm\sqrt{q_{c_{\text{true}}} - \hat{\tilde{Z}}_{\perp}^2}, \qquad \hat{\tilde{Z}}_1^{(B)} = \pm\sqrt{q_{c_{\text{true}}} + \hat{\tilde{Z}}_{\perp,0}^2 + 2\hat{\tilde{Z}}_{\perp,0}\hat{\tilde{Z}}_{\perp}}. \tag{36}$$

Since all $\hat{\tilde{Z}}_1$ must be real, our integral limits must also satisfy $-\sqrt{q_{c_{\text{true}}}} \leq \hat{\tilde{Z}}_{\perp} \leq \sqrt{q_{c_{\text{true}}}}$ for class A and $\hat{\tilde{Z}}_{\perp} \geq -\frac{1}{2}(\frac{q_{c_{\text{true}}}}{\hat{\tilde{Z}}_{\perp,0}} + \hat{\tilde{Z}}_{\perp,0})$ for class B. This leads to the solution

$$p_q\left(q_{c_{\text{true}}}\right) = \tilde{p}_q^{(A)}\left(q_{c_{\text{true}}}\right) + \tilde{p}_q^{(B)}\left(q_{c_{\text{true}}}\right), \tag{37}$$

where

$$\begin{aligned}
\tilde{p}_q^{(A)}\left(q_{c_{\text{true}}}\right) &= \int_{\max\left(-\sqrt{q_{c_{\text{true}}}},-\hat{\tilde{Z}}_{\perp,0}\right)}^{\sqrt{q_{c_{\text{true}}}}} \sum_{\hat{\tilde{Z}}_1^{(A)}} \left|\frac{\mathrm{d}\hat{\tilde{Z}}_1^{(A)}}{\mathrm{d}q_{c_{\text{true}}}}\right| \mathcal{N}\left(\hat{\tilde{Z}}_1^{(A)}\right) \mathcal{N}\left(\hat{\tilde{Z}}_{\perp}\right) \mathrm{d}\hat{\tilde{Z}}_{\perp}, \\
\tilde{p}_q^{(B)}\left(q_{c_{\text{true}}}\right) &= \int_{-\frac{1}{2}\left(\frac{q_{c_{\text{true}}}}{\hat{\tilde{Z}}_{\perp,0}} + \hat{\tilde{Z}}_{\perp,0}\right)}^{-\hat{\tilde{Z}}_{\perp,0}} \sum_{\hat{\tilde{Z}}_1^{(B)}} \left|\frac{\mathrm{d}\hat{\tilde{Z}}_1^{(B)}}{\mathrm{d}q_{c_{\text{true}}}}\right| \mathcal{N}\left(\hat{\tilde{Z}}_1^{(B)}\right) \mathcal{N}\left(\hat{\tilde{Z}}_{\perp}\right) \mathrm{d}\hat{\tilde{Z}}_{\perp}.
\end{aligned} \tag{38}$$

The derivatives are the Jacobian factors which quantify how the density is squeezed or diluted by the transformation between spaces. Noticing that $\mathcal{N}(x) + \mathcal{N}(-x) = 2\mathcal{N}(x)$ and $p_{\chi^2}^{(1)}\left(x^2\right) = \left|\frac{1}{x}\right| \mathcal{N}(x)$ for some variable $x$, and introducing the $\chi^2$-distribution with two degrees of freedom according to

$$p_{\chi^2}^{(2)}(q) = \int_0^q p_{\chi^2}^{(1)}\left(q - x^2\right) p_{\chi^2}^{(1)}\left(x^2\right) \mathrm{d}\left(x^2\right), \tag{39}$$

we can re-arrange Eq 38 to obtain the ultimate solution

$$p_q\left(q_{c_{\text{true}}}\right) = \begin{cases} \frac{1}{2}p_{\chi^2}^{(1)}\left(q_{c_{\text{true}}}\right) + \frac{1}{2}p_{\chi^2}^{(2)}\left(q_{c_{\text{true}}}\right) & \text{if } c_{t,2} = 0, \\ p_{\chi^2}^{(2)}\left(q_{c_{\text{true}}}\right) & \text{if } q_{c_{\text{true}}} \leq \hat{\tilde{Z}}_{\perp,0}^2, \\ \frac{1}{2}p_{\chi^2}^{(2)}\left(q_{c_{\text{true}}}\right) + \tilde{p}_1\left(q_{c_{\text{true}}}\right) + \tilde{p}_2\left(q_{c_{\text{true}}}\right) & \text{otherwise}, \end{cases} \tag{40}$$

with

$$\begin{aligned}
\tilde{p}_1\left(q_{c_{\text{true}}}\right) &= \int_0^{\hat{\tilde{Z}}_{\perp,0}} p_{\chi^2}^{(1)}\left(q_{c_{\text{true}}} - \hat{\tilde{Z}}_{\perp}^2\right) \mathcal{N}\left(\hat{\tilde{Z}}_{\perp}\right) \mathrm{d}\hat{\tilde{Z}}_{\perp}, \\
\tilde{p}_2\left(q_{c_{\text{true}}}\right) &= \int_{\hat{\tilde{Z}}_{\perp,0}}^{\frac{1}{2}\left(\frac{q_{c_{\text{true}}}}{\hat{\tilde{Z}}_{\perp,0}} + \hat{\tilde{Z}}_{\perp,0}\right)} p_{\chi^2}^{(1)}\left(q_{c_{\text{true}}} + \hat{\tilde{Z}}_{\perp,0}^2 - 2\hat{\tilde{Z}}_{\perp,0}\hat{\tilde{Z}}_{\perp}\right) \mathcal{N}\left(\hat{\tilde{Z}}_{\perp}\right) \mathrm{d}\hat{\tilde{Z}}_{\perp}.
\end{aligned} \tag{41}$$

We note that Eq 40 depends only on the constant $\hat{\tilde{Z}}_{\perp,0}$, which in turn depends only on $\bar{N}_2$ and $\rho$. Once again, these quantities can be extracted from an Asimov scan, where $\rho$ is estimated using the Hessian matrix and $\bar{N}_2 = \sqrt{\sigma_{c^2}^{-2}}$.

### 4.2.1  An example

Let us define an example with $\bar{l}_1 = (-0.5, 0.3)$ and $\bar{n}_2 = (0.8, 0.1)$ which results in a strong negative correlation of $\rho = -0.79$. Fig 4 compares our PDF with Wilks and pseudo-experiments

for a selection of $c_{\text{true}}$ values. When $c_{t,2} = 0$, the distribution contains a $p_{\chi^2}^{(1)}$ component and so has a singularity at $q_{c_{\text{true}}} = 0$. As $c_{t,2}$ moves away from 0, the boundary at which Wilks' theorem is violated increases from 0 and the distribution tends towards $p_{\chi^2}^{(2)}$. Our method agrees well with the distribution of toys. The integrals of Eq 41 are estimated numerically using Gaussian quadrature summation [21].

Fig 5 shows the expected coverage of the 95% confidence interval obtained when assuming Wilks' theorem (left panel) compared with our method (right panel), estimated using pseudo-experiments. Once again, assuming Wilks' theorem results in significant over-coverage when $c_{t,2} \sim 0$. The extent of the problem now depends on the value of $\hat{\bar{Z}}_{\perp,0}$. A maximum coverage of 97.5% would be obtained when $\hat{\bar{Z}}_{\perp,0} = 0$, tending to 95% when $\hat{\bar{Z}}_{\perp,0} \gg 0$, with this example falling in between with a maximum coverage of approximately 96.8%. By contrast, once again our method provides the correct coverage in all cases. We note that Eqs 40-41 and Figs 4-5 are independent of $c_{t,1}$. This is because we never encounter the boundary of parameter space when profiling in the direction of $c_1$.

## 4.3 Quadratic x quadratic case

Say that Wilson coefficient $c_1$ now also modulates a quadratic contribution $\bar{n}_1$. Then

$$
\bar{N}_1^2 = \bar{n}_1 \cdot \bar{n}_1, \qquad \hat{\bar{n}}_1 = \frac{\bar{n}_1}{\sqrt{\bar{N}_1^2}}, \qquad \bar{Z}_1 = \bar{n}_1 \cdot \bar{z}, \qquad \hat{\bar{Z}}_1 = \hat{\bar{n}}_1 \cdot \bar{z},
$$
$$
\bar{M}^2 = \bar{n}_1 \cdot \bar{n}_2 = \rho \bar{N}_1 \bar{N}_2, \qquad \rho = \hat{\bar{n}}_1 \cdot \hat{\bar{n}}_2.
\tag{42}
$$

Whilst all elements of $n_1$ and $n_2$ are positive definite, $\bar{n}_1$ and $\bar{n}_2$ are not, and in this basis it is possible to have negative correlation coefficients. The $\chi^2$-function is

$$
\begin{aligned}
\chi^2(\bar{z}; c) &= \left| \bar{z} - \left(c_1^2 - c_{t,1}^2\right)\bar{n}_1 - \left(c_2^2 - c_{t,2}^2\right)\bar{n}_2 \right|^2 \\
&= |\bar{z}|^2 + \left(c_1^2 - c_{t,1}^2\right)^2 \bar{N}_1^2 + \left(c_2^2 - c_{t,2}^2\right)^2 \bar{N}_2^2 + 2\left(c_1^2 - c_{t,1}^2\right)\left(c_2^2 - c_{t,2}^2\right)\bar{M}^2 \\
&\quad - 2\left(c_1 - c_{t,1}\right)\bar{N}_1 \hat{\bar{Z}}_1 - 2\left(c_2^2 - c_{t,2}^2\right)\rho\bar{N}_2\hat{\bar{Z}}_1 - 2\left(c_2^2 - c_{t,2}^2\right)\sqrt{1 - \rho^2}\bar{N}_2\hat{\bar{Z}}_\perp,
\end{aligned}
\tag{43}
$$

for which the gradients are

$$
\begin{aligned}
\nabla_{c_1}\chi^2(\bar{z}; c) &= 4c_1\left[\left(c_1^2 - c_{t,1}^2\right)\bar{N}_1^2 + \left(c_2^2 - c_{t,2}^2\right)\bar{M}^2 - \bar{N}_1\hat{\bar{Z}}_1\right], \\
\nabla_{c_2}\chi^2(\bar{z}; c) &= 4c_2\left[\left(c_2^2 - c_{t,2}^2\right)\bar{N}_2^2 + \left(c_1^2 - c_{t,1}^2\right)\bar{M}^2 - \rho\bar{N}_2\hat{\bar{Z}}_1 - \sqrt{1 - \rho^2}\bar{N}_2\hat{\bar{Z}}_\perp\right].
\end{aligned}
\tag{44}
$$

Once again we find the maximum likelihood estimates by setting the gradients equal to zero and choosing the global $\chi^2$-minimum. In Eq 44 we see that $\nabla_{c_1}\chi^2(\bar{z}; c) = 0$ and $\nabla_{c_2}\chi^2(\bar{z}; c) = 0$ each have two solutions, which combine to give four different classes of events. We will now consider these in turn.

---

**Class AA:** In this category we find four equal $\chi^2$-minima at

$$
\begin{aligned}
\left(c_{\text{MLE},1}^2 - c_{t,1}^2\right)\bar{N}_1^2 + \left(c_{\text{MLE},2}^2 - c_{t,2}^2\right)\bar{M}^2 - \bar{N}_1\hat{\bar{Z}}_1 &= 0, \\
\left(c_{\text{MLE},2}^2 - c_{t,2}^2\right)\bar{N}_2^2 + \left(c_{\text{MLE},1}^2 - c_{t,1}^2\right)\bar{M}^2 - \rho\bar{N}_2\hat{\bar{Z}}_1 - \sqrt{1 - \rho^2}\bar{N}_2\hat{\bar{Z}}_\perp &= 0,
\end{aligned}
\tag{45}
$$

which solve to give

$$
c_{\text{MLE},1}^2 = c_{t,1}^2 + \frac{1}{\bar{N}_1}\hat{\bar{Z}}_1 - \frac{\rho}{\bar{N}_1\sqrt{1 - \rho^2}}\hat{\bar{Z}}_\perp, \qquad c_{\text{MLE},2}^2 = c_{t,2}^2 + \frac{\rho}{\bar{N}_2\sqrt{1 - \rho^2}}\hat{\bar{Z}}_\perp.
\tag{46}
$$

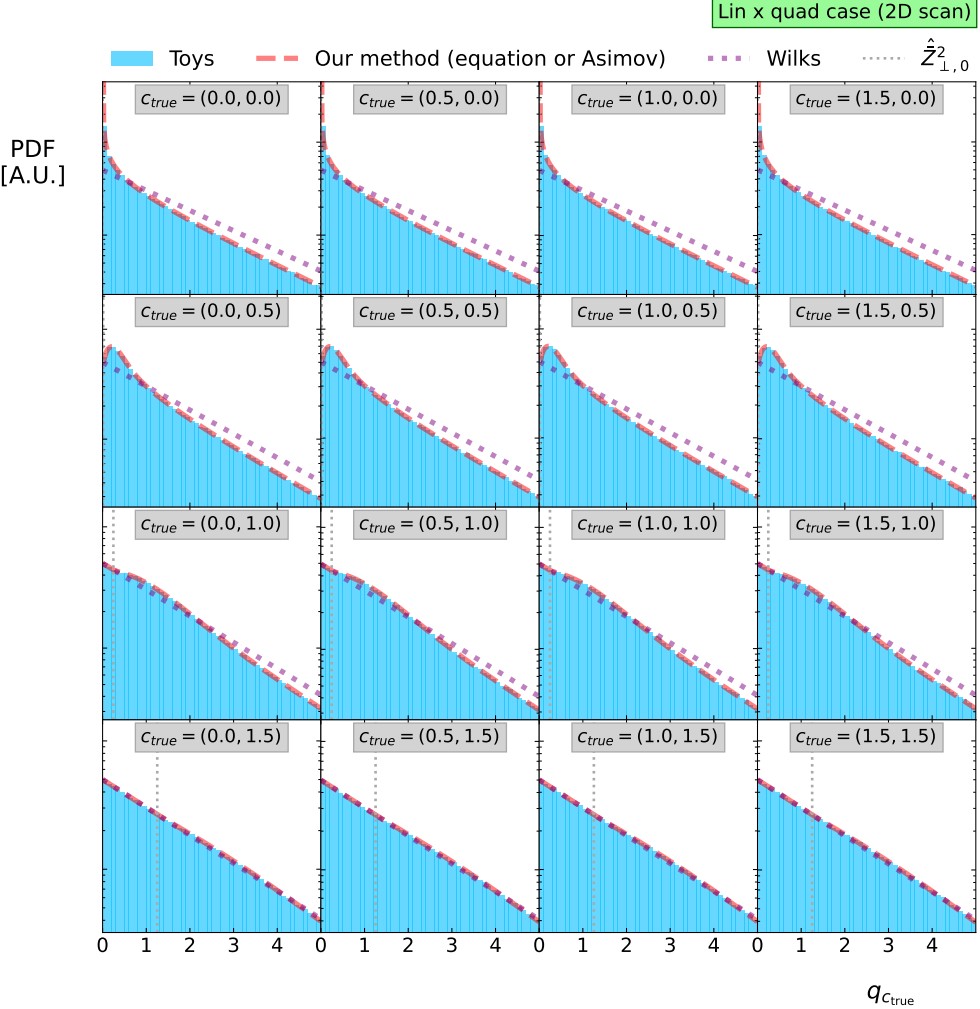

Figure 4: Density of pseudo-experiments compared with our method for a 2D example with $\bar{l}_1 = (-0.5, 0.3)$, $\bar{n}_2 = (0.8, 0.1)$ and a selection of $c_{\text{true}}$. Our method is able to describe cases with $c_{\text{t},2}^2 \sim 0$ where Wilks' theorem is violated.

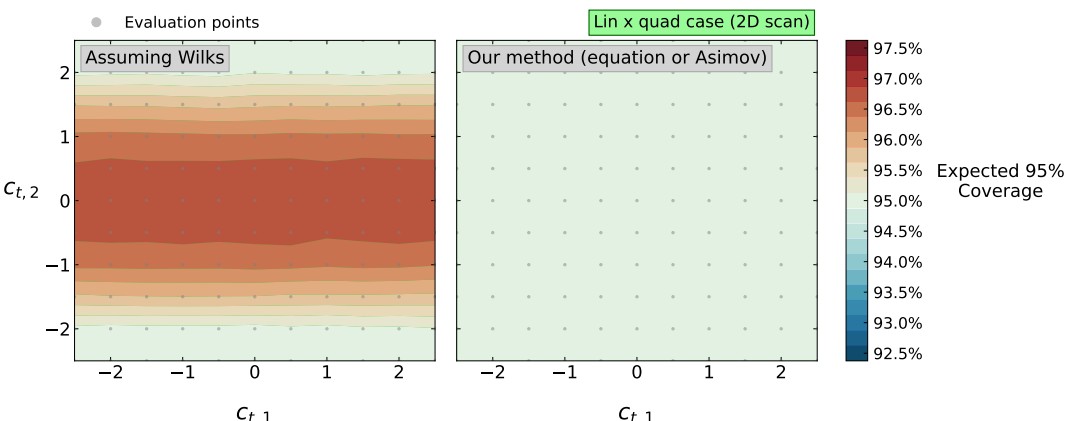

Figure 5: Expected 95% coverage assuming Wilks (left) and using our method (right) for a 2D example with $\bar{l}_1 = (-0.5, 0.3)$, $\bar{n}_2 = (0.8, 0.1)$ and a selection of $c_{\text{true}}$. Our method provides correct coverage for $|c_{\text{t},2}| \lesssim 2\sigma_{c_2}$ where Wilks' theorem is violated.

An event falls into this category when $\nabla_{c_1}\chi^2(\bar{z};c)$ and $\nabla_{c_2}\chi^2(\bar{z};c)$ both have three real roots. This leads to the class conditions

$$\hat{\hat{Z}}_1 \geq \frac{\rho}{\sqrt{1-\rho^2}}\hat{\hat{Z}}_\perp - \kappa_1, \qquad \hat{\hat{Z}}_\perp \geq -\sqrt{1-\rho^2}\,\kappa_2, \tag{47}$$

where we define the convenient constants $\kappa_1 = \bar{N}_1\,c_{t,1}^2$ and $\kappa_2 = \bar{N}_2\,c_{t,2}^2$. Plugging the maximum likelihood estimates back into the $\chi^2$-function results in

$$q_{c_{\text{true}}} = \hat{\hat{Z}}_1^2 + \hat{\hat{Z}}_\perp^2, \qquad \hat{\hat{Z}}_1 = \pm\sqrt{q_{c_{\text{true}}} - \hat{\hat{Z}}_\perp^2}, \tag{48}$$

for class AA events.

---

**Class AB:** In this category we find the two equal $\chi^2$-minima at

$$\left(c_{\text{MLE},1}^2 - c_{t,1}^2\right)\bar{N}_1^2 - c_{t,2}^2\bar{M}^2 - \bar{N}_1\hat{\hat{Z}}_1 = 0, \qquad c_{\text{MLE},2} = 0, \tag{49}$$

which solve to give

$$c_{\text{MLE},1}^2 = c_{t,1}^2 + \frac{1}{\bar{N}_1}\hat{\hat{Z}}_1 + \frac{\bar{N}_2}{\bar{N}_1}\rho\,c_{t,2}^2, \qquad c_{\text{MLE},2} = 0. \tag{50}$$

An event falls into this category when $\nabla_{c_1}\chi^2(\bar{z};c)$ has three real roots and $\nabla_{c_2}\chi^2(\bar{z};c)$ has only one. This leads to the class conditions

$$\hat{\hat{Z}}_1 \geq -\rho\kappa_2 - \kappa_1, \qquad \hat{\hat{Z}}_\perp < -\sqrt{1-\rho^2}\,\kappa_2. \tag{51}$$

Plugging the maximum likelihood estimates back into the $\chi^2$-function results in

$$q_{c_{\text{true}}} = \hat{\hat{Z}}_1^2 - 2\sqrt{1-\rho^2}\,\kappa_2\,\hat{\hat{Z}}_\perp - \left(1-\rho^2\right)\kappa_2^2, \tag{52}$$

with the inverse transformation

$$\hat{\hat{Z}}_1 = \pm\sqrt{q_{c_{\text{true}}} + 2\sqrt{1-\rho^2}\,\kappa_2\,\hat{\hat{Z}}_\perp + (1-\rho^2)\kappa_2^2}, \tag{53}$$

for class AB events.

---

**Class BA:** In this category we find the two equal $\chi^2$-minima at

$$c_{\text{MLE},1} = 0, \qquad \left(c_{\text{MLE},2}^2 - c_{t,2}^2\right)\bar{N}_2^2 - c_{t,1}^2\bar{M}^2 - \rho\bar{N}_2\hat{\hat{Z}}_1 - \sqrt{1-\rho^2}\bar{N}_2\hat{\hat{Z}}_\perp = 0, \tag{54}$$

which solve to give

$$c_{\text{MLE},1} = 0, \qquad c_{\text{MLE},2}^2 = c_{t,2}^2 + \frac{1}{\bar{N}_2}\rho\hat{\hat{Z}}_1 + \frac{1}{\bar{N}_2}\sqrt{1-\rho^2}\,\hat{\hat{Z}}_\perp + \frac{\bar{N}_1}{\bar{N}_2}\rho\,c_{t,1}^2. \tag{55}$$

An event falls into this category when $\nabla_{c_1}\chi^2(\bar{z};c)$ has only one real root and $\nabla_{c_2}\chi^2(\bar{z};c)$ has three. This leads to the class conditions

$$\hat{\hat{Z}}_1 < \frac{\rho}{\sqrt{1-\rho^2}}\hat{\hat{Z}}_\perp - \kappa_1, \qquad \hat{\hat{Z}}_\perp \geq -\frac{1}{\sqrt{1-\rho^2}}\left(\kappa_2 + \rho\left(\hat{\hat{Z}}_1 + \kappa_1\right)\right). \tag{56}$$

Plugging the maximum likelihood estimates back into the $\chi^2$-function results in

$$q_{c_{\text{true}}} = \left(\rho\hat{\hat{Z}}_1 + \sqrt{1-\rho^2}\,\hat{\hat{Z}}_\perp\right)^2 - 2\left(1-\rho^2\right)\kappa_1\,\hat{\hat{Z}}_1 + 2\rho\sqrt{1-\rho^2}\,\kappa_1\,\hat{\hat{Z}}_\perp - \left(1-\rho^2\right)\kappa_1^2, \tag{57}$$

with the inverse transformation

$$
\begin{aligned}
\alpha_1 &= \rho^2, \qquad \beta_1 = 2\sqrt{1-\rho^2}\left(\rho\hat{\bar{Z}}_\perp - \sqrt{1-\rho^2}\kappa_1\right), \\
\gamma_1 &= (1-\rho^2)\hat{\bar{Z}}_\perp^2 + 2\rho\sqrt{1-\rho^2}\,\kappa_1\,\hat{\bar{Z}}_\perp - (1-\rho^2)\kappa_1^2 - q_{c_{\text{true}}}, \\
\hat{\bar{Z}}_1 &= -\frac{1}{2\alpha_1}\beta_1 \pm \frac{1}{2\alpha_1}\sqrt{\beta_1^2 - 4\alpha_1\gamma_1},
\end{aligned}
\tag{58}
$$

for class BA events.

**Class BB:** In this category we find one $\chi^2$-minimum at

$$
c_{\text{MLE},1} = 0, \qquad\qquad c_{\text{MLE},2} = 0. \tag{59}
$$

An event falls into this category when $\nabla_{c_1}\chi^2(\bar{z};c)$ and $\nabla_{c_2}\chi^2(\bar{z};c)$ both have only one real root. This leads to the class conditions

$$
\hat{\bar{Z}}_1 < -\rho\kappa_2 - \kappa_1, \qquad \hat{\bar{Z}}_\perp < -\frac{1}{\sqrt{1-\rho^2}}\left(\kappa_2 + \rho\left(\hat{\bar{Z}}_1 + \kappa_1\right)\right). \tag{60}
$$

Plugging the maximum likelihood estimates back into the $\chi^2$-function results in

$$
q_{c_{\text{true}}} = -2(\kappa_1 + \rho\kappa_2)\hat{\bar{Z}}_1 - 2\sqrt{1-\rho^2}\,\kappa_2\hat{\bar{Z}}_\perp - \left(\kappa_1^2 + \kappa_2^2 + 2\rho\kappa_1\kappa_2\right), \tag{61}
$$

with the inverse transformation

$$
\hat{\bar{Z}}_1 = \frac{1}{2(\kappa_1 + \rho\kappa_2)}\left[-2\sqrt{1-\rho^2}\,\kappa_2\hat{\bar{Z}}_\perp - \left(\kappa_1^2 + \kappa_2^2 + 2\rho\kappa_1\kappa_2 + q_{c_{\text{true}}}\right)\right], \tag{62}
$$

for class BB events.

To calculate $p_q\left(q_{c_{\text{true}}}\right)$, we must once again integrate over one of the degrees of freedom. Let us choose $\hat{\bar{Z}}_\perp$ as the integration parameter.[1] Let us explicitly integrate over the contribution from each class separately such that

$$
p_q\left(q_{c_{\text{true}}}\right) = \sum_{X \in \text{Classes}} \int_{\hat{\bar{Z}}_\perp} \left|\frac{\mathrm{d}\hat{\bar{Z}}_1^{(X)}}{\mathrm{d}q_{c_{\text{true}}}}\right| \mathcal{N}\left(\hat{\bar{Z}}_1^{(X)}\right) \mathcal{N}\left(\hat{\bar{Z}}_\perp\right) \mathrm{d}\hat{\bar{Z}}_\perp, \tag{63}
$$

where the set Classes = $\{AA+, AA-, AB+, AB-, BA+, BA-, BB\}$ now separates the positive and negative square-root solutions for $\hat{\bar{Z}}_1$ into separate classes. We must now determine the integral limits for each class. As in the linear x quadratic case, these are defined by combining the class boundary conditions with the requirement that all $\hat{\bar{Z}}_1$ be real solutions. We find that, as we manipulate these conditional statements, an inequality must change direction when we divide out a factor of $\rho < 0$. This means that the resulting integral limits are different for the cases $\rho \geq 0$ and $\rho < 0$. These are shown in Tables 1 and 2 respectively, where $q_1 = \rho\sqrt{1-\rho^2}\kappa_1$ and $q_2 = \sqrt{1-\rho^2}\sqrt{q - (1-\rho^2)\kappa_1^2}$.

With the integral limits defined, we may now use Eq 63 to evaluate the PDF. To help this computation, we identify the following two special cases. Firstly, the integral contribution from

---

[1]We can imagine using a different integration parameter, such as the polar angle, to construct an integral which does not diverge near domain boundaries. We don't add this extra complication here.

class AA+ is equal to $p_{\chi^2}^{(2)}\big(q_{c_{\text{true}}}\big)$ when $\sqrt{q_{c_{\text{true}}}} < \sqrt{1-\rho^2}\,\kappa_2$ and $\sqrt{q_{c_{\text{true}}}} < q_1 + q_2$. Secondly, when $c_{\text{true}} = (0,0)$ then the full PDF simplifies to

$$p_q\big(q_{c_{\text{true}}}\big) \;=\; \frac{1}{2}p_{\chi^2}^{(1)}\big(q_{c_{\text{true}}}\big) \;+\; \frac{1}{2}\int_0^{\sqrt{1-\rho^2}\sqrt{q_{c_{\text{true}}}}} p_{\chi^2}^{(1)}\Big(q_{c_{\text{true}}} - \hat{\hat{Z}}_\perp^2\Big)\mathcal{N}\big(\hat{\hat{Z}}_\perp\big)\mathrm{d}\hat{\hat{Z}}_\perp \;+\; \text{const}. \quad (64)$$

We note that the constants $\bar{N}_1$, $\bar{N}_2$ and $\rho$ fully define the PDF and may be extracted from an Asimov scan using the now-usual method.

### 4.3.1 An example with $\rho \geq 0$

Let us define an example with $\bar{n}_1 = (0.5,\ 0.1)$ and $\bar{n}_2 = (0.4,\ 0.4)$ which results in a strong positive correlation of $\rho = 0.83$.

Once again we compute our PDF integral numerically using Gaussian quadrature summation. Fig 6 compares the evaluated PDF with the distribution of pseudo-experiments. We observe that our PDF agrees well with the pseudo-experiments for all values of $c_{\text{true}}$ shown, whereas Wilks' theorem breaks down when either $c_{t,1}$ or $c_{t,2}$ approach 0. We observe that the PDF has a singularity at $q_{c_{\text{true}}} = 0$ when either $c_{t,1} = 0$ or $c_{t,2} = 0$.

When calculating the corresponding CDF, instead of computing how much density is contained within a singularity at $q_{c_{\text{true}}} = 0$, we simply anchor the CDF to unity at a value of $q_{c_{\text{true}}} = 40$. Fig 7 shows the expected coverage of the 95% confidence intervals obtained when using (left) Wilks' theorem and (right) our CDF. We see that assuming Wilks' theorem leads to a coverage as high as 98.75% for $c_{\text{true}} \sim (0,0)$. This is because three quadrants of the $\big(c_1^2, c_2^2\big)$-plane may be found with either $c_{\text{MLE},1}^2 = 0$ or $c_{\text{MLE},2}^2 = 0$, therefore the exclusion rate is one quarter of the intended 5%. By constrast, our method provides the correct coverage for all values of $c_{\text{true}}$.

We note that Figs 6 and 7 only show the 'positive-positive' quadrant of the $\big(c_{t,1}, c_{t,2}\big)$-plane. This is because we are only sensitive to the parameter values squared, therefore hypotheses related by $c_{t,1} \to -c_{t,1}$ and/or $c_{t,2} \to -c_{t,2}$ are degenerate. The other three quadrants are therefore identical up to reflection about the axes.

### 4.3.2 An example with $\rho < 0$

Let us define an example with $\bar{n}_1 = (-0.5,\ 0.3)$ and $\bar{n}_2 = (0.8,\ 0.1)$ which results in a strong negative correlation of $\rho = -0.79$.

Figure 8 compares our evaluated PDF distribution with pseudo-experiments. Once again, our method closely agrees with the pseudo-experiment distribution when $c_{\text{true}} \sim (0,0)$ where Wilks' theorem breaks down. Figure 9 shows the expected coverage of the 95% confidence limits obtained using (left) Wilks' theorem and (right) our method. Once again we see that assuming Wilks leads to over-coverage when $c_{\text{true}} \sim (0,0)$, whereas our method provides correct coverage for all $c_{\text{true}}$ values.

Table 1: Conditions defining when each class contributes to $p_q\left(q_{c_{\text{true}}}\right)$ when $\rho \geq 0$.

| | |
|---|---|
| **AA+** | $\max\left(-\sqrt{q_{c_{\text{true}}}}, -\sqrt{1-\rho^2}\kappa_2\right) \leq \hat{\hat{Z}}_\perp < \sqrt{q_{c_{\text{true}}}}$ and either $\hat{\hat{Z}}_\perp < \frac{\sqrt{1-\rho^2}}{\rho}\kappa_1$ or $q_1-q_2 \leq \hat{\hat{Z}}_\perp < q_1+q_2$ . |
| **AA−** | $\max\left(-\sqrt{q_{c_{\text{true}}}}, -\sqrt{1-\rho^2}\kappa_2\right) \leq \hat{\hat{Z}}_\perp < \min\left(\sqrt{q_{c_{\text{true}}}}, \frac{\sqrt{1-\rho^2}}{\rho}\kappa_1\right)$ and either $q < \sqrt{1-\rho^2}\kappa_1$ or $\hat{\hat{Z}}_\perp < q_1-q_2$ or $\hat{\hat{Z}}_\perp \geq q_1+q_2$ . |
| **AB+** | $-\frac{q_{c_{\text{true}}}+(1-\rho^2)\kappa_2^2}{2\sqrt{1-\rho^2}\kappa_2} \leq \hat{\hat{Z}}_\perp < -\sqrt{1-\rho^2}\kappa_2$ . |
| **AB−** | $-\frac{q_{c_{\text{true}}}+(1-\rho^2)\kappa_2^2}{2\sqrt{1-\rho^2}\kappa_2} \leq \hat{\hat{Z}}_\perp < \min\left(-\sqrt{1-\rho^2}\kappa_2, \frac{(\kappa_1+\rho\kappa_2)^2}{2\sqrt{1-\rho^2}\kappa_2} - \frac{q_{c_{\text{true}}}+(1-\rho^2)\kappa_2^2}{2\sqrt{1-\rho^2}\kappa_2}\right)$ . |
| **BA+** | $\frac{\sqrt{1-\rho^2}}{\rho}\kappa_1 \leq \hat{\hat{Z}}_\perp < \frac{\rho^2 q_{c_{\text{true}}}+(1-\rho^2)\kappa_1^2}{2\rho\sqrt{1-\rho^2}\kappa_1}$ and either $\hat{\hat{Z}}_\perp < q_1-q_2$ or $\hat{\hat{Z}}_\perp \geq q_1+q_2$ . |
| **BA−** | $\frac{\rho(q_{c_{\text{true}}}-\kappa_1^2-\kappa_2^2)-2\kappa_1\kappa_2}{2\sqrt{1-\rho^2}\kappa_1} \leq \hat{\hat{Z}}_\perp < \frac{\rho^2 q_{c_{\text{true}}}+(1-\rho^2)\kappa_1^2}{2\rho\sqrt{1-\rho^2}\kappa_1}$ and either $\hat{\hat{Z}}_\perp \geq \frac{\sqrt{1-\rho^2}}{\rho}\kappa_1$ or $q_1-q_2 \leq \hat{\hat{Z}}_\perp < q_q+q_2$ . |
| **BB** | $\frac{(\kappa_1+\rho\kappa_2)^2}{2\sqrt{1-\rho^2}\kappa_2} - \frac{q_{c_{\text{true}}}+(1-\rho^2)\kappa_2^2}{2\sqrt{1-\rho^2}\kappa_2} \leq \hat{\hat{Z}}_\perp < \frac{\rho(q_{c_{\text{true}}}-\kappa_1^2-\kappa_2^2)-2\kappa_1\kappa_2}{2\sqrt{1-\rho^2}\kappa_1}$ . |

Table 2: Conditions defining when each class contributes to $p_q\left(q_{c_{\text{true}}}\right)$ when $\rho < 0$.

| | |
|---|---|
| **AA+** | $\max\left(-\sqrt{q}, -\sqrt{1-\rho^2}\,\kappa_2\right) \leq \hat{\hat{Z}}_\perp < \sqrt{q}$ and either $\hat{\hat{Z}}_\perp \geq \frac{\sqrt{1-\rho^2}}{\rho}\kappa_1$ or $q < \sqrt{1-\rho^2}\,\kappa_1$ or $q_1-q_2 \leq \hat{\hat{Z}}_\perp < q_1+q_2$ . |
| **AA−** | $\max\left(-\sqrt{q}, -\sqrt{1-\rho^2}\,\kappa_2, \frac{\sqrt{1-\rho^2}}{\rho}\kappa_1\right) \leq \hat{\hat{Z}}_\perp < \sqrt{q}$ and either $q < \sqrt{1-\rho^2}\,\kappa_1$ or $\hat{\hat{Z}}_\perp < q_1-q_2$ or $\hat{\hat{Z}}_\perp \geq q_1+q_2$. |
| **AB+** | $-\frac{q+(1-\rho^2)\kappa_2^2}{2\sqrt{1-\rho^2}\,\kappa_2} \leq \hat{\hat{Z}}_\perp < -\sqrt{1-\rho^2}\,\kappa_2$ and either $\kappa_1 + \rho\kappa_2 \geq 0$ or $\hat{\hat{Z}}_\perp \geq \frac{(\kappa_1+\rho\kappa_2)^2}{2\sqrt{1-\rho^2}\kappa_2} - \frac{q+(1-\rho^2)\kappa_2^2}{2\sqrt{1-\rho^2}\,\kappa_2}$. |
| **AB−** | $-\frac{q+(1-\rho^2)\kappa_2^2}{2\sqrt{1-\rho^2}\kappa_2} \leq \hat{\hat{Z}}_\perp < \min\left(-\sqrt{1-\rho^2}\kappa_2, \frac{(\kappa_1+\rho\kappa_2)^2}{2\sqrt{1-\rho^2}\kappa_2} - \frac{q+(1-\rho^2)\kappa_2^2}{2\sqrt{1-\rho^2}\kappa_2}\right)$ and $\kappa_1+\rho\kappa_2 \geq 0$. |
| **BA+** | $\frac{\rho^2 q+(1-\rho^2)\kappa_1^2}{2\rho\sqrt{1-\rho^2}\kappa_1} \leq \hat{\hat{Z}}_\perp < \min\left(\frac{\sqrt{1-\rho^2}}{\rho}\kappa_1, \frac{\rho(q-\kappa_1^2-\kappa_2^2)-2\kappa_1\kappa_2}{2\sqrt{1-\rho^2}\kappa_1}\right)$ and $\kappa_1+\rho\kappa_2 < 0$ and either $\hat{\hat{Z}}_\perp < q_1-q_2$ or $\hat{\hat{Z}}_\perp \geq q_1+q_2$. |
| **BA−** | $\hat{\hat{Z}}_\perp \geq \frac{\rho^2 q+(1-\rho^2)\kappa_1^2}{2\rho\sqrt{1-\rho^2}\kappa_1}$ and either $\hat{\hat{Z}}_\perp < \frac{\sqrt{1-\rho^2}}{\rho}\kappa_1$ or $q_1-q_2 \leq \hat{\hat{Z}}_\perp < q_1+q_2$ and either $\kappa_1+\rho\kappa_2 < 0$ or $\hat{\hat{Z}}_\perp \geq \frac{\rho(q-\kappa_1^2-\kappa_2^2)-2\kappa_1\kappa_2}{2\sqrt{1-\rho^2}\kappa_1}$ . |
| **BB** | $\frac{(\kappa_1+\rho\kappa_2)^2}{2\sqrt{1-\rho^2}\kappa_2} - \frac{q+(1-\rho^2)\kappa_2^2}{2\sqrt{1-\rho^2}\kappa_2} \leq \hat{\hat{Z}}_\perp < \frac{\rho(q-\kappa_1^2-\kappa_2^2)-2\kappa_1\kappa_2}{2\sqrt{1-\rho^2}\kappa_1}$ if $\kappa_1+\rho\kappa_2 \geq 0$ otherwise $\frac{\rho(q-\kappa_1^2-\kappa_2^2)-2\kappa_1\kappa_2}{2\sqrt{1-\rho^2}\kappa_1} \leq \hat{\hat{Z}}_\perp < \frac{(\kappa_1+\rho\kappa_2)^2}{2\sqrt{1-\rho^2}\kappa_2} - \frac{q+(1-\rho^2)\kappa_2^2}{2\sqrt{1-\rho^2}\kappa_2}$. |

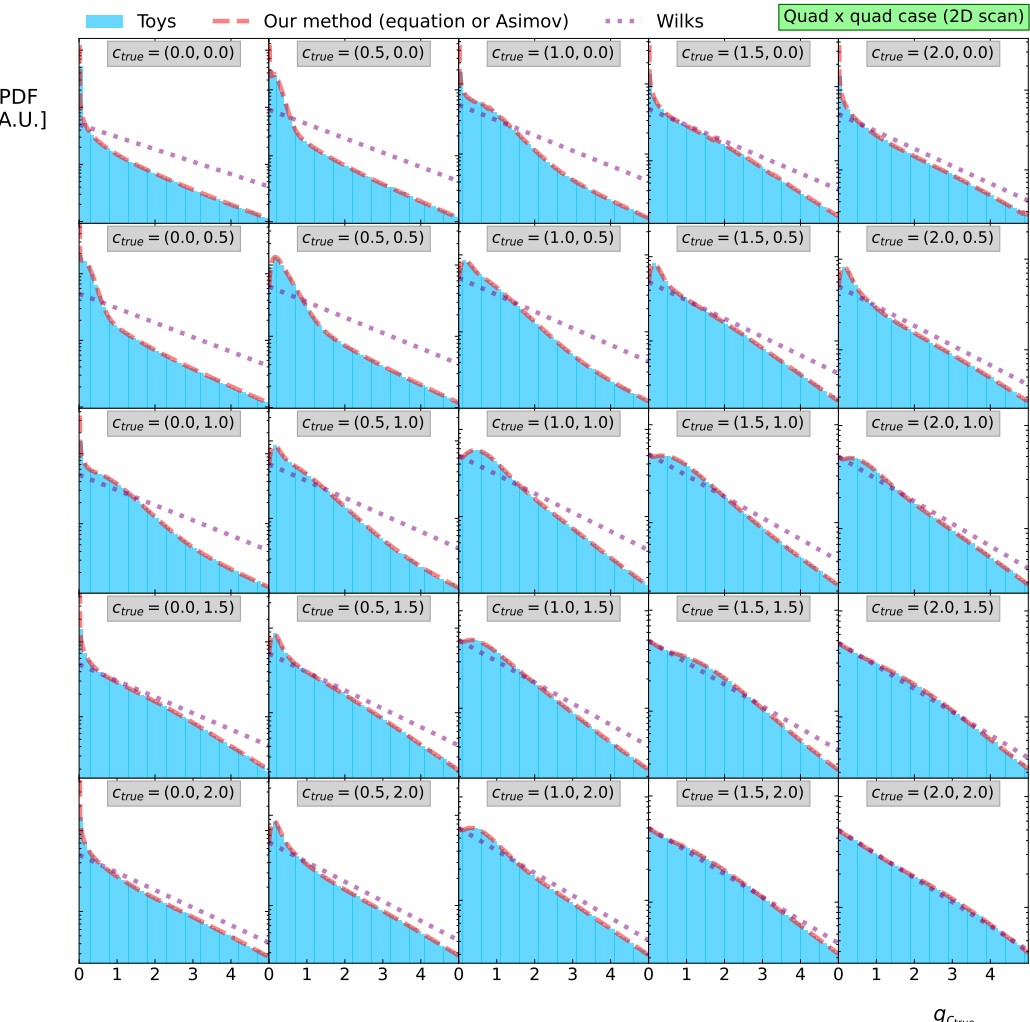

Figure 6: Density of pseudo-experiments compared with our method for a 2D example with $\bar{n}_1 = (0.5, 0.1)$, $\bar{n}_2 = (0.4, 0.4)$ and a **strong positive correlation** of $\rho = 0.83$. Our method is able to describe cases with $c_{\text{true}} \sim (0,0)$ where Wilks' theorem is violated.

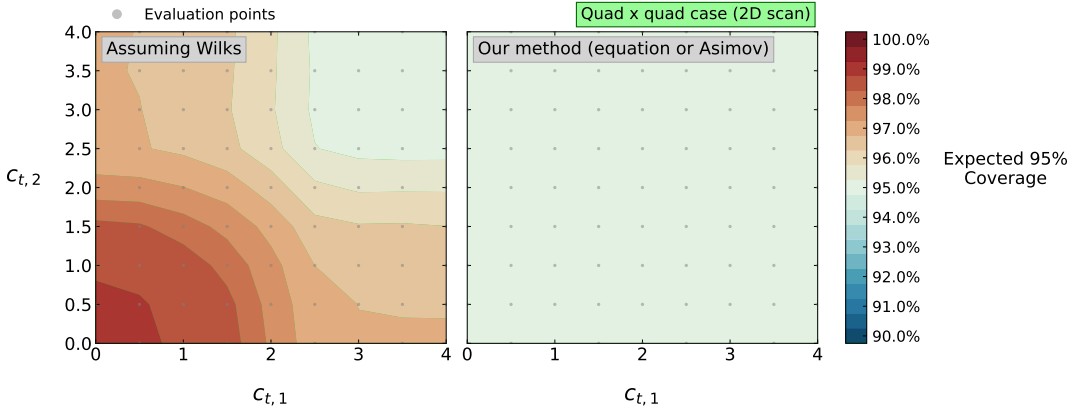

Figure 7: Expected 95% coverage assuming Wilks (left) and using our method (right) for a 2D example with $\bar{n}_1 = (0.5, 0.1)$, $\bar{n}_2 = (0.4, 0.4)$ and a **strong positive correlation** of $\rho = 0.83$. Our method provides correct coverage for all $c_{\text{true}}$.

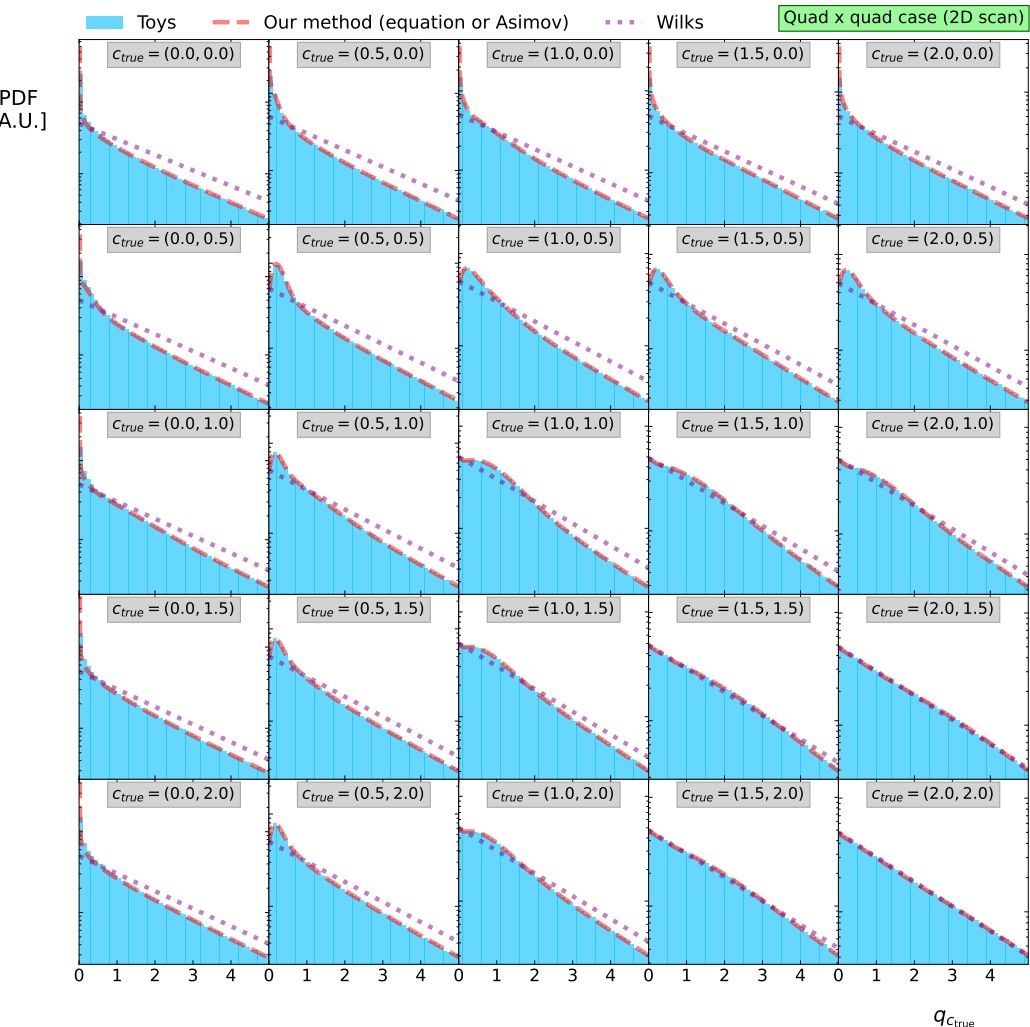

Figure 8: Density of pseudo-experiments compared with our method for a 2D example with $\bar{n}_1 = (-0.5, 0.3)$, $\bar{n}_2 = (0.8, 0.1)$ and a **strong negative correlation** of $\rho = -0.79$. Our method is able to describe cases with $c_{\text{true}} \sim (0, 0)$ where Wilks' theorem is violated.

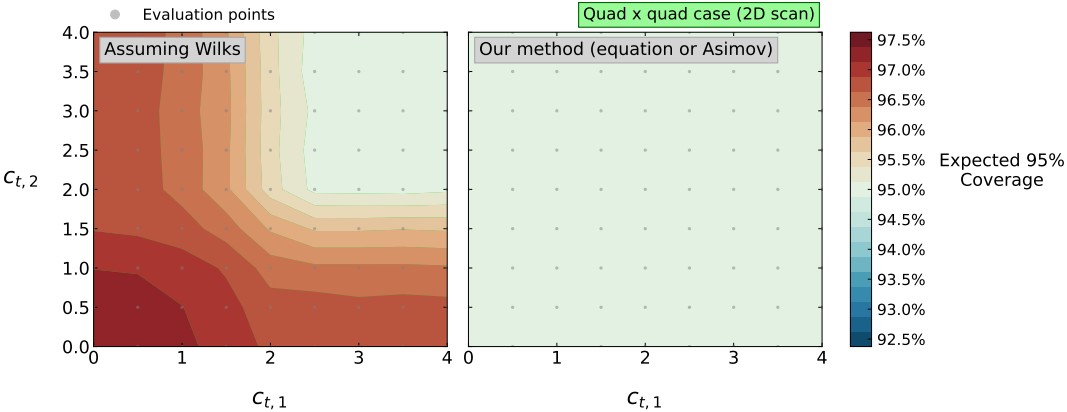

Figure 9: Expected 95 % coverage assuming Wilks (left) and using our method (right) for a 2D example with $\bar{n}_1 = (-0.5, 0.3)$, $\bar{n}_2 = (0.8, 0.1)$ and a **strong negative correlation** of $\rho = -0.79$. Our method provides correct coverage for all $c_{\text{true}}$.

# 5 Solution with one (linear and quadratic) parameter

We now return to the one-parameter case, but allow both the linear and quadratic components to be non-zero. We refer to this as the "linear plus quadratic" case. First we will consider a single-bin measurement. This may be solved using a similar method to the linear and quadratic cases.

We will then consider a multi-bin measurement. This turns out to be significantly harder to solve. The intuition behind this is as follows. In the purely linear case, the new physics contribution is always parallel to the unit-vector $\hat{\bar{l}}$, and therefore the observation may be summarised by the random variable $\hat{\bar{Z}}_l = \bar{z} \cdot \hat{\bar{l}}$. Following the same logic, in the purely quadratic case we may summarise a measurement by the random variable $\hat{\bar{Z}}_n = \bar{z} \cdot \hat{\bar{n}}$. However, in the "linear plus quadratic" case we find a new physics contribution of $c\bar{l} + c^2\bar{n}$. This vector changes direction as we profile $c$, provided that $\bar{l}$ and $\bar{n}$ are not parallel. This means that our measurement may not be sufficiently summarised by the projection onto a single unit-vector. Appendix C presents a geometric explanation of how this leads to both non-Gaussian and non-$\chi^2$-like behaviour in the distribution of $q_{c_{\text{true}}}$. In the current section we solve the true form of this distribution using a partly-numerical algorithm.

## 5.1 Single-bin measurements

Let us define a 'predictor function' $\bar{k}(c)$ as

$$\bar{k}(c) = (c - c_{\text{true}})\bar{l} + (c^2 - c_{\text{true}}^2)\bar{n}, \tag{65}$$

with the derivative $\nabla_c \bar{k}(c) = \bar{l} + 2c\bar{n}$. We see that $\bar{k}(c)$ has a minimum value of

$$\bar{k}_{\text{min}} = -\frac{\bar{l}^2}{4\bar{n}} - c_{\text{true}}\bar{l} - c_{\text{true}}^2\bar{n}, \tag{66}$$

which occurs when $c = c_{\text{thresh}} := -\frac{\bar{l}}{2\bar{n}}$. We can then write $\chi^2(\bar{z}; c) = |\bar{z} - \bar{k}(c)|^2$, which has the derivative

$$\nabla_c \chi^2(\bar{z}; c) = 2(\bar{z} - \bar{k}(c)) \cdot \nabla_c \bar{k}(c). \tag{67}$$

For a one-bin measurement, the quantities $\bar{z}$, $\bar{l}$ and $\bar{n}$ are simply numbers and we find the maximum likelihood estimates

$$
\begin{aligned}
c_{\text{MLE}}(\bar{z}) &= \begin{cases} -\frac{1}{2\bar{n}}\left[\bar{l} \pm \sqrt{4\bar{n}(\bar{z} - \bar{k}_{\text{min}})}\right] & \text{if } \bar{z} \geq \bar{k}_{\text{min}} \\ c_{\text{thresh}} & \text{otherwise} \end{cases}, \\
\bar{k}_{\text{MLE}}(\bar{z}) &= \begin{cases} \bar{z} & \text{if } \bar{z} \geq \bar{k}_{\text{min}} \\ \bar{k}_{\text{min}} & \text{otherwise} \end{cases}.
\end{aligned}
\tag{68}
$$

This tells us that $c_{\text{MLE}}$ becomes fixed to the boundary value $c_{\text{thresh}}$ if $\bar{z} < \bar{k}_{\text{min}}$, resulting in the solution $\bar{k}_{\text{MLE}} = \bar{k}_{\text{min}}$. Above this threshold we find that $\bar{k}_{\text{MLE}}$ behaves as a Gaussian random variable. Solving for $q_{c_{\text{true}}}$ we find

$$
q_{c_{\text{true}}} = \begin{cases} \bar{z}^2 & \text{if } \bar{z} \geq \bar{k}_{\text{min}} \\ 2\bar{k}_{\text{min}}\bar{z} - \bar{k}_{\text{min}}^2 & \text{otherwise} \end{cases}, \tag{69}
$$

and we obtain the final distribution

$$
p_q(q_{c_{\text{true}}}) = \begin{cases} p_{\chi^2}^{(1)}(q_{c_{\text{true}}}) & \text{if } q_{c_{\text{true}}} < \bar{k}_{\text{min}}^2 \\ \frac{1}{2}p_{\chi^2}^{(1)}(q_{c_{\text{true}}}) + \mathcal{G}(q_{c_{\text{true}}}; -\bar{k}_{\text{min}}^2, 2\bar{k}_{\text{min}}) & \text{otherwise} \end{cases}. \tag{70}
$$

### 5.1.1 Connection to the purely quadratic case

We see that the form of Eq 70 is identical to the fully quadratic case shown in Eq 11. This is because the single-bin "linear plus quadratic" problem is actually just a reparameterision of the quadratic one. We can see this by re-writing the "linear plus quadratic" predictor function such that

$$
\begin{aligned}
\bar{k}(c) &= (c - c_{\text{true}})\bar{l} + (c^2 - c_{\text{true}}^2)\bar{n} \\
&= \left(\sqrt{\bar{n}}\, c + \frac{\bar{l}}{2\sqrt{\bar{n}}}\right)^2 - \left(\sqrt{\bar{n}}\, c_{\text{true}} + \frac{\bar{l}}{2\sqrt{\bar{n}}}\right)^2 \\
&\equiv \left(c'^2 - c'^2_{\text{true}}\right),
\end{aligned}
\tag{71}
$$

where $c' := \sqrt{\bar{n}}\, c + \frac{\bar{l}}{2\sqrt{\bar{n}}}$. We can therefore solve the distribution of $q_{c_{\text{true}}}$ in the single-bin "linear plus quadratic" case by applying the purely quadratic solution in this alternative parameterisation, reproducing the result of Eq 70.

We may understand the relationship between the two approaches as follows. When we use the quadratic form, we calculate $\chi^2$ by linearly combining the normally distributed random variable $\bar{z}$ with the Wilson co-efficient (in this case $c'$). Upon $\chi^2$-maximisation, this causes the maximum likelihood estimate $c'_{\text{MLE}}$ to become normally distributed up to it's boundary, at which point it becomes fixed. When using the "linear plus quadratic" parameterisation, we calculate $\chi^2$ by linearly combining $\bar{z}$ with $\bar{k}$, not $c$. The remaining steps in the derivation remain identical, but now using $\bar{k}$ as the random-variable-of-interest.

However, this reparameterisation may only be performed in the single-bin case. This is because it is not possible to re-write a single Wilson coefficient in a way which simultaneously transforms multiple bins into a purely quadratic form (except in the trivial case when multiple bins follow precisely the same linear and quadratic dependence). In general, we may perform this transformation for any one bin of the measurement, but all other bins remain in a "linear plus quadratic" form. We therefore cannot use such a reparameterisation to trivially solve the more difficult multi-bin case.

### 5.1.2 An example

Let us define an example with $\bar{l} = 0.3$ and $\bar{n} = 0.5$, for which $c_{\text{thresh}} = -0.3$. Fig 10a shows the distributions of $c_{\text{MLE}}$. When $\bar{z} \geq \bar{k}_{\text{min}}$, these contain two degenerate smooth but non-Gaussian peaks, otherwise they are distributed as a delta function at $c_{\text{thresh}}$. Fig 10b shows the distributions of $\bar{k}_{\text{MLE}}$. These are normally distributed above $\bar{k}_{\text{min}}$, with a delta function at $\bar{k}_{\text{min}}$ otherwise.

The distribution of $\bar{k}_{\text{MLE}}$ is analogous to the quadratic case studied previously, in which $c_{\text{MLE}}^2$ was Gaussian distributed above a boundary at 0, the smallest possible data fluctuation expressible by the model, and as a delta function on this boundary. The intuition is identical as we now study $\bar{k}_{\text{MLE}}$, except that $\bar{k}_{\text{MLE}}$ reaches a boundary at a value of $\bar{k}_{\text{min}}$ which is non-zero and changes as a function of $c_{\text{true}}$.

Fig 10c shows the pseudo-experiment analysis for this example. Assuming Wilks' theorem leads to a significant mis-modelling of the PDF and CDF when $c_{\text{true}}$ is close to $c_{\text{thresh}}$, and the corresponding 95% confidence intervals have an expected coverage of 97.5%. The true distribution tends towards a $\chi^2$-distribution when $c_{\text{true}}$ is far from $c_{\text{thresh}}$, therefore the expected coverage approaches 95%. By contrast, our method provides a good description of the PDF and CDF and good coverage for all values of $c_{\text{true}}$.

(a) Distribution of $c_{\mathrm{MLE}}$.

(b) Distribution of $\bar{k}_{\mathrm{MLE}}$.

(c) Distributions of the PDF and CDF, and the expected coverage.

Figure 10: Analysis of a one-bin "linear plus quadratic" example with $\bar{l} = 0.3$ and $\bar{n} = 0.5$.

## 5.2 Multi-bin measurements

We will now solve the general multi-bin case. To do this, we first determine how to derive $q_{c_{\text{true}}}$ from the independent normally distributed random variables $\bar{z}$. We call this the forwards transformation, and it progresses through a sequence of latent variables. We then derive the inverse transformation for each latent step, and learn any constraints which limit their accessible values. This will tell us (i) all the different channels through which $q_{c_{\text{true}}}$ may be obtained, and (ii) the full inverse transformation through each channel. We may then apply the change-of-variables formula to obtain the PDF over $q_{c_{\text{true}}}$. In the interest of efficiency, these steps are presented as compactly as possible. Further details are presented in Appendix D for the interested reader.

### 5.2.1 The forwards transformation

Let us write $\bar{x} = c_{\text{true}}\bar{l} + c_{\text{true}}^2 \bar{n} + \bar{z}$, so that the $\chi^2$ may be written as

$$\chi^2(\bar{x};c) = \left| \bar{x} - c\,\bar{l} - c^2\,\bar{n} \right|^2 . \tag{72}$$

To find the global minimum of Eq 72, we must solve the cubic equation

$$\nabla_c \chi^2(\bar{x};c) = -2\left[ Ac^3 + Bc^2 + Cc + D \right] = 0, \tag{73}$$

where

$$A = -2\bar{N}^2, \quad B = -3\bar{M}^2, \quad C = 2\bar{X}_n - \bar{L}^2, \quad D = \bar{X}_l . \tag{74}$$

We note that $\{A, B\}$ are constants, whereas $\{\bar{X}_n, \bar{X}_l, C, D\}$ are Gaussian random variables obtained as linear transformations of $\bar{Z}_l$ and $\bar{Z}_n$ using

$$\begin{aligned}
\bar{X}_l &= \bar{x}\cdot\bar{l} = c_{\text{true}}\bar{L}^2 + c_{\text{true}}^2\bar{M}^2 + \bar{Z}_l, \\
\bar{X}_n &= \bar{x}\cdot\bar{n} = c_{\text{true}}\bar{M}^2 + c_{\text{true}}^2\bar{N}^2 + \bar{Z}_n.
\end{aligned} \tag{75}$$

From this we see that the problem is summarised by only two degrees of freedom, here represented by $\bar{X}_l$ and $\bar{X}_n$, regardless of the dimensionality of $\bar{z}$. We solve this cubic equation analytically by forming the intermediate quantities

$$\begin{aligned}
\Delta_0 &= B^2 - 3AC, \\
\Delta_1 &= 2B^3 - 9ABC + 27A^2D, \\
\Gamma &= \sqrt[3]{\frac{\Delta_1 + \sqrt[2]{\Delta_1^2 + 4\Delta_0^3}}{2}},
\end{aligned} \tag{76}$$

where $\sqrt[2]{\ldots}$ is the positive square root and we select the principal cube root $\sqrt[3]{z} = |z|\exp\left[\frac{1}{3}i\arctan\frac{\text{Im}[z]}{\text{Re}[z]}\right]$. The three roots of Eq 73 are then given by

$$c_i = -\frac{1}{3A}\left( B + \xi^i\Gamma + \frac{\Delta_0}{\xi^i\Gamma} \right), \tag{77}$$

where $i \in \{0, 1, 2\}$ and $\xi = \frac{1}{2}\left(-1 + i\sqrt{3}\right)$. We obtain $c_{\text{MLE}}$ by selecting the real root that results in the smallest $\chi^2$, and compute $q_{c_{\text{true}}} = \chi^2(\bar{x};c_{\text{true}}) - \chi^2(\bar{x};c_{\text{MLE}})$.

### 5.2.2 PDFs of $\Delta_0$ and $\Delta_1$

We know that $\hat{\bar{Z}}_l$ and $\hat{\bar{Z}}_n$ are Gaussian random variables centred on zero with correlation $\rho$ and widths of $\bar{L}$ and $\bar{N}$ respectively. Propagating through the linear transformations described by Eqs 74-76, we find that $\Delta_0$ and $\Delta_1$ are **independent Gaussian random variables** with densities $p_{\Delta_0}(\Delta_0) = \mathcal{G}(\Delta_0; \mu_{\Delta_0}, \sigma_{\Delta_0})$ and $p_{\Delta_1}(\Delta_1) = \mathcal{G}(\Delta_1; \mu_{\Delta_1}, \sigma_{\Delta_1})$ defined by

$$
\begin{aligned}
\mu_{\Delta_0} &= B^2 - 3A\big(2c_{\text{true}}\bar{M}^2 + 2c_{\text{true}}^2\bar{N}^2 - \bar{L}^2\big), \\
\sigma_{\Delta_0} &= 12\bar{N}^3, \\
\mu_{\Delta_1} &= 2B^3 - 9AB\big(2c_{\text{true}}\bar{M}^2 + 2c_{\text{true}}^2\bar{N}^2 - \bar{L}^2\big) + 27A^2\big(c_{\text{true}}\bar{L}^2 + c_{\text{true}}^2\bar{M}^2\big), \\
\sigma_{\Delta_1} &= 108\sqrt{1-\rho^2}\,\bar{L}\,\bar{N}^4.
\end{aligned}
\tag{78}
$$

Since $\Delta_0$ and $\Delta_1$ are independent, may be easily calculated from $\bar{z}$, and follow known distributions, we may consider them to be the fundamental random variables that summarise a measurement. We will therefore formulate the PDF over $q_{c_{\text{true}}}$ in terms of $p_{\Delta_0}$ and $p_{\Delta_1}$, instead of inverting all the way back to $\bar{z}$.

### 5.2.3 Inverting $\Gamma$

The latent variable $\Gamma$ is a complex number. Say that $\Delta_0$ is externally chosen. For a given $\Gamma$, we can find $\Delta_1$ by inverting Eq 76 to find

$$
\Delta_1 = \Gamma^3 + \left(\frac{\Delta_0}{\Gamma}\right)^3.
\tag{79}
$$

There are several conditions which $\Delta_1$ must satisfy. In Eq 76 we calculated $\Gamma$ using only the **positive square root** and **principal cube root**. This requires:

- Either $\text{Re}\big[\Gamma^3 - \frac{1}{2}\Delta_1\big] \geq 0$ with zero imaginary part, or $\text{Im}\big[\Gamma^3 - \frac{1}{2}\Delta_1\big] > 0$ with zero real part.

- $\text{Re}[\Gamma] \geq 0$ and $\text{Im}[\Gamma] \geq 0$.

Furthermore, we know from section 5.2.2 that $\Delta_1$ is a real-valued random variable. We must therefore enforce $\text{Im}[\Delta_1] = 0$ on all inverse transformations. Writing $\Gamma = \Gamma_r + i\Gamma_i$ and expanding Eq 79, this becomes

$$
\text{Im}[\Delta_1] = \Gamma_i\big(3\Gamma_r^2 - \Gamma_i^2\big)\left(1 - \left(\frac{\Delta_0}{\Gamma_r^2 + \Gamma_i^2}\right)\right) = 0.
\tag{80}
$$

Exploiting this factorisation, we find three different modes in the $\Gamma$ distribution. We categorise these events as Types A-C and define them as follows:

$$
\begin{array}{ll}
\textbf{Type A} & \Gamma_i = 0, \\
\textbf{Type B} & 3\Gamma_r^2 = \Gamma_i^2, \\
\textbf{Type C} & \Delta_0 = \Gamma_r^2 + \Gamma_i^2.
\end{array}
\tag{81}
$$

### 5.2.4 Inverting $c_0$

Let us now progress one step further through the transformation to consider the stationary points $c_0$ and $c_1$. We find that $c_2$ will always be either a local $\chi^2$-maximum or a complex minimum. This means that it can never correspond to a real-valued global minimum, and we do not need to consider it further.

Let us define $\beta_{c_0} := B + 3Ac_0$. Inverting Eq 77 we find

$$\Gamma^{\pm} = -\frac{1}{2}\beta_{c_0} \pm \frac{1}{2}\sqrt{\beta_{c_0}^2 - 4\Delta_0}\,, \tag{82}$$

where we use the superscript $\pm$ to distinguish positive and negative square-root solutions. Since the transformation through latent variable $\Gamma$ separated events into three distinct modes, we also find three modes in the $c_0$ distribution. Let us consider these separately.

For Type A events, we require $\Gamma_i = 0$. This is satisfied when $\text{Im}\left[\beta_{c_0}\right] = 0$ and $\Delta_0 \leq \frac{1}{4}\beta_{c_0}^2$. Therefore $c_0$ is always real for Type A events, but only exists for a limited range of $\Delta_0$. We find that only $\Gamma^+$ satisfies the conditions placed on $\Gamma$ for Type A events.

For Type B events, we require $3\Gamma_r^2 = \Gamma_i^2$. Since events with $\Gamma_r = 0$ are already placed in the Type A category, this means that all Type B events must have have $\text{Im}[c_0] \neq 0$. This means they cannot be real solutions for the $\chi^2$-minimum. As we progress forwards through the transformation, complex values of $c_0$ will not contribute to the density over $q_{c_{\text{true}}}$, and so we will not consider these further.

For Type C events, we find the solution $\Gamma_r = -\frac{1}{2}\beta_{c_0}$ and $\Gamma_i = \sqrt{\Delta_0 - \Gamma_r^2}$. Since $\Gamma_r$ and $\Gamma_i$ must be real, we find that $c_0$ is always real for Type B events, but only exists when $\Delta_0 > \frac{1}{4}\beta_{c_0}^2$. We find that only $\Gamma^-$ satisfies the conditions placed on $\Gamma$ for Type C events.

### 5.2.5 Inverting $c_1$

Let us define $\beta_{c_1} := B + 3Ac_1$. Inverting Eq 77 we find

$$\Gamma^{\pm} = -\frac{1}{2\xi}\beta_{c_1} \pm \frac{1}{2\xi}\sqrt{\beta_{c_1}^2 - 4\Delta_0}\,. \tag{83}$$

For Type A events, we require $\Gamma_i = 0$. Since Eq 83 includes factors of $\xi$, we find that this is only possible if $c_1$ is complex. As we are not interested in complex solutions to $c_1$, we will not consider Type A solutions further.

For Type B events, we require $\Gamma_i^2 = 3\Gamma_r^2$. We find that this is satisfied by the solution $\Gamma^+$ provided that $\Delta_0 \leq \frac{1}{4}\beta_{c_1}^2$. For Type C events, we require $\Delta_1 = \Gamma_r^2 + \Gamma_i^2$. We find that

$$\Gamma_r = \beta_{c_1} - \sqrt{3}\,\Gamma_i\,, \qquad \Gamma_i = \frac{1}{4}\left(\sqrt{3}\,\beta_{c_1} + \sqrt{4\Delta_0 - \beta_{c_1}^2}\right), \tag{84}$$

corresponding to the solution $\Gamma^-$ exists when $\Delta_0 > \frac{1}{4}\beta_{c_1}^2$.

### 5.2.6 Inverting $c_{\text{MLE}}$

We have found that four different channels provide candidates for the global $\chi^2$-minimum. These are Type A events, which must correspond to the solutions $c_0$, Type B events, which must correspond to $c_1$, or Type C events, for which the solutions may be either $c_0$ or $c_1$. Furthermore, we have derived several conditions that the solutions in each channel must satisfy. In the forwards direction, $c_{\text{MLE}}$ may be obtained by selecting whichever of these four solutions is (i) real, (ii) satisfies the conditions of the corresponding channel and (iii) provides the smallest $\chi^2$ if conditions (i) and (ii) are met. In the inverse direction, we plug $c_{\text{MLE}}$ into all four channels and select whichever stationary point satisfies conditions (i)-(iii).

### 5.2.7 PDF of $q_{c_{\text{true}}}$

The final step in our forwards transformation takes us from $c_{\text{MLE}}$ to $q_{c_{\text{true}}}$. To invert this, we must find the roots of the *quartic* equation

$$q_4\,c_{\text{MLE}}^4 + q_3\,c_{\text{MLE}}^3 + q_2\,c_{\text{MLE}}^2 + q_1\,c_{\text{MLE}} + q_0 = 0\,, \tag{85}$$

where

$$q_4 = \bar{N}^2, \quad q_3 = 2\bar{M}^2, \quad q_2 = \bar{L}^2 - 2\bar{X}_n, \quad q_1 = -2\bar{X}_l,$$
$$q_0 = q_{c_{\text{true}}} + 2c_{\text{true}}\bar{X}_l + 2c_{\text{true}}^2\bar{X}_n - c_{\text{true}}^2\bar{L}^2 - 2c_{\text{true}}^3\bar{M}^2 - c_{\text{true}}^4\bar{N}^2. \tag{86}$$

We can solve this analytically using the formula provided in Appendix E, from which we obtain the four roots $c_{\text{MLE}}^{(1)} - c_{\text{MLE}}^{(4)}$. All four of these roots may contribute to the density over $q_{c_{\text{true}}}$.

Unfortunately we now encounter a problem: to calculate the factors $\{q_0, q_1, q_2\}$, we had to already know $\{\Delta_0, \Delta_1\}$ from which to obtain $\{\bar{X}_l, \bar{X}_n\}$. We will calculate our PDF by performing an integral over $\Delta_0$, therefore this value is externally selected and so poses no problem. However, **we do not know the value of $\Delta_1$ until after we have completed the inverse transformation** $\{q_{c_{\text{true}}}, \Delta_0\} \to \Delta_1$.

To solve the PDF, we must find the values of $\Delta_1$ for which the inverse transformation $\{q_{c_{\text{true}}}, \Delta_0\} \to \Delta_1$ results in the same value of $\Delta_1$ as that which is assumed when performing the first step $\{q_{c_{\text{true}}}, \Delta_0, \Delta_1\} \to c_{\text{MLE}}$. We have not been able to find an analytic solution to this problem. Instead we leave this for future work and fall back to a numerical method here, noting that this is likely computationally slower.

Our numerical method proceeds as follows. For a given pair $\{q_{c_{\text{true}}}, \Delta_0\}$, we scan across the support of $\Delta_1$ values. At each step we hypothesise this value to be a real solution and call this $\Delta_1^{\text{hyp}}$. We then reconstruct $c_{\text{MLE}}^{(1)-(4)}$, and for each one follow the full inverse transformation to reconstruct $\Delta_1$, and call this $\Delta_1^{\text{reco}}$. We then use the `iMinuit` package [22, 23] to find all points for which

$$\left| \Delta_1^{\text{hyp}} - \Delta_1^{\text{reco}} \right| = 0. \tag{87}$$

We then ask whether each solution $\{c_{\text{MLE}}, \Delta_1\}$ satisfies the validity conditions for any of the channels A, B, C($c_0$) or C($c_1$). If so, they are counted in the list of allowed $\Delta_1|q_{c_{\text{true}}}, \Delta_0$. The PDF for $q_{c_{\text{true}}}$ is then calculated as

$$p_q(q_{c_{\text{true}}}) = \int_{-\infty}^{+\infty} \sum_{\Delta_1|q_{c_{\text{true}}}, \Delta_0} \left| \frac{\mathrm{d}\Delta_1}{\mathrm{d}q_{c_{\text{true}}}} \right| p_{\Delta_0}(\Delta_0) \, p_{\Delta_1}(\Delta_1) \, \mathrm{d}\Delta_0, \tag{88}$$

where the Jacobian factor is obtained by using the `JAX` auto-differentiation package [24] to calculate $\frac{\mathrm{d}q_{c_{\text{true}}}}{\mathrm{d}\Delta_1}$, which is the reciprocal of the Jacobian factor because $q_{c_{\text{true}}}$ and $\Delta_1$ are both real numbers.

### 5.2.8 The final algorithm

The final method for computing the PDF $p_q(q_{c_{\text{true}}})$ is summarised as follows: use Gaussian quadrature summation to compute the integral of Eq 88 over an arbitrarily wide interval of e.g. $\mu_{\Delta_0} - 5\sigma_{\Delta_0} \leq \Delta_0 \leq \mu_{\Delta_0} + 5\sigma_{\Delta_0}$, where the set "$\Delta_1|q_{c_{\text{true}}}, \Delta_0$" is obtained using Algorithm 1. The Jacobian factor is calculated as **the reciprocal of** $\frac{\mathrm{d}q_{c_{\text{true}}}}{\mathrm{d}\Delta_1}$, which is obtained by applying the `JAX` auto-differentiation package to the forwards transformation.

A drawback of our method is that it relies upon two nested numerical procedures: one to perform the integration over $\Delta_0$, and a second to obtain $\Delta_1$ at each step. It is not obvious that this is computationally less demanding than using pseudo-experiments to compute $p$-values with a desired level of precision. However, we note that the complexity of our method increases linearly with precision, whereas pseudo-experiments scale quadratically, and so our method always performs better in the high-precision regime.

A second drawback is that our method cannot resolve different $\Delta_1$ solutions closer than a tolerance of $\delta_{\text{min}} \cdot \sigma_{\Delta_1}$. In Appendix D we see that this occurs naturally whenever $q_{c_{\text{true}}}$ approaches 0. Therefore we will always underestimate the density at $q_{c_{\text{true}}} \approx 0$ by approximately 50%. Currently we mitigate this problem by choosing $\delta_{\text{min}}$ to be small.

---

**Algorithm 1** Identify all values of $\Delta_1$ consistent with a given $\left(q_{c_\text{true}}, \Delta_0\right)$-pair.

---

**Input:** $\left(q_{c_\text{true}}, \Delta_0\right)$ and configurable constants $n_\sigma \sim 5$, $s_\text{min} \sim 5e-3$ and $\delta_\text{min} \sim 1e-3$
**Returns:** List of $\Delta_1$ values of length 0, 1 or 2

1: Create empty list
2: **for all** $\Delta_1^\text{hyp}$ in a broad linear scan between $\mu_{\Delta_1} - n_\sigma \sigma_{\Delta_1}$ and $\mu_{\Delta_1} + n_\sigma \sigma_{\Delta_1}$ **do**
3:      **for** first two $c_\text{MLE}$ solutions of quartic equation $q_{c_\text{true}}\left(c_\text{MLE}, \Delta_0, \Delta_1^\text{hyp}\right)$ **do**
4:          Obtain $\Delta_1^\text{reco}$ using inverse transformation of $(c_\text{MLE}, \Delta_0)$
5:          Calculate score $s \leftarrow |\Delta_1^\text{hyp} - \Delta_1^\text{reco}|$
6:          **Require:** $s$ is in a local minimum (refine $\Delta_1^\text{hyp}$ using `iMinuit` optimisation)
7:          **Require:** $s < s_\text{min}$
8:          **Require:** $\Delta_1^\text{hyp}$ solution satisfies Class $A$, $B$, $C(c_0)$ or $C(c_1)$ conditions
9:          **Require:** $|\Delta_1^\text{hyp} - \Delta_1^i| > \delta_\text{min} \cdot \sigma_{\Delta_1}$ for all $\Delta_1^i$ already in list
10:         Add $\Delta_1^\text{hyp}$ to list
11:     **end for**
12: **end for**
13: Return list

---

We note that Algorithm 1 only considers the first two of the four solutions for $c_\text{MLE}$. This is because the zeros obtained by profiling the score as a function of $c_\text{MLE}^{(1)}$ are identical to those when using $c_\text{MLE}^{(3)}$, and likewise for $c_\text{MLE}^{(2)}$ and $c_\text{MLE}^{(4)}$. Skipping the final two solutions therefore achieves a factor 2 improvement in execution time with no degradation in performance.

### 5.2.9 An example

Consider an example defined by

$$
\begin{aligned}
\bar{n} &= \{\; 2.0, \quad 1.6,\; 1.2,\; 0.0,\; 1.2,\; 1.6,\; 2.0\}, \\
\bar{l} &= \{-0.4,\; -0.2, \quad 0,\; 0.2,\; 0.6,\; 1.0,\; 1.2\},
\end{aligned}
\tag{89}
$$

with $\bar{N} = 4.00$, $\bar{L} = 1.74$ and $\rho = 0.92$. Fig 11 compares our solution with the distribution of psuedo-experiments and the result of assuming Wilks' theorem. We see that Wilks' theorem is strongly violated, whereas our solution agrees well with the toys. The bottom panel shows the expected coverage of a 95% confidence interval. Our solution falls close to the ideal line at 95% and significantly outperforms Wilks. The small deviation from 95% is likely due to the precision associated with numerically integrating the sharp peak in the PDF when $q_{c_\text{true}} \sim 0$. We notice that Wilks' theorem tended to over-cover in all the other cases studied. However, here we observe it to significantly **under-cover** for many of the hypotheses tested. This is a significant concern, as it could lead to exaggerated claims of significance when excluding hypotheses.

## 6 Summary & Conclusion

The main purpose of this work is to highlight that Wilks' theorem is not satisfied in an EFT fit which contains significant contributions from the quadratic component. In general, calculating $p$-values assuming Wilks' theorem is seen to cause significant over-coverage close to the Standard Model hypothesis. However, we also observe worrying cases of *under-coverage* when the linear and quadratic components both contribute. We explain why Wilks' theorem

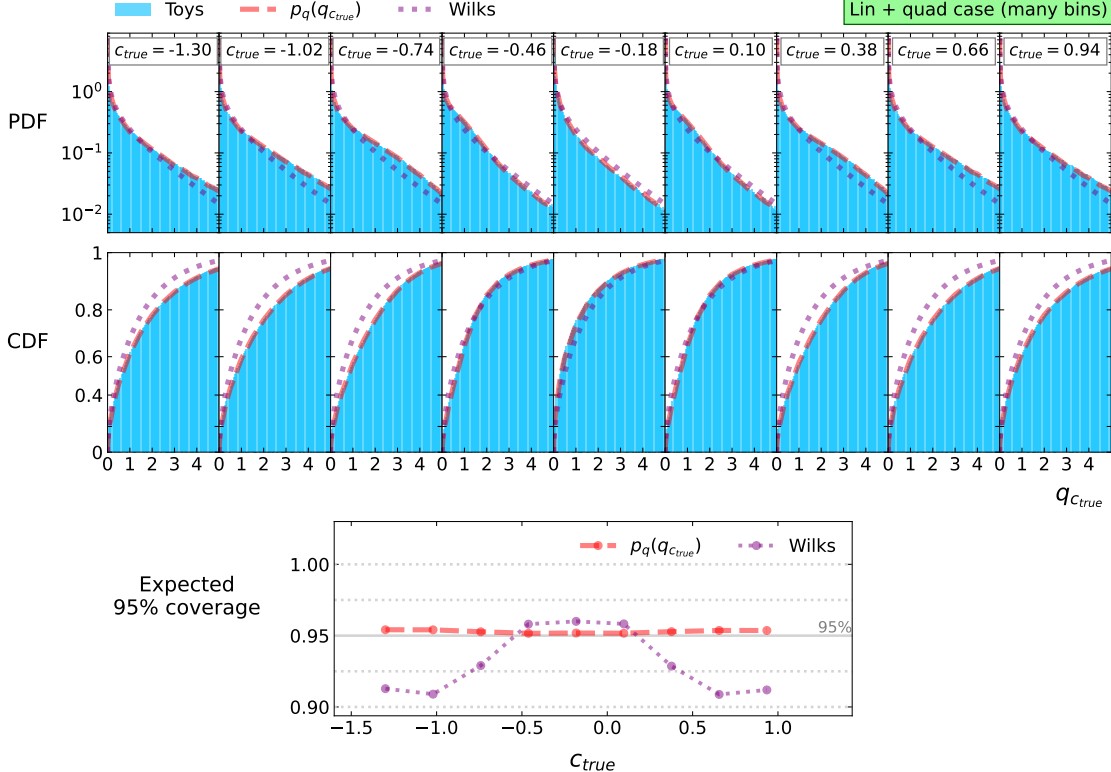

Figure 11: Distributions of the PDF and CDF along with the expected coverage for the multi-bin "linear plus quadratic" example defined by Eq 89.

is violated in these cases. It is because we break one of its key conditions: when profiling the Wilson coefficients we reach a boundary of parameter space, whereas Wilks' theorem assumes that we stay within a continuous bulk.

We derive the correct form of the asymptotic distribution for $q_{c_{\text{true}}}$ when profiling up to two Wilson coefficients, each of which modulating purely linear or quadratic contributions. These distributions are defined by parameters which may be obtained experimentally by scanning the likelihood profile of an Asimov dataset. We speculate that this Asimov approach may help analysers to obtain an approximate distribution in non-idealised circumstances, for example in the presence of non-Gaussian nuisance parameters. We also provide the correct distribution for a "linear plus quadratic" model constrained by a single-bin measurement.

We then discover that the general "linear plus quadratic" case cannot be easily parameterised, because the EFT contribution is found to change direction as we profile the Wilson coefficient. We successfully solve this distribution, but are forced to include a numerical step which is computationally-inefficient compared with the other cases studied.

We note the following opportunities for future work:

- It would be highly desirable to derive the asymptotic distribution of $q_{c_{\text{true}}}$ when profiling arbitrary numbers of EFT parameters.

- We can study how the two-parameter solutions are affected when we introduce non-zero interference terms $t_{i,j}$ between the contributions of EFT coefficients $c_i$ and $c_j$. This introduces a third basis vector and an additional parametric dependence which makes it more complicated to find the global minimum of the $\chi^2$-profile.

- We can study how the solutions are impacted when we introduce non-Gaussian nuisance

parameters. We expect that profiling these will impact the covariance in a nonlinear way, and we do not predict how this will impact the distribution of $q_{c_{\text{true}}}$.

- In the "linear plus quadratic" case, computational performance may be improved by finding a fully analytic solution to supersede our partly numerical one.

- We can study the "quadratic plus quartic" case, which may arise when constraining mass-dimension-8 models. This differs from the "linear plus quadratic" case by preventing the linear component from providing negative contributions.

## Acknowledgements

**Funding information**   We thank Nicolas Berger for insightful discussions and comments on the manuscript. Florian Bernlochner is supported by DFG Emmy-Noether Grant No. BE 6075/1-1. Stephen Menary is supported through a grant from the Alan Turing Institute, London, UK, and through the Science and Technology Facilities Council grant ST/N000374/1.

## A   Visualising the linear x quadratic integrals

In Section 4.2 we derived the density $p_q(q_{c_{\text{true}}})$ for the two-parameter case where $c_1$ modulates a linear EFT model and $c_2$ modulates a quadratic one. We did this using the change-of-variables method with respect to the two independent normally distributed random variables $\hat{\tilde{Z}}_1$ and $\hat{\tilde{Z}}_\perp$. In this appendix we visualise this calculation. All plots are made using the example defined Section 4.2.1 with a test hypothesis of $c_{\text{true}} = (0.7, -0.5)$.

The grey-scale histogram in Fig 12 shows the distribution of toys in the $(\hat{\tilde{Z}}_1, \hat{\tilde{Z}}_2)$-plane (left-hand panel) and $(\hat{\tilde{Z}}_1, \hat{\tilde{Z}}_\perp)$-plane (right-hand panel). Correlation coefficients are displayed in the top left corner. We can see that the random variables $\hat{\tilde{Z}}_1$ and $\hat{\tilde{Z}}_\perp$ are indeed uncorrelated. Furthermore, the planes are shaded red for events classified as class A, and blue for class B. The class boundary falls at $\hat{\tilde{Z}}_\perp = -\hat{\tilde{Z}}_{\perp,0}$.

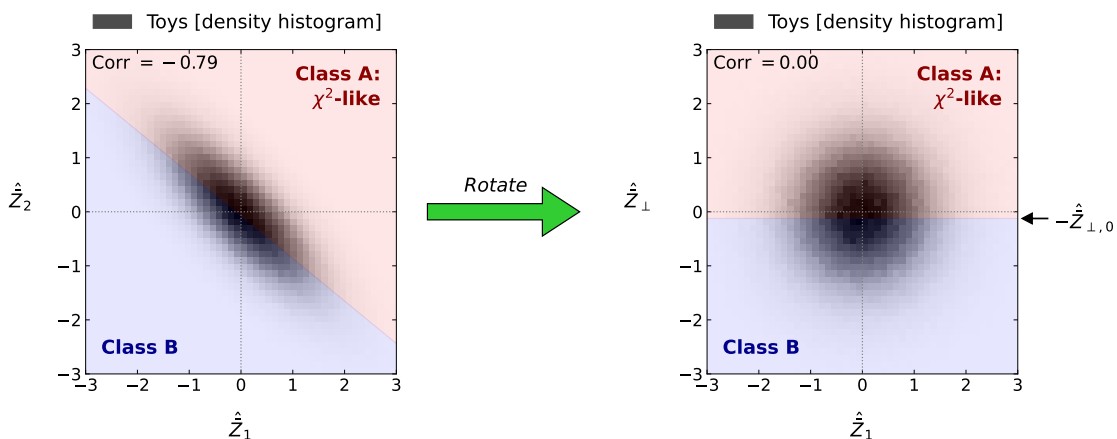

Figure 12: Distribution of events in the $(\hat{\tilde{Z}}_1, \hat{\tilde{Z}}_2)$- and $(\hat{\tilde{Z}}_1, \hat{\tilde{Z}}_\perp)$-planes.

Since we are inverting a $2 \rightarrow 1$ transformation, there is a contour of possible $(\hat{\tilde{Z}}_1, \hat{\tilde{Z}}_\perp)$-observations which lead to any given $q_{c_{\text{true}}}$. To visualise this, Fig 13 shows the contours for four different values of $q_{c_{\text{true}}}$. These form closed loops of increasing radius as $q_{c_{\text{true}}}$ increases. To

evaluate $p_q$ for a given $q_{c_{\text{true}}}$, we must integrate over the densities of the observations spanned by the corresponding contour. At each point, the form of the integrand depends on whether it is in a class A or B region. This leads to the expression of Eq 38.

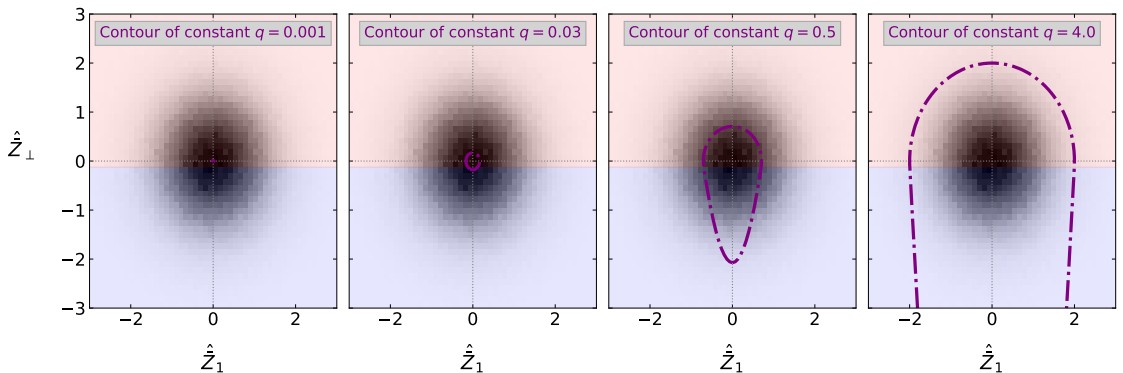

Figure 13: $(\hat{\hat{Z}}_1, \hat{\hat{Z}}_\perp)$-contours which lead to four different values of $q \equiv q_{c_{\text{true}}}$. Fig 14 shows the integrands when we integrate the density along each contour.

Fig 14 shows the resulting integrand as a function of the integration parameter $\hat{\hat{Z}}_\perp$. Class A contributions are shown in red and class B in blue. Construction lines show how the class boundary limits are identified. Since $\hat{\hat{Z}}_{\perp,0} = 0.12$ in this example, the first panel with $q_{c_{\text{true}}} = 0.001$ contains only contributions from class A events. As shown by Eq 40, we find that $q_{c_{\text{true}}}$ is $\chi^2$-distributed with two degrees of freedom in this regime. This is because the parameter $c_2$ does not encounter its boundary during the optimisation step. The second panel with $q_{c_{\text{true}}} = 0.03$ only just transitions into the class B region and the shape of the integrand looks quite similar. As $q_{c_{\text{true}}}$ increases further, the two solutions of $\hat{\hat{Z}}_1^{(B)}$ tend towards constant values and so the blue integrand contribution tends towards a Gaussian distribution.

We observe that the integrand tends to diverge at the domain boundaries. This is because the contours shown in Fig 13 run perpendicular to the $\hat{\hat{Z}}_\perp$-axis. We note that a more efficient numerical integration may be performed by forming an integrand that does not contain such singularities. This may be achieved by choosing a different integration parameter, such as the polar angle from the origin, but we do not consider this optimisation in this work.

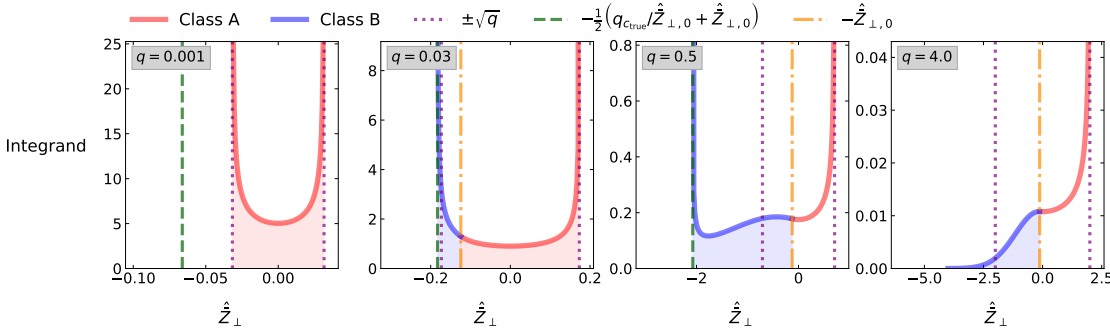

Figure 14: Integrands used to compute $p_q\left(q_{c_{\text{true}}}\right)$ for four different values of $q_{c_{\text{true}}}$. These correspond to density integrals over the contours shown in Fig 13.

# B Visualising the quadratic x quadratic integrals

In Section 4.3 we derived the density $p_q(q_{c_{\text{true}}})$ for the two-parameter case where $c_1$ and $c_2$ both modulate quadratic EFT models. In this appendix we visualise the PDF integrals in the same way as in Appendix A. All plots are made using the example defined Section 4.3.1.

Suppose that $c_{\text{true}} = (0.7, 0.5)$. Fig 15 shows how the $(\hat{\hat{Z}}_1, \hat{\hat{Z}}_\perp)$-plane is segmented into the four different classes of event. Four panels are shown, each showing a contour corresponding to a different value of $q \equiv q_{c_{\text{true}}}$. Once again these are closed loops around the origin with a radius that increases with $q_{c_{\text{true}}}$.

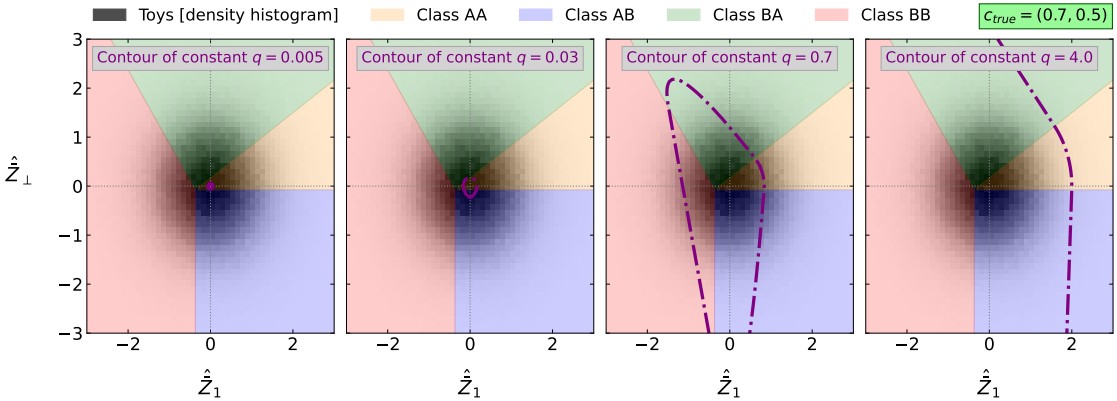

Figure 15: $(\hat{\hat{Z}}_1, \hat{\hat{Z}}_\perp)$-contours which lead to four different values of $q \equiv q_{c_{\text{true}}}$. Fig 16 shows the integrands when we integrate the density along each contour.

Fig 16 shows the integrand we obtain when using $\hat{\hat{Z}}_\perp$ as the integration parameter. This is a stacked plot where the individual contribution from event class X is labelled $f_X^\pm$, and $\pm$ indicates whether we take the solution with a positive or negative square-root where applicable. The density and region-boundaries can be understood by viewing the corresponding contour of Fig 15 and projecting onto the $\hat{\hat{Z}}_\perp$-axis.

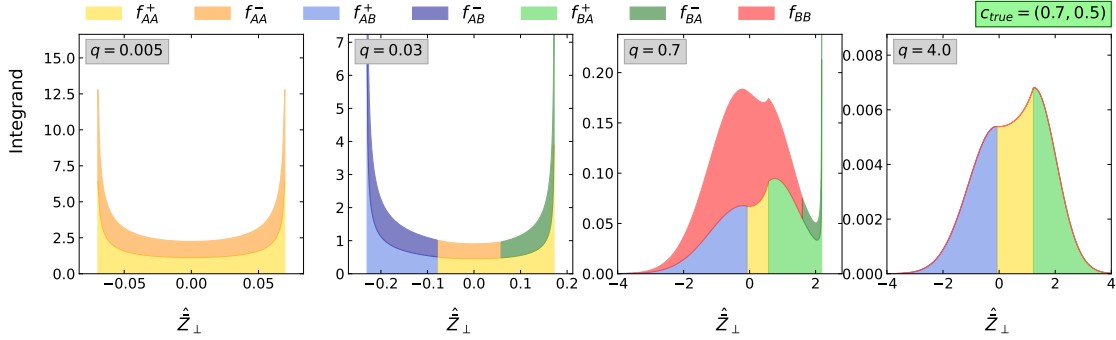

Figure 16: Integrands used to compute $p_q\left(q_{c_{\text{true}}}\right)$ for four different values of $q \equiv q_{c_{\text{true}}}$, showing stacked contributions from the different classes. These correspond to density integrals over the contours shown in Fig 15.

When $q_{c_{\text{true}}}$ is small, corresponding to the left-most panel, the whole contour is contained within the AA region. For these events we expect $q_{c_{\text{true}}}$ to follow a $\chi^2$-distribution with two degrees of freedom. We see that the integrand diverges at the domain boundaries because the contour runs parallel to the $\hat{\hat{Z}}_\perp$ axis. As $q_{c_{\text{true}}}$ increases, we move to the second and third

panels, and observe contributions from class AB, BA and BB events. In the final panel $q_{c_{\text{true}}}$ is large enough that the contour does not close over the domain shown, and the integrand appears to smoothly approach 0.

Fig 17 shows the $q_{c_{\text{true}}}$-contours in the special cases when $c_{\text{t},1} = 0$ and/or $c_{\text{t},2} = 0$. We observe that the contours do not close but rather extend to infinity. In the limit $q_{c_{\text{true}}} \to 0$, the contour converges to a *line* instead of a single point. This causes a singularity in $p_q(q_{c_{\text{true}}} = 0)$, which is observed in Fig 6.



Figure 17: Contours $q \equiv q_{c_{\text{true}}}$ for different $c_{\text{true}}$ hypotheses with $c_{\text{t},1} = 0$ and/or $c_{\text{t},2} = 0$.



# C  Geometric derivation of the one-parameter results

In this appendix we use a geometric perspective to derive the results of the three one-parameter cases. In doing so we answer the question: why is the asymptotic distribution of $q_{c_{\text{true}}}$ harder to find in the "linear plus quadratic" example?

Consider a two-bin example with $\bar{l} = (0.9, \ 1.1)$ and $\bar{n} = (1.5, \ 0.9)$. Using our de-correlated co-ordinate system, the measurement is represented by the two independent normally distributed random variables $\bar{z} \equiv (\bar{z}_1, \bar{z}_2)$. Since the $\chi^2$ is calculated as

$$\chi^2(\bar{z}; c) \ = \ \left| \bar{z} - \bar{k}(c) \right|^2 , \tag{C.1}$$

we can see that the predictor function $\bar{k}(c)$ may be represented as a vector on the same space as $\bar{z}$. As we vary the Wilson coefficient $c$, the predictor function $\bar{k}(c)$ sweeps out a parametric curve on this space. This is shown visually in Fig 18 for a value of $c_{\text{true}} = 0.5$.

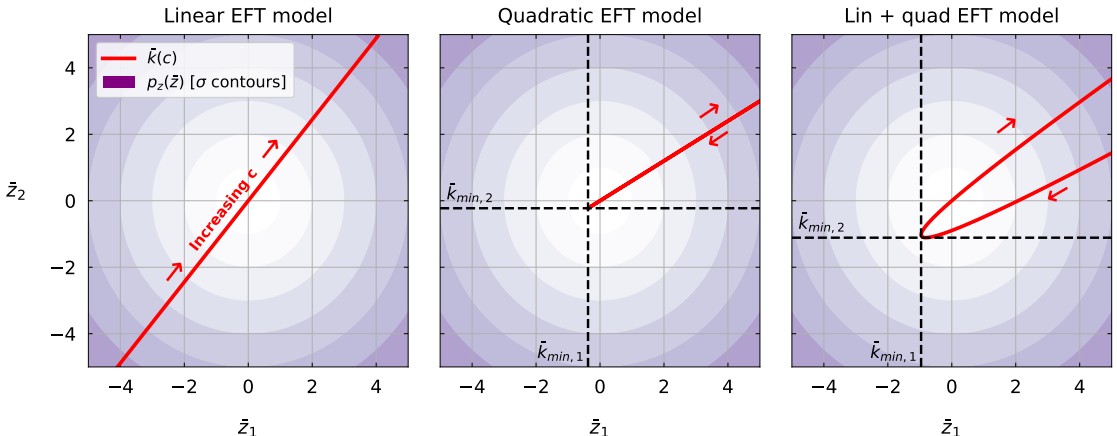

Figure 18: Visualisation of $\bar{k}(c)$ in a 2-dimensional $\bar{z}$-space.

Concentric purple filled contours represent the PDF $p_{\bar{z}}(\bar{z})$, measured in increments of the standard deviation $\sigma = 1$. The density is highest at the centre of the circle, and decreases as we move outwards. The thick red line represents the curve $\bar{k}(c)$, and red arrows show how this curve is profiled as $c$ is increased from $-\infty$ to $+\infty$. Three panels separately show linear, quadratic and linear + quadratic examples. We see that the shape of the curve depends on the type of EFT model being considered. We can understand these shapes in the following way:

**Linear case:** $\bar{k}(c) = (c - c_{\text{true}})\bar{l}$ and so $\bar{k}(c)$ is a linear parametric curve parallel to $\bar{l}$. There are no values of $\bar{z}_1$ or $\bar{z}_2$ which cannot be accessed.

**Quadratic case:** $\bar{k}(c) = (c^2 - c_{\text{true}}^2)\bar{n}$ is also linear, since it always runs parallel to $\bar{n}$ for any value of $c$. However, this curve is finite in extent. Since $\bar{k}(\pm c)$ are degenerate, the curve must 'turn around' at a value of $\bar{k}(c = 0) = (\bar{k}_{\text{min},1}, \bar{k}_{\text{min},2})$. Any values of $\bar{z}_1$ or $\bar{z}_2$ beyond this cannot be accessed. The 'turn around' point is only equal to the origin if $c_{\text{true}} = 0$.

**Lin+quad case:** $\bar{k}(c) = (c - c_{\text{true}})\bar{l} + (c^2 - c_{\text{true}}^2)\bar{n}$ describes a convex curve which is parallel to $\bar{l}$ when $c = c_{\text{true}}$ and parallel to $\bar{n}$ when $c = \pm\infty$. Limiting values $\bar{k}_{\text{min},1}$ and $\bar{k}_{\text{min},2}$ are obtained at different values $c$.

Say that we make a measurement, meaning that we sample $\bar{z}' \sim p_{\bar{z}}(\bar{z})$, and observe the event shown in Fig 19.

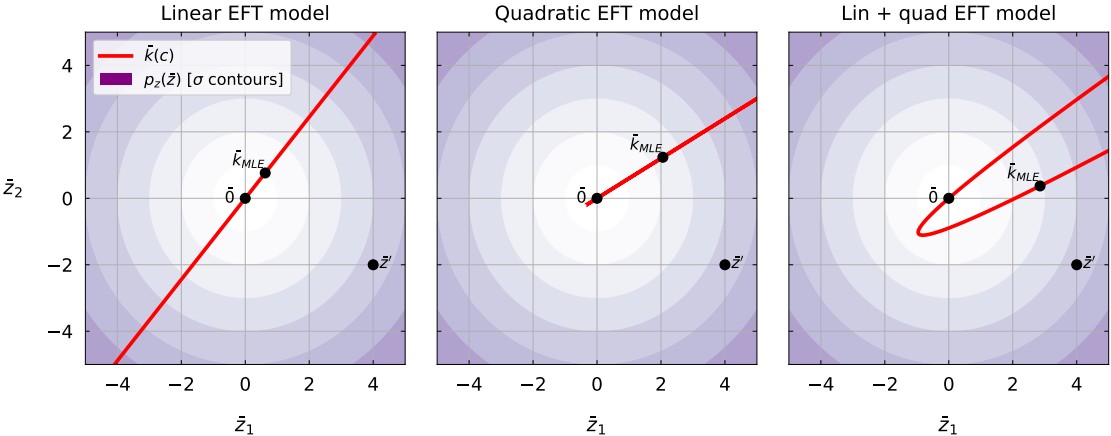

Figure 19: Visualisation of the points of interest for a candidate observation $\bar{z}'$.

We identify three points of interest: the observation $\bar{z}'$, the origin $\bar{0}$, and the maximum likelihood estimate $\bar{k}_{\mathrm{MLE}}$. We note that $\bar{0}$ must always lie on the curve because it is the solution to $\bar{k}(c = c_{\mathrm{true}})$. The first key insight is the realisation that **we can calculate $q_{c_{\mathrm{true}}}$ using the geometric relationships between these points**. This is because $\chi^2(\bar{z}'; c) = |\bar{z}' - \bar{k}(c)|^2$, and so we can interpret $\chi^2(\bar{z}'; c)$ as the Euclidean distance-squared between $\bar{z}'$ and $\bar{k}(c)$. Therefore

$$
\begin{aligned}
q_{c_{\mathrm{true}}} &= \chi^2(\bar{z}'; c_{\mathrm{true}}) - \chi^2(\bar{z}'; c_{\mathrm{MLE}}) \\
&:= \chi^2_{c_{\mathrm{true}}} - \chi^2_{c_{\mathrm{MLE}}},
\end{aligned}
\tag{C.2}
$$

can be computed using the lengths of the lines connecting $\bar{0}$, $\bar{z}'$ and $\bar{k}_{\mathrm{MLE}}$. This is shown in Fig 20.

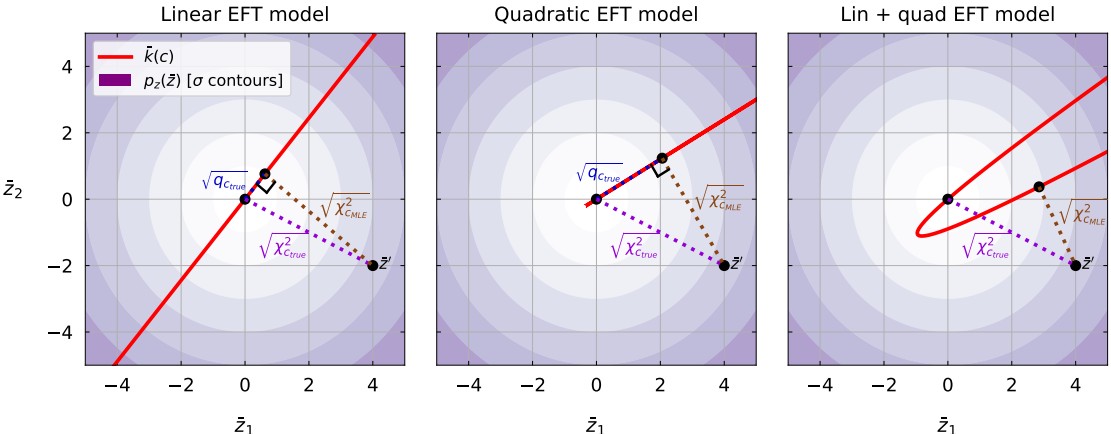

Figure 20: Visualisation showing how $q_{c_{\mathrm{true}}}$ is calculated for a candidate observation $\bar{z}'$. The test-statistic behaves like a $\chi^2$ random variable for linear and quadratic models.

The next key insight is that **the line labelled $\sqrt{\chi^2_{c_{\mathrm{MLE}}}}$ must be tangent to the curve at $\bar{k}_{\mathrm{MLE}}$**, since $\bar{k}_{\mathrm{MLE}}$ can be thought of as the point of "minimum distance of approach" between $\bar{k}(c)$ and $\bar{z}'$. **From this fact alone we can infer whether $q_{c_{\mathrm{true}}}$ follows a $\chi^2$-distribution**. To show how this is done, once again we will consider the three cases in turn:

**Linear case:** The curve is just a straight line. This means that, since the line $\sqrt{\chi^2_{c_{\text{MLE}}}}$ is tangent to $\bar{k}(c)$ at a single point $\bar{k}_{\text{MLE}}$ it must also be tangent to the whole curve. This means that **the points $\{\bar{0}, \bar{z}', \bar{k}_{\text{MLE}}\}$ define a right-angled triangle**, and Equation C.2 is equivalent to Pythagoras' theorem, where $q_{c_{\text{true}}}$ is the length-squared of the base. Since we know that the curve is parallel to $\bar{l}$, this means that $q_{c_{\text{true}}} = (\bar{z}' \cdot \hat{\bar{l}})^2$. Since $\bar{z}'$ is drawn from a two-dimensional normal distribution, we see that $q_{c_{\text{true}}}$ must be distributed as a $\chi^2$-distribution with one degree of freedom. Thus we have reproduced the result of section 3.

**Quadratic case:** In Fig 20, we see that $\bar{z}'$ falls in the section of the plane perpendicular to $\bar{k}(c)$. Just like the in the linear case, we see that the points $\{\bar{0}, \bar{z}', \bar{k}_{\text{MLE}}\}$ define a right-angled triangle. The same argument tells us that $q_{c_{\text{true}}} = \left(\bar{z}' \cdot \hat{\bar{n}}\right)^2$ and must follow a $\chi^2$-distribution with one degree of freedom.

**Lin+quad case:** The points $\left(\bar{O}, \bar{z}', \bar{k}_{\text{MLE}}\right)$ no longer form a right-angled triangle. Since the curve is not straight, the internal angle changes with every $\bar{z}'$, and we cannot easily infer the distribution of $q_{c_{\text{true}}}$. However, we can see that when $\bar{k}_{\text{MLE}}$ is far from the turning point, the internal angle approaches a right-angle and so the distribution of $q_{c_{\text{true}}}$ must approach a $\chi^2$-distribution. As $\bar{k}_{\text{MLE}}$ approaches the turning point, the deviation from a $\chi^2$-distribution will increase in a smooth way.

Let us now move $\bar{z}'$ into a new position which is beyond the reach of the quadratic model. This new example is shown in Fig 21.

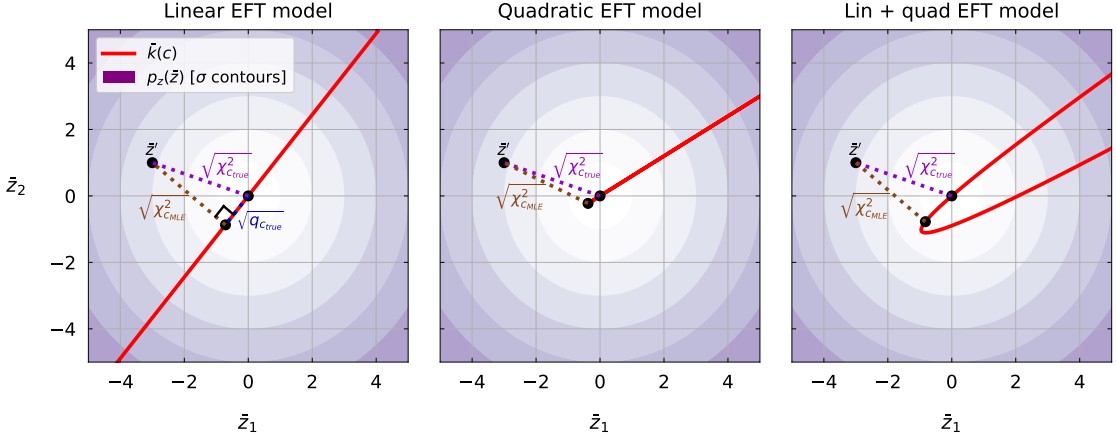

Figure 21: Visualisation showing how $q_{c_{\text{true}}}$ is calculated for another candidate observation $\bar{z}'$. The test-statistic behaves like a Gaussian random variable for the quadratic model.

Let us again consider the three cases in turn:

**Linear case:** The points $\left(\bar{O}, \bar{z}', \bar{k}_{\text{MLE}}\right)$ will always form a right-angled triangle and $q_{c_{\text{true}}}$ will always follow a $\chi^2$-distribution.

**Quadratic case:** The points $\left(\bar{O}, \bar{z}', \bar{k}_{\mathrm{MLE}}\right)$ now form a *scalene* triangle. If $d_{A,B}$ denotes the vector between points $A$ and $B$ then we use the triangle identity $\left|d_{A,B}\right|^2 = \left|d_{A,C}\right|^2 + \left|d_{B,C}\right|^2 - 2|d_{A,C} \cdot d_{B,C}|$ to write

$$\chi^2_{c_{\mathrm{MLE}}} = \chi^2_{c_{\mathrm{true}}} + \left|\bar{k}_{\mathrm{MLE}}\right|^2 - 2\left(\bar{z}' \cdot \bar{k}_{\mathrm{MLE}}\right), \qquad (C.3)$$

with $\chi^2_{c_{\mathrm{true}}} = \left|\bar{z}'\right|^2$. For all events in this category, we know that $\bar{k}_{\mathrm{MLE}} = \bar{k}\left(c = 0\right) = -c_{\mathrm{true}}\bar{n}$. It then follows that

$$q_{c_{\mathrm{true}}} = \chi^2_{c_{\mathrm{true}}} - \chi^2_{c_{\mathrm{MLE}}} = -2c_{\mathrm{true}}\left(\bar{z}' \cdot \bar{n}\right) - c_{\mathrm{true}}^2 |\bar{n}|^2. \qquad (C.4)$$

Since $\bar{n}$ is a constant vector and $\bar{z}'$ is drawn from a normal distribution, we see that $q_{c_{\mathrm{true}}}$ must follow a Gaussian distribution.

**Lin+quad case:** The curvature of $\bar{k}(c)$ means that no single point $\bar{k}_{\mathrm{MLE}}$ is the same solution for many $\bar{z}'$. However, when the curvature is abrupt, many $\bar{k}_{\mathrm{MLE}}$ may be found in a small region close to the turning point, leading to behaviour which approaches Gaussian for very sharp curvatures. The behaviour smoothly moves away from Gaussian as $\bar{k}_{\mathrm{MLE}}$ evolves away from the turning point.

By combining these two case studies we form a geometric understanding of why the three classes of model lead to different distributions of $q_{c_{\mathrm{true}}}$. In the linear case, $\bar{k}(c)$ extends across the whole plane and there are no possible placements of $\bar{z}'$ for which the points $\left(\bar{O}, \bar{z}', \bar{k}_{\mathrm{MLE}}\right)$ will not form a right-angled triangle. **Therefore $q_{c_{\mathrm{true}}}$ always follows a $\chi^2$-distribution**. In the quadratic case, $q_{c_{\mathrm{true}}}$ follows a $\chi^2$-distribution only when $\bar{z}'$ falls in the region perpendicular to $\bar{k}(c)$. Otherwise the remaining infinitely-wide region in $\bar{z}$-space results in a single value of $\bar{k}_{\mathrm{MLE}}$, from which we find that $q_{c_{\mathrm{true}}}$ is Gaussian distributed. **Therefore $q_{c_{\mathrm{true}}}$ follows a mixture of a $\chi^2$-distribution with a Gaussian distribution**.

In the "linear + quadratic" case, the curvature of $\bar{k}(c)$ means that no right-angled triangles or degenerate $\bar{k}_{\mathrm{MLE}}$ may be identified. The distribution of $q_{c_{\mathrm{true}}}$ only contains approximately $\chi^2$- and Gaussian components corresponding to limiting regions of $\bar{z}$-space, with most points representing a smooth transition between the two. **Therefore it is in general not possible to decompose this case into a mixture of $\chi^2$- and Gaussian distributions**.

For completeness, Fig 22 visualises the contours of constant $q_{c_{\mathrm{true}}}$ in our plane. In the linear case, they are always perpendicular to the line $\bar{k}(c)$, since this runs parallel to $\bar{n}$ and $q_{c_{\mathrm{true}}} \sim (\bar{z} \cdot \bar{n})^2$ is only sensitive to the projection $\bar{z} \cdot \bar{n}$. In the quadratic case, the contours are less densely spaced in the Gaussian region because $q_{c_{\mathrm{true}}} \sim \bar{z} \cdot \bar{n}$ rather than $q_{c_{\mathrm{true}}} \sim (\bar{z} \cdot \bar{n})^2$. In the "linear + quadratic" case, contours are curved and only appear Gaussian or $\chi^2$-like in limiting regions.

In this section we used a two-dimensional $\bar{z}$-space to help with visualisation. However, **our arguments generalise to arbitrary dimensions**. This is because the problem is defined by the two basis vectors $\bar{l}$ and $\bar{n}$. These always define a two-dimensional plane, and increasing the dimensionality of $\bar{z}$ simply embeds this plane in a higher dimensional space (in which contours are promoted to hyper-surfaces).

**Bonus: explaining why the $c_{\mathrm{MLE}}$ distribution has three modes:**

We can use our geometric picture to explain the shape of the $c_{\mathrm{MLE}}$ distribution. Let us view our curve $\bar{k}(c)$ for four different values of $c_{\mathrm{true}}$. These are shown in the top row of Fig 23.

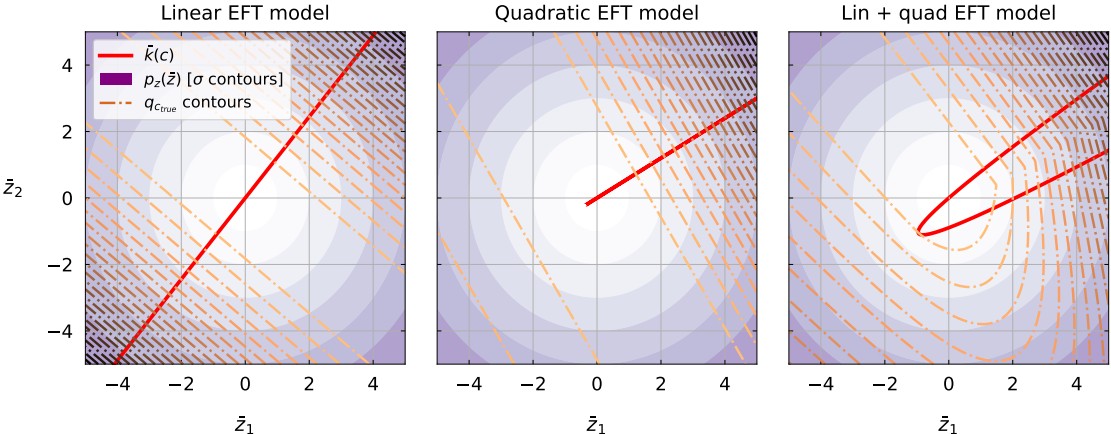

Figure 22: Visualisation showing how contours of the test-statistic $q_{c_{\text{true}}}$ behave differently in the linear, quadratic and 'linear + quadratic" EFT models.

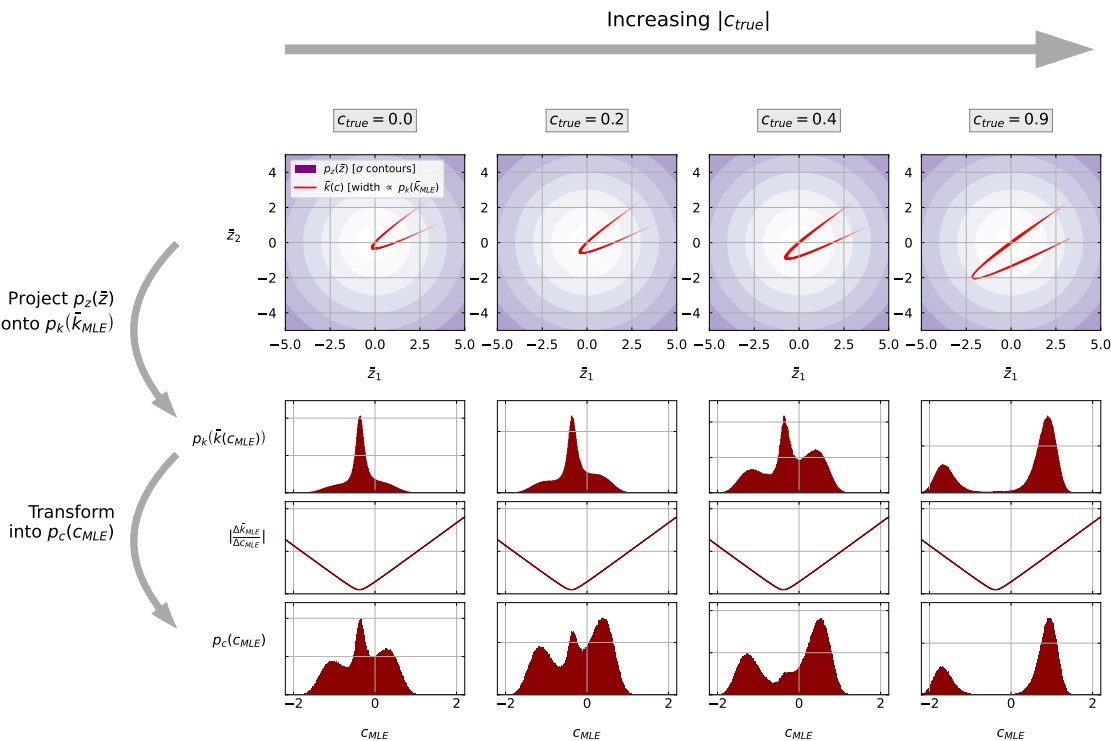

Figure 23: Moving vertically from top to bottom, we explain why the distribution of $c_{\text{MLE}}$ has three peaks when considering "linear + quadratic" EFT models. Moving horizontally from left to right, we explain why the middle peak vanishes as $|c_{\text{true}}|$ increases.

The $\bar{k}(c)$ curve is presented with a width that represents the density $p_k$ of $\bar{k}_{\text{MLE}}$ at each point. This density is also shown parametrically as a function of $c_{\text{MLE}}$ as the uppermost of the three histogram panels. We see that there are up to three peaks in this distribution: a central-peak of $\bar{k}_{\text{MLE}}$ values concentrated on the curve's turning-point, and two side-peaks for $\bar{k}_{\text{MLE}}$ concentrated on its arms. When the curve turns at a point close to the origin, as in the first three columns, much of the density corresponds to a $\bar{k}_{\text{MLE}}$ close to the turning-point. This means that the central peak is very large. As $|c_{\text{true}}|$ increases, the turning-point moves away from the high-density region of the $\bar{z}$-plane and so the amplitude of this central peak decreases.

Instead it becomes more likely that we will observe a $\bar{k}_{\text{MLE}}$ on one of the two arms. This is observed as the increase in amplitude and eventual domination of the two 'side peaks' in the $p_k(\bar{k}_{\text{MLE}})$ distribution of Fig 23.

To understand how this relates to the distribution of $p_c(c_{\text{MLE}})$, we may transform the probability densities according to

$$p_c(c_{\text{MLE}}) \; = \; p_k(k(c_{\text{MLE}})) \; \cdot \; \left|\frac{\mathrm{d}k}{\mathrm{d}c}\right|_{c=c_{\text{MLE}}} , \tag{C.5}$$

where $\left|\frac{\mathrm{d}k}{\mathrm{d}c}\right|$ is the Jacobian factor describing how the density of an infinitesimal volume is modified by the transformation between spaces. This is plotted in the middle red panel. This factor appears to enhance the density of $c_{\text{MLE}}$ far from the centre of the distribution, reflecting the fact that small changes in $c$ tend to cause larger changes in $\bar{k}$ when far from the turning-point (this is because the $c^2$ term dominates in this region).

The resulting distribution of $p_c(c_{\text{MLE}})$ is shown in the bottom panel of Fig 23. The three-peak structure of $p_k(k(c_{\text{MLE}}))$ has interacted with a Jacobian factor enhancing contributions far from the centre to cause a prominent three-peak structure in $p_c(c_{\text{MLE}})$. We now understand that this will tend towards a two-peak structure when the curvature of $\bar{k}(c)$ is not very sharp, since the $\bar{k}_{\text{MLE}}$ around the turning-point will be diluted over a wider interval, or when the turning-point is far from the high-density region of the $\bar{z}$-plane, for example when $|c_{\text{true}}|$ is large.

# D  Additional details for the "linear plus quadratic" case

This appendix provides some additional details for the "linear plus quadratic" multi-bin case. Here we will visualise the distribution of each latent variable as we progress through the forwards transformation to model $q_{c_{\text{true}}}$. For this purpose, let us consider the example defined by Eq 89 with an arbitrary test hypothesis of $c_{\text{true}} = -0.5$.

**PDF of $\Gamma$**

Fig 24(top panel) shows the distribution of $\Gamma = \Gamma_r + i\Gamma_i$ estimated using pseudo-experiments ("Toys"). In section 5.2.3, we found that the requirement of real $\Delta_1$ causes three different modes to appear in the distribution of $\Gamma$, which we labelled Types A-C. These are clearly visible, with Type A events falling along the axis $\Gamma_r = 0$, Type B events falling along the line $3\Gamma_r^2 = \Gamma_i^2$, and Type C events dispersed in the angle between. The three bottom panels show the distributions for each individual mode, where Type A and B events have only one degree of freedom whereas Type C events have two.

We can derive the PDFs for these distributions. For Type A and B events, we must integrate over the spare degree of freedom. We write

$$\tilde{p}_\Gamma^{(A) \text{ or } (B)}(\Gamma) \; = \; \int_{-\infty}^{+\infty} \left|\frac{\mathrm{d}\Delta_1}{\mathrm{d}\Gamma}\right| \, p_{\Delta_1}(\Delta_1) \, p_{\Delta_0}(\Delta_0) \, \mathrm{d}\Delta_0 , \tag{D.1}$$

where $\Delta_1$ is obtained using the solutions in section 5.2.3 and is easily differentiated with respect to $\Gamma$. Since each Type describes only a fraction of the total density, this PDF is not normalised to unity, and we use $\tilde{p}$ to denote the un-normalised PDF. We do not include $\Delta_0$ in the Jacobian factor because it is externally selected by the integration and not the result of transforming $\Gamma$, and there is no summation over $\Delta_1$ because we find a maximum of one $\Delta_1$

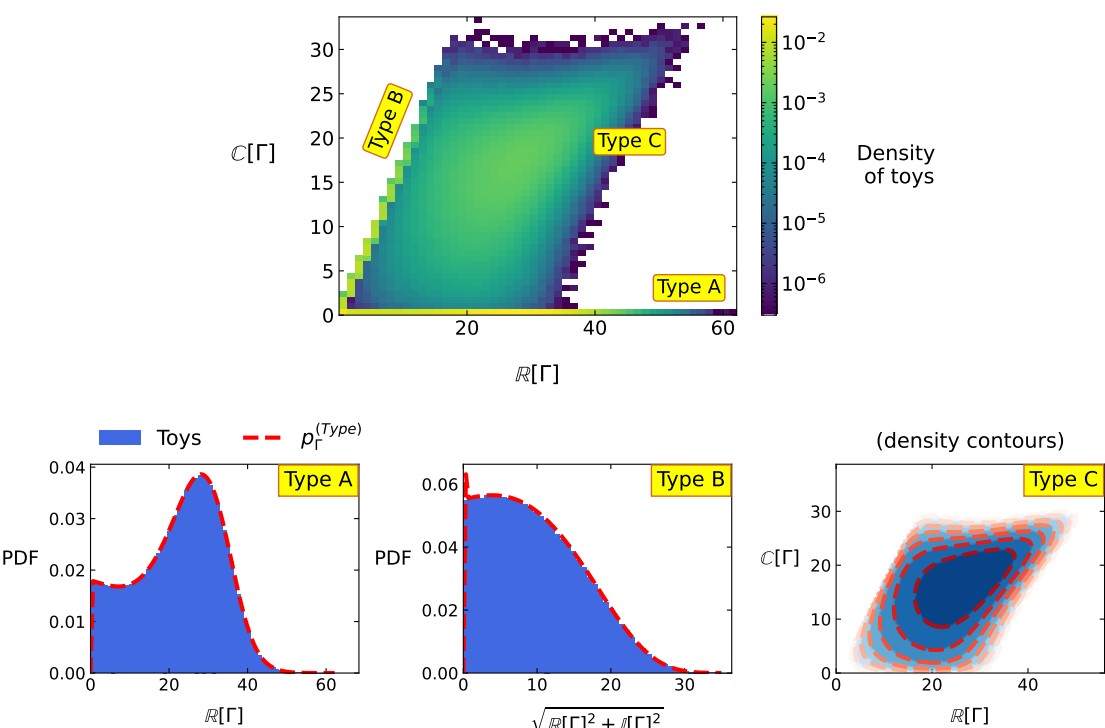

Figure 24: Top: density of $\Gamma$ in the complex plane estimated using toys. Three different modes are visible and designated types A, B and C. Bottom: comparison between the the toys (blue histogram) and evaluated PDF (red line) for the three modes.

solution for any given $(\Gamma, \Delta_0)$-pair. In Fig 24(bottom left and middle) we see that these formulae agree well with the toys. The integral is estimated numerically using Gaussian quadrature summation between limits of $\mu_{\Delta_0} \pm 5\sigma_{\Delta_0}$.

For Type C events we have a $2 \rightarrow 2$ transformation where the Jacobian factor is defined as the determinant of the Jacobian matrix

$$|\mathrm{Jac}(\Delta_0, \Delta_1; \Gamma)| = \begin{vmatrix} \frac{\partial \Delta_0}{\partial \Gamma_r} & \frac{\partial \Delta_0}{\partial \Gamma_i} \\ \frac{\partial \Delta_1}{\partial \Gamma_r} & \frac{\partial \Delta_1}{\partial \Gamma_i} \end{vmatrix}, \tag{D.2}$$

where for Type C events we have $\Delta_0 = \Gamma_r^2 + \Gamma_i^2$ and so

$$\frac{\partial \Delta_0}{\partial \Gamma_r} = 2\Gamma_r, \qquad \frac{\partial \Delta_0}{\partial \Gamma_i} = 2\Gamma_i. \tag{D.3}$$

By differentiating Eq 76 we find

$$\frac{\partial \Gamma}{\partial \Delta_1} = \frac{\Gamma}{3\left(\Delta_1 + \sqrt{\Delta_1^2 - 4\Delta_0^3}\right)}\left(1 + \frac{\Delta_1}{\sqrt{\Delta_1^2 + 4\Delta_0^3}}\right), \tag{D.4}$$

and so

$$\frac{\partial \Delta_1}{\partial \Gamma_r} = \left(\frac{\partial \Gamma_r}{\partial \Delta_1}\right)^{-1} = \left(\mathrm{Re}\left[\frac{\partial \Gamma}{\partial \Delta_1}\right]\right)^{-1}, \qquad \frac{\partial \Delta_1}{\partial \Gamma_i} = \left(\frac{\partial \Gamma_i}{\partial \Delta_1}\right)^{-1} = \left(\mathrm{Im}\left[\frac{\partial \Gamma}{\partial \Delta_1}\right]\right)^{-1}. \tag{D.5}$$

The un-normalised PDF is then

$$\tilde{p}_\Gamma^{(C)}(\Gamma) = |\mathrm{Jac}(\Delta_0, \Delta_1; \Gamma)|\, p_{\Delta_1}(\Delta_1)\, p_{\Delta_0}(\Delta_0). \tag{D.6}$$

Fig 24(bottom right) shows contours of density for Type C events evaluated using toys (filled blue) and using our formula (red lines). Good agreement is observed,

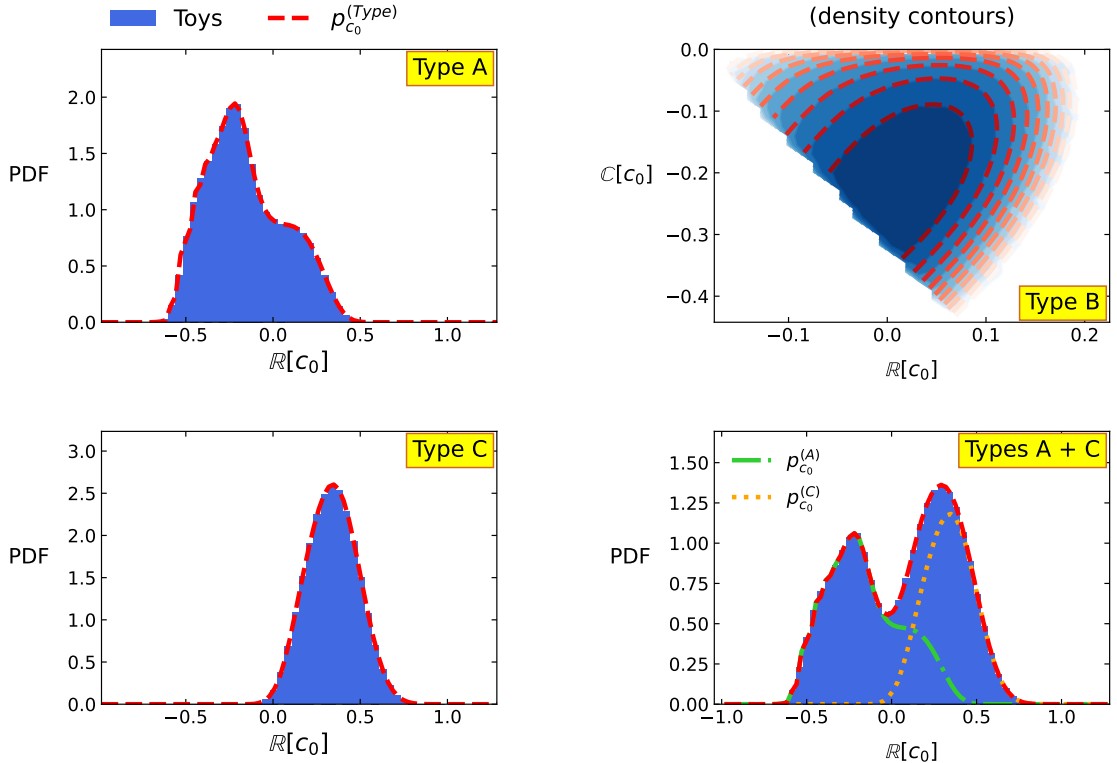

Figure 25: Densities of $c_0$ in the three different event categories. Type A and C events fall on the real axis whilst Type B events fall in the complex plane.

**PDF of $c_0$**

In section 5.2.4 we found that Type A and C events always lead to real $c_0$, but Type A solutions only exist when $\Delta_0 \leq \frac{1}{4}\beta_{c_0}^2$ and Type C solutions when $\Delta_0 > \frac{1}{4}\beta_{c_0}^2$. By contrast, Type B events fall in the complex plane. The toy distributions are shown in Fig 25. Even though Type B solutions do not contribute to the density over $q_{c_{\text{true}}}$, we may still study them here out of curiosity. Using $\beta_{c_0} = B + 3Ac_0$ we found

$$\Gamma^{\pm} = -\frac{1}{2}\beta_{c_0} \pm \frac{1}{2}\sqrt{\beta_{c_0}^2 - 4\Delta_0}, \qquad \frac{d\Gamma^{\pm}}{dc_0} = \frac{3}{2}A\left(-1 \pm \frac{\beta_{c_0}}{\sqrt{\beta_{c_0}^2 - 4\Delta_0}}\right), \qquad (D.7)$$

where the superscript $\pm$ distinguishes solutions with positive and negative square-roots. For Type A and C events we had to take the positive and negative square-root solutions respectively. Using the solutions of section 5.2.4 we can write the un-normalised PDFs as

$$\tilde{p}_{c_0}^{(A)}(c_0) = \int_{-\infty}^{\frac{1}{4}\beta_{c_0}^2} \left|\frac{d\Gamma^+}{dc_0} \cdot \frac{d\Delta_1}{d\Gamma}\right| p_{\Delta_1}(\Delta_1)\, p_{\Delta_0}(\Delta_0)\, d\Delta_0,$$

$$\tilde{p}_{c_0}^{(C)}(c_0) = \int_{\frac{1}{4}\beta_{c_0}^2}^{+\infty} \left|\frac{d\Gamma^-}{dc_0} \cdot \frac{d\Delta_1}{d\Gamma}\right| p_{\Delta_1}(\Delta_1)\, p_{\Delta_0}(\Delta_0)\, d\Delta_0.$$

(D.8)

For Type B events, we have the condition $\text{Im}\,[\Gamma]^2 = 3\text{Re}\,[\Gamma]^2$. Let us write $\Gamma = \Gamma_r\left(1 + \sqrt{3}i\right)$ and $\beta_{c_0} = \beta_{c_0,r} + i\beta_{c_0,i}$, and let us re-write Eq 82 as

$$\Gamma^2 + \beta_{c_0}\Gamma + \Delta_0 = 0. \qquad (D.9)$$

Separately expanding the real and imaginary parts of this equation gives

$$-2\,\Gamma_r^2 \,+\, \Gamma_r\,\beta_{c_0,r} \,-\, \sqrt{3}\,\Gamma_r\,\beta_{c_0,i} \,+\, \Delta_0 \,=\, 0,$$
$$\Gamma_r\left(\sqrt{3}\,\Gamma_r \,+\, \sqrt{3}\,\beta_{c_0,r} \,+\, \beta_{c_0,i}\right) \,=\, 0,$$

(D.10)

leading to the solutions

$$\Gamma_r \,=\, -\frac{1}{2}\left(\beta_{c_0,r} \,+\, \frac{\beta_{c_0,i}}{\sqrt{3}}\right), \qquad\qquad \Gamma_i \,=\, \sqrt{3}\Gamma_r,$$
$$\Delta_0 \,=\, 2\Gamma_r^2 \,-\, \Gamma_r\beta_{c_0,r} \,+\, \sqrt{3}\Gamma_r\beta_{c_0,i}, \qquad \Delta_1 \,=\, \Gamma^3 \,+\, \left(\frac{\Delta_0}{\Gamma}\right)^3.$$

(D.11)

We note that the solution $\Gamma_r = 0$ implies $\Gamma_i = 0$, and so these events are modelled as Type A. Since Type B events are dispersed throughout the complex plane, they have two degrees of freedom, and we have a $2 \to 2$ transformation. The Jacobian is

$$|\mathrm{Jac}(\Delta_0, \Delta_1; c_0)| \,=\, \begin{vmatrix} \frac{\partial \Delta_0}{\partial \mathrm{Re}[c_0]} & \frac{\partial \Delta_0}{\partial \mathrm{Im}[c_0]} \\ \frac{\partial \Delta_1}{\partial \mathrm{Re}[c_0]} & \frac{\partial \Delta_1}{\partial \mathrm{Im}[c_0]} \end{vmatrix},$$

(D.12)

with the un-normalised PDF

$$\tilde{p}_{\Gamma}^{(B)}(c_0) \,=\, |\mathrm{Jac}(\Delta_0, \Delta_1; c_0)|\; p_{\Delta_1}(\Delta_1)\; p_{\Delta_0}(\Delta_0).$$

(D.13)

To model only the real $c_0$, we combine the un-normalised PDFs according to

$$\tilde{p}_{c_0}^{(A+C)}(c_0) \,=\, \tilde{p}_{c_0}^{(A)}(c_0) \,+\, \tilde{p}_{c_0}^{(C)}(c_0).$$

(D.14)

To ensure they provide the correct relative contributions, it is important that we do no normalise the PDF until after combining event categories. In Fig 25, good agreement is observed between the toys and our PDFs in all categories.

**PDF of $c_1$**

Using the same approach as for $c_0$, for $c_1$ we have

$$\Gamma^{\pm} \,=\, -\frac{1}{2\xi}\beta_{c_1} \,\pm\, \frac{1}{2\xi}\sqrt{\beta_{c_1}^2 - 4\Delta_0}, \qquad \frac{\mathrm{d}\Gamma^{\pm}}{\mathrm{d}c_1} \,=\, \frac{3}{2\xi}A\left(-1 \pm \frac{\beta_{c_1}}{\sqrt{\beta_{c_1}^2 - 4\Delta_0}}\right).$$

(D.15)

We find that only Type B and C events result in real $c_1$, and will not consider Type A events further. We find that Type B events only exist when $\Delta_0 \leq \frac{1}{4}\beta_{c_1}^2$ and we take the solution $\Gamma^+$, whereas Type C events only exist when $\Delta_0 > \frac{1}{4}\beta_{c_1}^2$ and we take the solution $\Gamma^-$. The un-normalised PDFs are therefore

$$\tilde{p}_{c_1}^{(B)}(c_1) \,=\, \int_{-\infty}^{\frac{1}{4}\beta_{c_1}^2} \left|\frac{\mathrm{d}\Gamma^+}{\mathrm{d}c_1} \cdot \frac{\mathrm{d}\Delta_1}{\mathrm{d}\Gamma}\right| p_{\Delta_1}(\Delta_1)\; p_{\Delta_0}(\Delta_0)\; \mathrm{d}\Delta_0,$$
$$\tilde{p}_{c_1}^{(C)}(c_1) \,=\, \int_{\frac{1}{4}\beta_{c_1}^2}^{+\infty} \left|\frac{\mathrm{d}\Gamma^-}{\mathrm{d}c_1} \cdot \frac{\mathrm{d}\Delta_1}{\mathrm{d}\Gamma}\right| p_{\Delta_1}(\Delta_1)\; p_{\Delta_0}(\Delta_0)\; \mathrm{d}\Delta_0.$$

(D.16)

Fig 26 demonstrates that our PDFs agree well with the distribution of pseudo-experiments for real $c_1$ in Type B (left), Type C (middle) and combined (right) events.

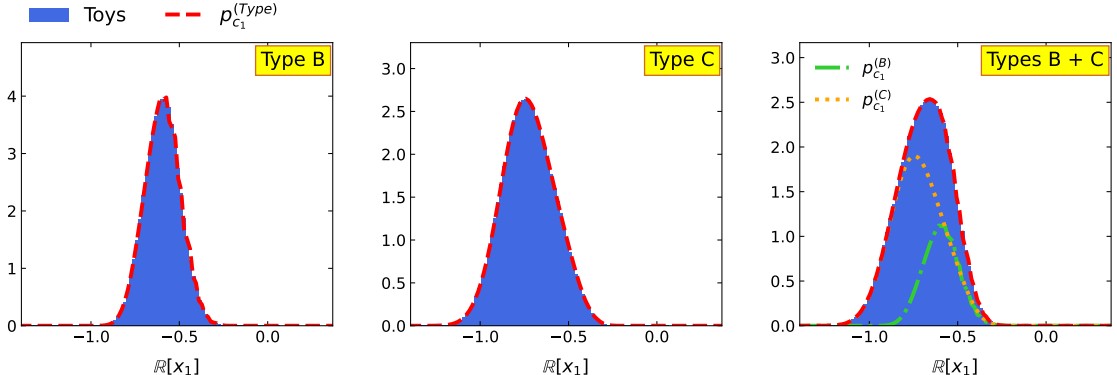

Figure 26: Densities of $c_1$ in different event categories. Type B and C events fall on the real axis. Type A events are not considered because they fall in the complex plane.

**PDF of $c_{\mathrm{MLE}}$**

We obtain $c_{\mathrm{MLE}}$ by selecting whichever of $c_0$ and $c_1$ is real and achieves the smallest $\chi^2$. If an event is Type A, we have seen that only $c_0$ is a real solution, and so this must be $c_{\mathrm{MLE}}$. Likewise $c_1$ is the only real solution for Type B events and must be $c_{\mathrm{MLE}}$. For Type C events, we must not allow $c_0$ and $c_1$ both to contribute density from the same event. Rather, the solution $c_0$ must contribute density only when it results in a smaller $\chi^2$ than $c_1$, and vice versa.

To achieve this, first we define

$$
\begin{aligned}
\tilde{\chi}^2\left(\bar{X}_l, \bar{X}_n; c\right) &:= \chi^2(\bar{x}; c) - |\bar{x}|^2 \\
&= c^2 \bar{L}^2 + 2c^3 \bar{M}^2 + c^4 \bar{N}^2 - 2c\bar{X}_l - 2c^2 \bar{X}_n,
\end{aligned}
\tag{D.17}
$$

which we interpret as "the part of the $\chi^2$ which varies as we profile $c$". This is the only part we need to consider when optimising the $\chi^2$ for a given event, since the rest is constant. We define $c_1^{(c_0)}$ as "the value of $c_1$ calculated from the event which produced $c_0$" (and vice versa for $c_0^{(c_1)}$). This is easily computed from $\Gamma$, which we already calculate as part of the inverse transformation. We then define the indicator function

$$
\mathcal{I}_{c_0}\left[\bar{X}_l, \bar{X}_n; c_0\right] = 
\begin{cases}
1 & \text{if } \mathrm{Im}\left[c_1^{(c_0)}\right] \neq 0, \\
1 & \text{if } \tilde{\chi}^2\left(\bar{X}_l, \bar{X}_n; c_0\right) < \tilde{\chi}^2\left(\bar{X}_l, \bar{X}_n; c_1^{(c_0)}\right), \\
0 & \text{otherwise},
\end{cases}
\tag{D.18}
$$

and vice versa $\mathcal{I}_{c_1}\left[\bar{X}_l, \bar{X}_n; c_1\right]$ for $c_1$. We use these factors to define PDFs for Type C events which only contribute density in the $c_0$ and $c_1$ channels when the solution is a global $\chi^2$ minimum

$$
\begin{aligned}
\tilde{p}_{c_{\mathrm{MLE}}}^{(C, c_0)}(c_0) &= \int_{\frac{1}{4}\beta_{c_0}^2}^{+\infty} \left|\frac{\mathrm{d}\Delta_1}{\mathrm{d}c_0}\right| p_{\Delta_1}(\Delta_1)\, p_{\Delta_0}(\Delta_0)\, \mathcal{I}_{c_0}\left[\bar{X}_l, \bar{X}_n; c_0\right] \mathrm{d}\Delta_0, \\
\tilde{p}_{c_{\mathrm{MLE}}}^{(C, c_1)}(c_1) &= \int_{\frac{1}{4}\beta_{c_1}^2}^{+\infty} \left|\frac{\mathrm{d}\Delta_1}{\mathrm{d}c_0}\right| p_{\Delta_1}(\Delta_1)\, p_{\Delta_0}(\Delta_0)\, \mathcal{I}_{c_1}\left[\bar{X}_l, \bar{X}_n; c_1\right] \mathrm{d}\Delta_0.
\end{aligned}
\tag{D.19}
$$

For brevity we represent the Jacobian factor as a single derivative whilst still computing it using the chain rule expansion. Combining all channels, the total un-normalised PDF for $c_{\mathrm{MLE}}$ is

$$
\tilde{p}_{c_{\mathrm{MLE}}}(c_{\mathrm{MLE}}) = \tilde{p}_{c_0}^{(A)}(c_{\mathrm{MLE}}) + \tilde{p}_{c_1}^{(B)}(c_{\mathrm{MLE}}) + \tilde{p}_{c_{\mathrm{MLE}}}^{(C, c_0)}(c_{\mathrm{MLE}}) + \tilde{p}_{c_{\mathrm{MLE}}}^{(C, c_1)}(c_{\mathrm{MLE}}).
\tag{D.20}
$$

Fig 27(left) shows a scatter plot of toys in the $(\Delta_0, \Delta_1)$-plane. Different colours tell us which event category (A, B or C) and solution ($c_0$ or $c_1$) the event corresponds to. We find that the four different channels are separated by distinct boundaries. Fig 27(right) shows the distribution of $c_{\text{MLE}}$. Events from different channels overlap and combine into a smooth three-peak structure. Our PDF is shown as the red dotted line, which is seen to agree well with the toys.

Fig 28 shows how the distribution of $c_{\text{MLE}}$ evolves as a function of $c_{\text{true}}$. We observe that the three-peak structure smoothly transitions into a distribution of two distinct peaks as $|c_{\text{true}}|$ becomes large. This behaviour is explained using a geometric perspective in Appendix C.

**PDF of $q_{c_{\text{true}}}$**

The final step in our forwards transformation is from $c_{\text{MLE}}$ to $q_{c_{\text{true}}}$. This PDF is already considered in detail in Section 5.2.7. Here we will visualise what is happening when we apply our numerical solution.

Fig 29(left panel) shows the scatter of toys in the $(\Delta_0, \Delta_1)$-plane, separated into the four individual channels following the same convention as Fig 27. In each of the subsequent panels, we plot only the toys which fall in a small interval around increasing values of $q_{c_{\text{true}}}$. We find that each value of $q_{c_{\text{true}}}$ corresponds to a horseshoe-like contour of events in the $(\Delta_0, \Delta_1)$-plane, for which the curvature increases with $q_{c_{\text{true}}}$. When we compute the corresponding PDF integral, we are integrating along the density of this contour. When $q_{c_{\text{true}}} = 0$ (second panel from the left), the two arms of the contour become inseparable and events fall along a radial line.

For a single test value of $q_{c_{\text{true}}} = 0.1$, Fig 30 compares the scatter of toys (left panel) with the values of $\Delta_1$ obtained when we scan $\Delta_0$ using Algorithm 1 (right panel). We find that there are either 0, 1 or 2 possible values of $\Delta_1$ for any given $\Delta_0$. Furthermore we find that the contour traced out by our numerical solution agrees well with the toys. Finally, each point is colour-coded according to which channel the solution is found in, and we see good agreement with the toys. Note that the toy distribution is obtained by sampling, and so the density of points represents the true event density. By contrast, our numerical solution is obtained by scanning across $\Delta_0$ and so the contour it traces is insensitive to the event density. This is why the tail of the 'horseshoe' is visible on the right-hand panel but not on the left.

Finally, Fig 31(right) shows a stacked histogram showing the distribution of toys as a func-

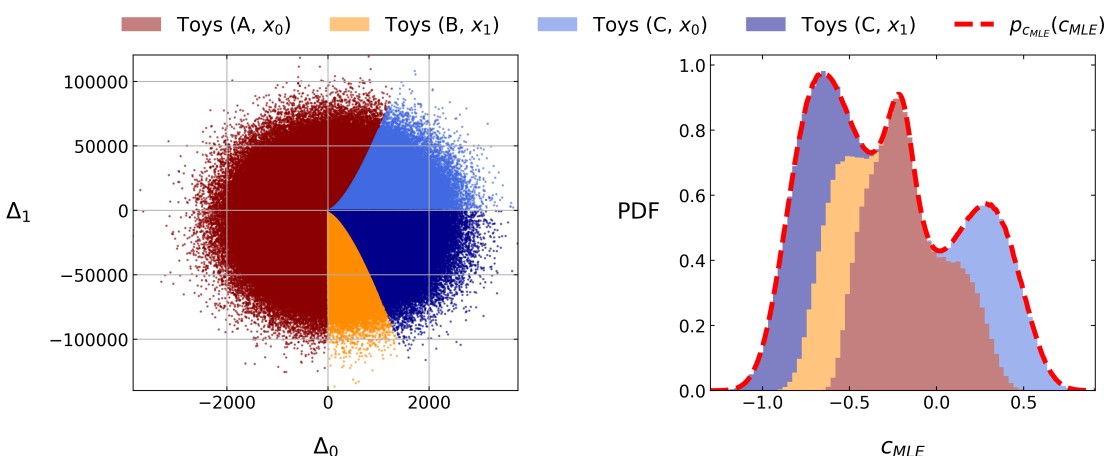

Figure 27: Left: scatter plot of toys in the $(\Delta_0, \Delta_1)$-plane. Right: stacked distribution of toys as a function of $c_{\text{MLE}}$ along with the modelled PDF.

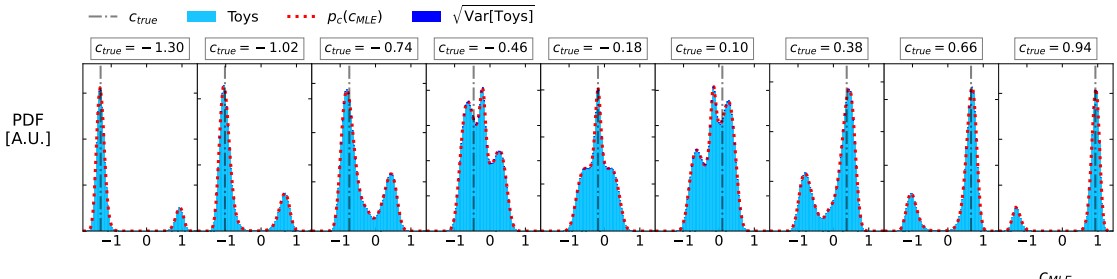

Figure 28: Distribution of $c_{\mathrm{MLE}}$ as a function of $c_{\mathrm{true}}$.

tion of $q_{c_{\mathrm{true}}}$, and its decomposition into the individual channels. We see that all four channels contribute across overlapping ranges of $q_{c_{\mathrm{true}}}$. We compare this with the PDF evaluated using our numerical method, and observe good agreement.

## E  Formula for the roots of a quartic equation

Consider the quartic equation defined as

$$q_4\, c_{\mathrm{MLE}}^4 \;+\; q_3\, c_{\mathrm{MLE}}^3 \;+\; q_2\, c_{\mathrm{MLE}}^2 \;+\; q_1\, c_{\mathrm{MLE}} \;+\; q_0 \;=\; 0\,, \tag{E.1}$$

with

$$\begin{aligned}
q_4 &= \bar{N}^2\,, \quad q_3 = 2\bar{M}^2\,, \quad q_2 = \bar{L}^2 - 2\bar{X}_n\,, \quad q_1 = -2\bar{X}_l\,, \\
q_0 &= q_{c_{\mathrm{true}}} + 2c_{\mathrm{true}}\bar{X}_l + 2c_{\mathrm{true}}^2\bar{X}_n - c_{\mathrm{true}}^2\bar{L}^2 - 2c_{\mathrm{true}}^3\bar{M}^2 - c_{\mathrm{true}}^4\bar{N}^2\,.
\end{aligned} \tag{E.2}$$

We solve the roots by writing

$$\begin{aligned}
q_5 &= 2q_2^3 - 9q_1q_2q_3 + 27q_4q_1^2 + 27q_3^2q_0 - 72q_4q_2q_0\,, \\[4pt]
q_6 &= q_5 + \sqrt{-4\left(q_2^2 - 3q_3q_1 + 12q_4q_0\right) + q_5^2}\,, \\[4pt]
q_7 &= \frac{q_2^2 - 3q_3q_1 + 12q_4q_0}{3q_4\sqrt[3]{\frac{1}{2}q_6}} \;+\; \frac{1}{3q_4}\sqrt[3]{\frac{1}{2}q_6}\,, \\[4pt]
q_8 &= \sqrt{\frac{q_3^2}{4q_4^2} - \frac{2q_2}{3q_4} + q_7}\,, \\[4pt]
q_9 &= \frac{q_3^2}{2q_4^2} - \frac{4q_2}{3q_4} - q_7 \\[4pt]
q_{10} &= \frac{1}{4q_8}\left(-\frac{q_3^3}{q_4^3} + 4\frac{q_3q_2}{q_4^2} - 8\frac{q_1}{q_4}\right)\,, \\[4pt]
q_{11} &= \frac{q_3}{4q_4}\,, \\[4pt]
q_{12} &= \frac{1}{2}q_8\,, \\[4pt]
q_{13} &= \frac{1}{2}\sqrt{q_9 - q_{10}}\,, \\[4pt]
q_{14} &= \frac{1}{2}\sqrt{q_9 + q_{10}}\,,
\end{aligned} \tag{E.3}$$

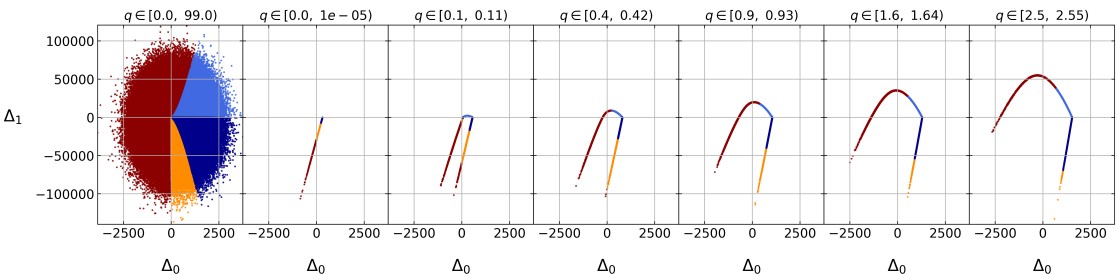

Figure 29: Left: scatter plot of toys in the $(\Delta_0, \Delta_1)$-plane. Moving across: toys in narrow slices of increasingly large $q_{c_{\text{true}}}$.

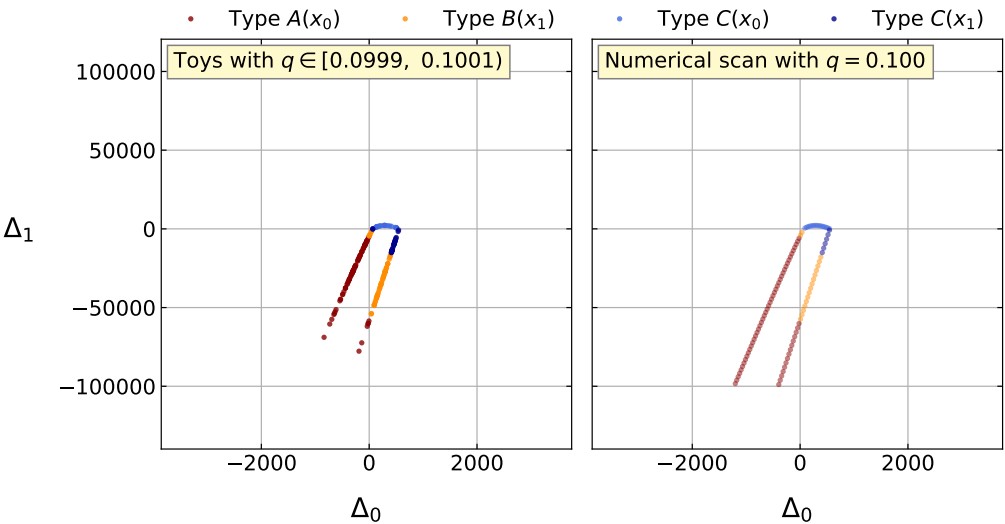

Figure 30: Left: distribution of toys in a narrow window around $q_{c_{\text{true}}} \approx 0.1$. Right: solutions identified by our numerical algorithm as we scan $\Delta_0$. Good agreement is observed.

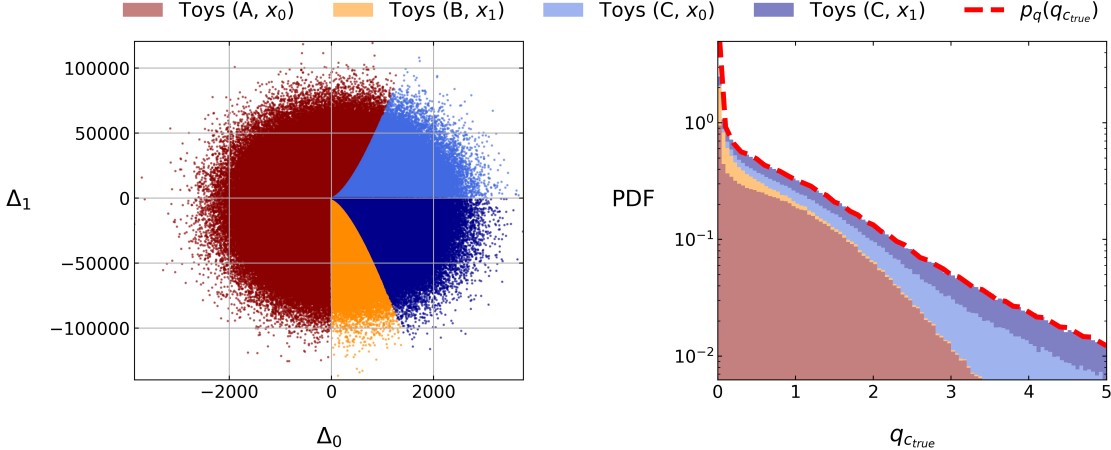

Figure 31: Left: distribution of toys in the $(\Delta_0, \Delta_1)$-plane. Right: stacked distribution of toys as a function of $q_{c_{\text{true}}}$ along with the modelled PDF.

and find the four solutions

$$
\begin{aligned}
c_{\text{MLE}}^{(1)} &= -q_{11} - q_{12} - q_{13}\,, \\
c_{\text{MLE}}^{(2)} &= -q_{11} - q_{12} + q_{13}\,, \\
c_{\text{MLE}}^{(3)} &= -q_{11} + q_{12} - q_{13}\,, \\
c_{\text{MLE}}^{(4)} &= -q_{11} + q_{12} + q_{13}\,.
\end{aligned}
\tag{E.4}
$$

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
