# Peer review of "Cover Your Bases: Asymptotic Distributions of the Profile Likelihood Ratio When Constraining Effective Field Theories in High-Energy Physics"

_SciPost Physics, doi:SciPost Phys. Core 6, 013 (2023)_

## Round 2 · Referee Report · Anonymous · 2022-10-19

Strengths

1- clarity

Weaknesses

1- limited originality and applicability

Report

The Asymptotic distribution of the Profile Likelihood Ratio (PLR) for quadratic dependence on the Wilson coefficient is obviously identical to the one derived (in half page) in Ref [12] of the manuscript, corresponding to the case in which the parameter of interest (\mu, in the notation of Ref [12]) is positive, by the identification \mu=c^2. The distribution in the case of linear dependence is the textbook Chi^2 result by Wilks. The two-parameters solutions described in Section 4 are original, as far as I can tell, but they constitute a rather trivial generalisation.

The most interesting part of the paper is Section 5, that identifies a strategy for the calculation of the Asymptotic distribution in the general case where both linear and quadratic terms contribute. However the study is limited to one single Wilson coefficient, making the resulting algorithm hardly useful in real EFT fits. Furthermore, computing the distribution in the idealised setup considered in the manuscript (Gaussian-distributed measurements of binned cross-sections) by pseudo-experiments is extremely fast as the Toy data consist in a bunch of Gaussians and the maximisation of the Likelihood a simple quadratic problem. Therefore it is unclear that an alternative strategy to compute the distribution along the line of the manuscript would be of practical relevance.

---

## Editorial Decision

published